# Global impacts of future urban expansion on terrestrial vertebrate diversity

Guangdong Li [1,2✉], Chuanglin Fang[1,2✉], Yingjie Li[3], Zhenbo Wang[1,2], Siao Sun [1,2], Sanwei He[4], Wei Qi [1,2], Chao Bao[1,2], Haitao Ma [1,2], Yupeng Fan[1,2], Yuxue Feng[1] & Xiaoping Liu [5,6✉]

Rapid urban expansion has profound impacts on global biodiversity through habitat conversion, degradation, fragmentation, and species extinction. However, how future urban expansion will affect global biodiversity needs to be better understood. We contribute to filling this knowledge gap by combining spatially explicit projections of urban expansion under shared socioeconomic pathways (SSPs) with datasets on habitat and terrestrial biodiversity (amphibians, mammals, and birds). Overall, future urban expansion will lead to 11–33 million hectares of natural habitat loss by 2100 under the SSP scenarios and will disproportionately cause large natural habitat fragmentation. The urban expansion within the current key biodiversity priority areas is projected to be higher (e.g., 37–44% higher in the WWF's Global 200) than the global average. Moreover, the urban land conversion will reduce local within-site species richness by 34% and species abundance by 52% per 1 km grid cell, and 7–9 species may be lost per 10 km cell. Our study suggests an urgent need to develop a sustainable urban development pathway to balance urban expansion and biodiversity conservation.

[1] Key Laboratory of Regional Sustainable Development Modeling, Institute of Geographic Sciences and Natural Resources Research (IGSNRR), Chinese Academy of Sciences (CAS), Beijing, China. [2] College of Resources and Environment, University of Chinese Academy of Sciences, Beijing, China. [3] Center for Systems Integration and Sustainability, Department of Fisheries and Wildlife, Environmental Science and Policy Program, Michigan State University, East Lansing, MI, USA. [4] School of Public Administration, Zhongnan University of Economics and Law, Wuhan, China. [5] Guangdong Key Laboratory for Urbanization and Geo-simulation, School of Geography and Planning, Sun Yat-Sen University, Guangzhou, China. [6] Southern Marine Science and Engineering Guangdong Laboratory (Zhuhai), Zhuhai, China. ✉email: ligd@igsnrr.ac.cn; fangcl@igsnrr.ac.cn; liuxp3@mail.sysu.edu.cn

The world's population is projected to reach 8.5–9.9 billion by 2050[1], with 55–78% living in urban areas[2]. This global explosion of urban population will undoubtedly cause an increasing demand for urban land. Indeed, urban land has already expanded much faster than urban population[3,4]. Although urban land covers only 0.2–2.4% of the global terrestrial surface[5], urban expansion has been a major driver of global land use change[6,7], which leads to habitat conversion and degradation[8–10], habitat fragmentation[11–13], and consequent biodiversity loss[7,14,15]. Global assessments show that urban expansion has caused about 50% loss of local within-site species richness (hereafter 'species richness') and 38% loss of total abundance of species in intensively used urbanized areas as compared to a naturally unimpacted baseline[16]. Most of the 20 Aichi Biodiversity Targets were not met by 2020 due to anthropogenic impacts, particularly the natural habitat loss and fragmentation caused by agricultural and urban land-use changes.

Future urban expansion is projected to be at an alarming rate (e.g., the global total urban area in 2100 will be roughly 1.8–5.9 times of that in 2000) under different climate change and development pathways[17,18]. To achieve the UN's Sustainable Development Goals (SDGs) by 2030, particularly SDG 11 (sustainable cities) and SDG 15 (sustainably managing habitat and halting biodiversity loss), it is urgent for us to understand how future urban expansion will impact the rate, magnitude, and spatial distribution of biodiversity loss. Such information will be crucial for policymakers to formulate a post-2020 agenda to maintain sustainable urbanization and prevent biodiversity loss. Despite the importance of biodiversity, few studies have assessed the multidimensional impacts of spatially-explicit projections of global urban expansion on biodiversity under various future scenarios[9], particularly how future urban expansion affects global habitat fragmentation, species richness loss, and species abundance loss. Although some studies have quantified the potential effects of urban expansion, they have mainly focused on only one aspect of impacts on biodiversity[9,15,19–22], or projected future urban expansion with a single scenario[23] and coarse resolution (>1 km).

We contribute to filling this knowledge gap by integrating comprehensive scenario analyses of future urban expansion and datasets on habitat and terrestrial biodiversity (three common terrestrial vertebrate taxa including amphibians, mammals, and birds) to quantify the impact of future urban expansion on global terrestrial biodiversity (see Supplementary Note 1). Specifically, we used a recently developed, spatially explicit urban expansion projection dataset (from 2020 to 2100, with 10-year intervals and 1 km spatial resolution) based on the five shared socioeconomic pathways (SSPs, see detailed description in Supplementary Note 2)[17]. We measured loss of natural habitat and biodiversity prioritization areas using spatial overlap analysis, and then assessed the impacts of future urban expansion on landscape fragmentation of natural habitats near urban areas. We further explored the relative percentage changes in species richness and species abundance on 1 km grids and examined the potential mean absolute change in species richness numbers on 10 km grids, based on the model estimates of biodiversity responses to future urban land cover change from the PREDICTS database[16]. Thus, this work provides insights into the relation between global urban land change and biodiversity. More importantly, it paves the way for more advanced studies on the social-ecological interaction related to future urbanization.

## Results

**Direct habitat loss**. According to the global projections of urban expansion under five SSPs[17] (Supplementary Note 3 and Supplementary Fig. 1), 36–74 million hectares (Mha) of land areas will be urbanized by 2100, representing a 54–111% increase compared with the baseline year of 2015. Among these, 11–33 Mha natural habitats (Supplementary Table 1) will become urban areas by 2100. Across SSP scenarios, the patterns of change in losses of total habitat, forest, shrubland, and grassland are consistent with the global projections of urban expansion (Fig. 1). In terms of urban encroachment on wetlands, wetland will undergo the largest loss under scenario SSP4 than under other scenarios. However, if the sustainable pathway of scenario SSP1 is properly implemented, this will enable us to conserve the global wetland. The greatest loss of other habitat will occur under scenario SSP3, but the minimal loss of other habitat will occur under scenario SSP1. Under the five different SSP scenarios, the United States, Nigeria, Australia, Germany, and the UK are consistently predicted to have greater habitat loss due to urban expansion (Supplementary Table 2).

There are obvious disparities in the hot spots and cold spots of habitat loss under the five SSP scenarios (Fig. 2 and Supplementary Figs. 2–6). Potential hot spots of habitat loss are concentrated in regions such as the northeastern, southern, and western coasts of the United States, the Gulf of Guinea coastal areas, Sub-Saharan Africa, and the Persian Gulf coastal areas. Under scenario SSP5, parts of central and western Europe will also become hot spots. However, under other scenarios, the cold spots will be particularly concentrated in eastern and southern Europe. East Asia and South Asia, which are represented by China, India, and Japan, are dominated by cold spots (Supplementary Figs. 2–6), because these regions may experience a decline in urban land demand from 2050 to 2100 (for examples in China, see Supplementary Figs. 7–11), although they are currently the most populous regions in the world.

Our scenario projections show that the largest natural habitat loss is expected to occur in the temperate broadleaf and mixed forests biome (except for scenario SSP3). In addition, many biomes will experience proportionate loss of natural habitat. These biomes include the tropical and subtropical coniferous forests biome, the temperate coniferous forests biome, the flooded grasslands and savannas biome, the Mediterranean forests, woodlands, and scrub biome, and the mangroves biome (Supplementary Table 3). Although the rate of future habitat loss is small at the global scale, it can be large in some areas. For example, the habitat in the temperate broadleaf and mixed forests may decrease by 1.4% under scenario SSP5. At the ecoregion scale, about 9% of 867 terrestrial ecoregions will lose more than 1% of habitat due to urban expansion (Supplementary Fig. 12). In the future, four ecoregions—the Atlantic coastal pine barrens, the coastal forests of the northeastern United States, and the Puerto Rican moist and dry forests—will experience more than 20% of habitat loss.

**Urban expansion threatens biodiversity prioritization schemes**. To reflect the potential impact of urban expansion on protected areas (Supplementary Note 4), the analyses presented here were based on the assumption that urban expansion within protected areas is not strictly restricted and can even occur in the currently gazetted protected areas (Supplementary Note 5, Supplementary Figs. 13 and 14). In 2015, urban areas with a total area of 30,594 km² were distributed in 28,152 protected areas, accounting for 12.6% of global protected areas (Supplementary Figs. 15 and 16). Moreover, 38% of the urban land-use changes within protected areas were due to the conversion of natural habitats into urban land between 1992 and 2015. If urban expansion continues without strict restrictions, 13.2–19.8% of the protected areas will be affected by urban land by 2100, and urban land will occur in

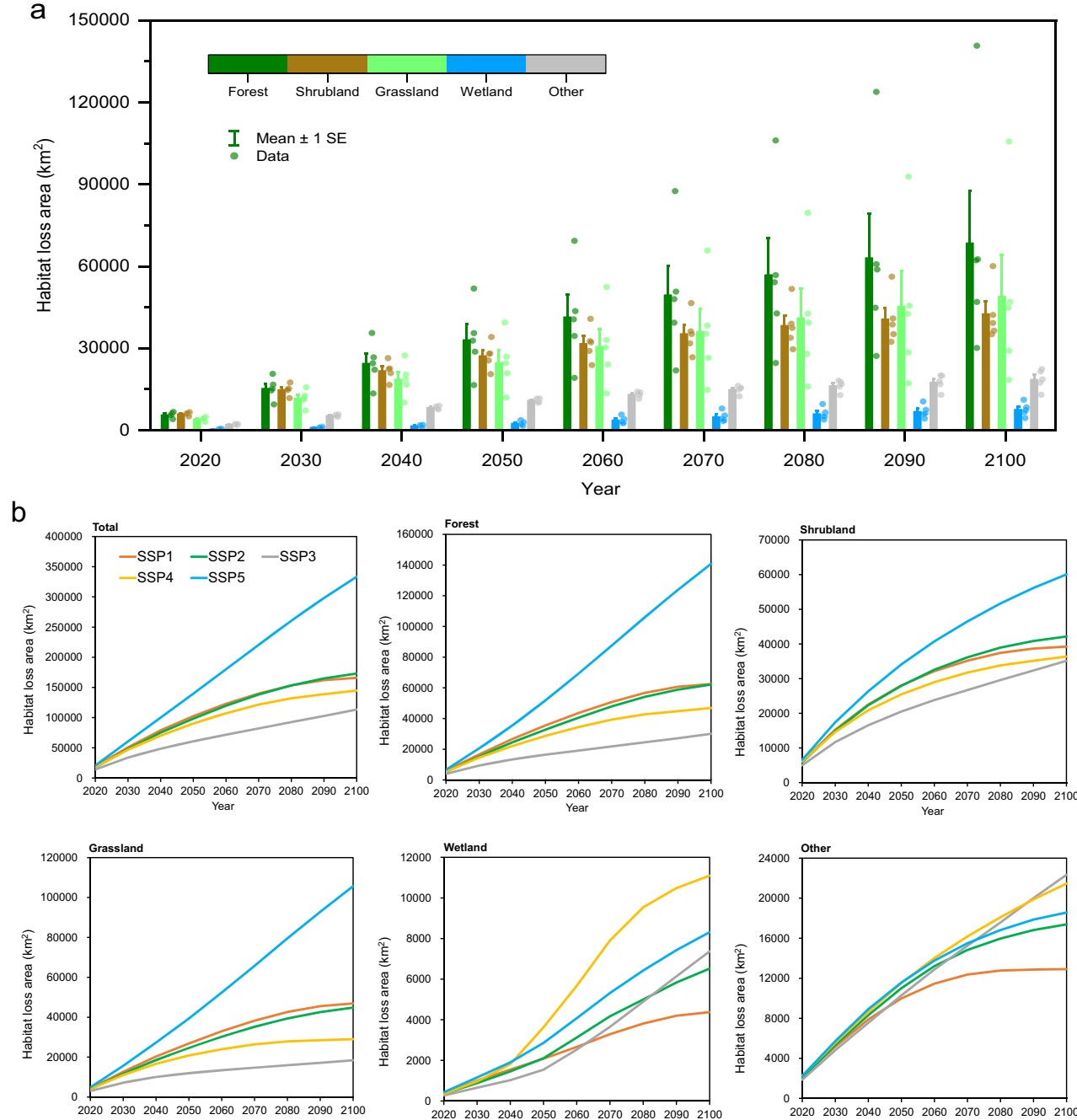

**Fig. 1 Future direct habitat loss due to urban expansion under SSP scenarios. a** The habitat loss by 2100 for each habitat type. Bars indicate the mean habitat loss area (five scenarios) for each habitat type. Error bars represent mean values ± 1 SEM for the loss of each habitat type under five scenarios, $n = 5$ scenarios. Points represent data in five scenarios. **b** The losses in total area, forest, shrubland, grassland, wetland, and other land.

29,563–44,400 protected areas with a total urban land area of up to 46,705–89,901 km² across the five SSP scenarios (the lowest and highest proportions of urban land in each protected area by 2100 under SSP3 and SSP5 scenarios are presented in Supplementary Figs. 17 and 18).

We also found that 0.90% of all terrestrial biodiversity hotspots (Supplementary Note 6), which are the world's most biologically rich yet threatened terrestrial regions[24], were urbanized in 2015. And this proportion (0.90%) is higher than that located in the rest of the Earth's surface (0.51%) in 2015. By 2100, the new urban expansion will additionally occupy 1.5–1.8% of hotspot areas under the five SSP scenarios (Supplementary Table 4). Five

biodiversity hotspots are projected to suffer the largest proportion of urban land conversion: the California Floristic Province (6–11%), Japan (6–8%), the North American Coastal Plain (4–8%), the Guinean Forests of West Africa (4–8%), and the Forests of East Australia (2–6%). In contrast, the East Melanesian Islands and the New Caledonia are almost unaffected by urban expansion. Biodiversity hotspots (e.g., the Guinean Forests of West Africa, the Coastal Forests of Eastern Africa, Eastern Afromontane, and the Polynesia-Micronesia) with few human disturbances in 2015 are projected to experience the highest percentage of future urban growth. Compared with the urban areas in 2015, by 2100, the urban areas in these four biodiversity

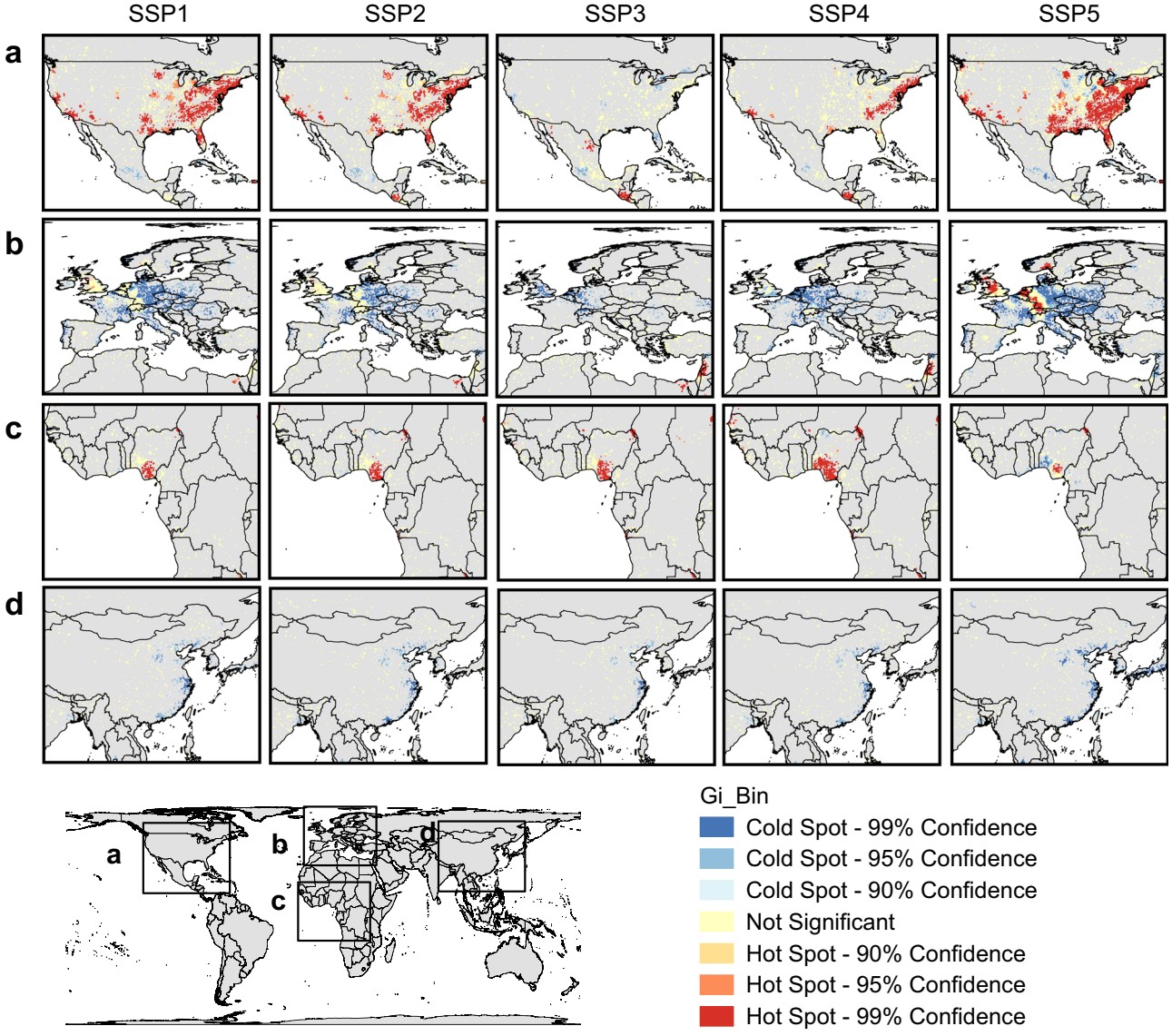

**Fig. 2 Future hot spots and cold spots of habitat loss due to urban expansion under SSP scenarios by 2100.** Figures for the United States (**a**), Europe (**b**), Africa (**c**), and China (**d**) are presented separately. The Gi_Bin identifies statistically significant hot spots and cold spots. Statistical significance was based on the *p*-value and *z*-score (two-sided), and no adjustments were made for multiple comparisons.

hotspots will experience a disproportionate increase of 281–708, 294–535, 169–305, and 33–337%, respectively.

The World Wildlife Fund (WWF) selected the ecoregions that are most crucial to the conservation of global biodiversity as Global 200[25] (Supplementary Note 7). However, about 93% of the Global 200 ecoregions will be affected by future urban expansion. Although the proportion of urban land in each ecoregion will be less than 1% in 2100, the urban area located in these ecoregions will experience an increase of 74–160% from 2015 to 2100 across the five SSP scenarios (Supplementary Table 4). Four ecologically vulnerable ecoregions that have the highest urban growth rates are the Sudd-Sahelian Flooded Grasslands and Savannas, the East African Acacia Savannas, the Hawaii Moist Forest, and the Congolian Coastal Forests. By 2100, the urban areas in these four ecoregions will increase by 877–9955, 527–646, 18–902, and 500–1037%, respectively.

The five SSP scenarios showed that the urban area is expected to increase by only 73–213 km² in the Last of the Wild areas[26] (see Supplementary Note 8 for descriptions about the Last of the Wild areas) by 2100 (Supplementary Table 4).

**Impacts of urban expansion on habitat fragmentation**. The increasing exposures of natural habitat to urbanized land use may cause long-term changes in the function and structure of the natural habitat that is adjacent to urban areas[13]. To examine this proximity effect, we investigated the impact of future urban expansion on the nearest distance between urban areas and natural habitat (i.e., the distance from patch edges of urban areas to patch edges of the nearest natural habitats) under different SSP scenarios. Although the global urban area is expected to increase by 36–74 Mha by 2100, the impacts of future urban expansion on adjacent natural habitat are disproportionately large. Future urban expansion will make urban areas much closer to patch edges of 34–40 Mha natural habitat, which will inevitably threaten the natural habitat and increase the risk of biodiversity decline. The effects of urban expansion on adjacent patch edges of natural habitats are remarkably different across different scenarios. Specifically, the area of affected adjacent natural habitat is expected to be 38.45, 34.24, 40.31, 37.84, and 39.42 Mha under SSP1 to SSP5 scenarios by 2100, with the smallest effect under scenario SSP2, and the largest effect under scenario SSP3.

Moreover, the scale of urban expansion does not correspond directly with the size of the impact. Several countries, including Mauritania, Algeria, Saudi Arabia, Western Sahara, and the United States, will have a large change in the distance from future urban areas to natural habitats due to urban expansion (Supplementary Table 5). Such effects also varied across different natural habitat types. The distance from the patch edges of urban areas to patch edges of (a) wetland, other land, and forest, (b) grassland, and (c) shrubland will generally be shortened by ~2000, ~1500 and ~900 m, respectively.

In addition to the effect on the distance to the habitat edge, urban-caused habitat fragmentation is also reflected in reducing mean patch size (MPS)[13], increasing mean edge index (edge density (ED), i.e., edge length on a per-unit area)[27], and enlarging isolation (mean Euclidean nearest neighbor distance, ENN_MN)[28] (Fig. 3). Taking the global ecoregions as the analysis unit, we found that within a 5 km buffer of urban areas, the median of MPS of natural habitats tends to show an overall decline trend, and the segmentation and subdivision of habitats become more obvious as future urban land expands. The median of MPS is the largest under scenario SSP1, followed by SSP4, SPP2, and SSP3 with some fluctuations in between, and the smallest MPS is found with the most fragmented landscape under scenario SSP5. A smaller patch size indicates that the inner parts of the habitat are subject to higher risk of being influenced by external disturbance. Future urban expansion also tends to cause an increase in the ED of natural habitat, which is often linked with smaller patches or more irregular shapes, and therefore poses a threat to biodiversity that influences many ecological processes (e.g., the spread of dispersal and predation)[13,27,28]. Scenario SSP1 shows the best performance in maintaining a low habitat ED and a high level of biodiversity conservation. However, under scenario SSP5, ED will experience a rapid increase in the second half of the 21st century. Meanwhile, the ENN_MN will increase substantially in the future, suggesting that areas with the same habitat type will become increasingly isolated, irregular, dispersed, or unevenly distributed due to the barrier of urban land. This will affect the speed of dispersal and patch recolonization. Scenario SSP1 is also most conducive to maintaining the proximity of natural habitats with the same habitat type. Other scenarios show relatively similar performance.

**Impacts of urban expansion on terrestrial biodiversity.** We focus on biodiversity in three common vertebrate taxa (i.e., amphibians, mammals, and birds) in our analyses. Future land system conversion to urban land will cause an average of 34% loss in the overall relative species richness. Land conversion from dense forest, mosaic grassland and open forest, mosaic grassland, and bare and natural grassland to urban land will cause the highest overall relative biodiversity loss (48%, 95% confidence interval (CI): 34–59% on a 1 km grid). These land systems with a high risk of biodiversity loss are concentrated in the United States, Europe, and Sub-Saharan Africa (Supplementary Fig. 19). Overall, the negative effect of future urban expansion on the total abundance of species will be more pronounced than that on species richness. Urban land changes will result in an average of 52% overall loss in relative total abundance of species. In particular, the losses of dense forest, natural grassland, and mosaic grassland, due to conversion to urban land, will lead to a high risk of species loss (62%, 95% CI: 38–76%).

In terms of the number of species (i.e., all amphibians, mammals, and birds), future urban expansion will cause an average loss of 7–9 species and a loss of up to ~197 species per 10 km grid cell by 2100 across the five SSP scenarios (Fig. 4 and Supplementary Fig. 20). Species loss is most likely to be concentrated in Sub-Saharan Africa (particularly the Gulf of Guinea coast), the United States, and Europe. In addition, southeastern Brazil, India, and the eastern coast of Australia are also relatively high-risk areas. However, the specific effects of urban expansion vary substantially across different SSP scenarios. For instance, under scenario SSP5, urban expansion will pose a fatal threat to the global species richness in areas with urban development potential (species richness loss will occur in ~740 Mha land areas), whereas under the divided pathway (SSP4) and regional rivalry pathway (SSP3) scenarios, urban expansion will threaten the richest biodiversity hotspots, such as Sub-Saharan Africa and Latin America (Supplementary Fig. 20).

We also found a loss of up to 12 species of threatened amphibians, mammals, and birds (including vulnerable, endangered, or critically endangered categories defined in the IUCN Red List), and a loss of up to 40 species of small-ranged amphibians, mammals, and birds (small-ranged species are species with a geographic range size smaller than the median range size for that taxon)[29] due to future urban expansion by 2100. There are a few scattered areas that will be hotspots for the loss of threatened species, such as West Africa, East Africa, northern India, and the eastern coast of Australia (Supplementary Fig. 21). The loss of small-ranged species will concentrate in fewer areas (Supplementary Fig. 22). We have identified 30 conservation priority ecoregions with high risks of habitat loss and small-ranged species loss due to future urban expansion (Supplementary Table 6). These conservation priority ecoregions are all found in Latin America and Sub-Saharan Africa (Supplementary Fig. 23). However, some hotspots outside of these conservation priority regions, such as tropical Southeast Asia, the west coast of the United States, and northern New Zealand, will also be affected (Supplementary Fig. 23).

The top 5% 10 km grid cells with the highest loss in species richness (28–38 species potentially being lost) scatter across adjacent urban areas. However, only 6.4–8.6% of these regions are covered by the current global network of protected areas. These areas are often overlooked, and thus receive relatively low conservation spending. Ecoregions in Sub-Saharan African, Central and South America, Southeast Asia, and Australia will be responsible for the top 43% of average species loss across the SSP scenarios (Fig. 5). Kenya, Swaziland, Brunei, Zambia, Republic of Congo, and Zimbabwe will face the largest potential species richness loss (approximately > 29 species lost per 10 km grid cell) under all five SSP scenarios (Supplementary Fig. 24 and Supplementary Table 7).

## Discussion

This study analyzed the multifaceted effects of future urban expansion on global terrestrial biodiversity under five SSP scenarios. Changes in global terrestrial biodiversity were captured by natural habitat loss, encroachment in biodiversity prioritization areas, habitat fragmentation, and loss of species richness and abundance in three common vertebrate taxa (i.e., amphibians, mammals, and birds). Overall, we found that future urban expansion will remarkably influence global terrestrial biodiversity, but the future rate, magnitude, and spatial distribution of habitat and biodiversity decline depend on the specific development path chosen by humans. Nevertheless, there are still opportunities to adjust future urban development trajectories and to intervene against the long-term negative impacts of urbanization on biodiversity. In particular, if the sustainable pathway (i.e., scenario SSP1) of urban expansion is properly implemented, humans will be able to maintain a relatively low natural habitat loss, low habitat fragmentation, and a high level of species conservation (SSP1 with the lowest cropland expansion and higher forest area is also generally more beneficial for biodiversity conservation

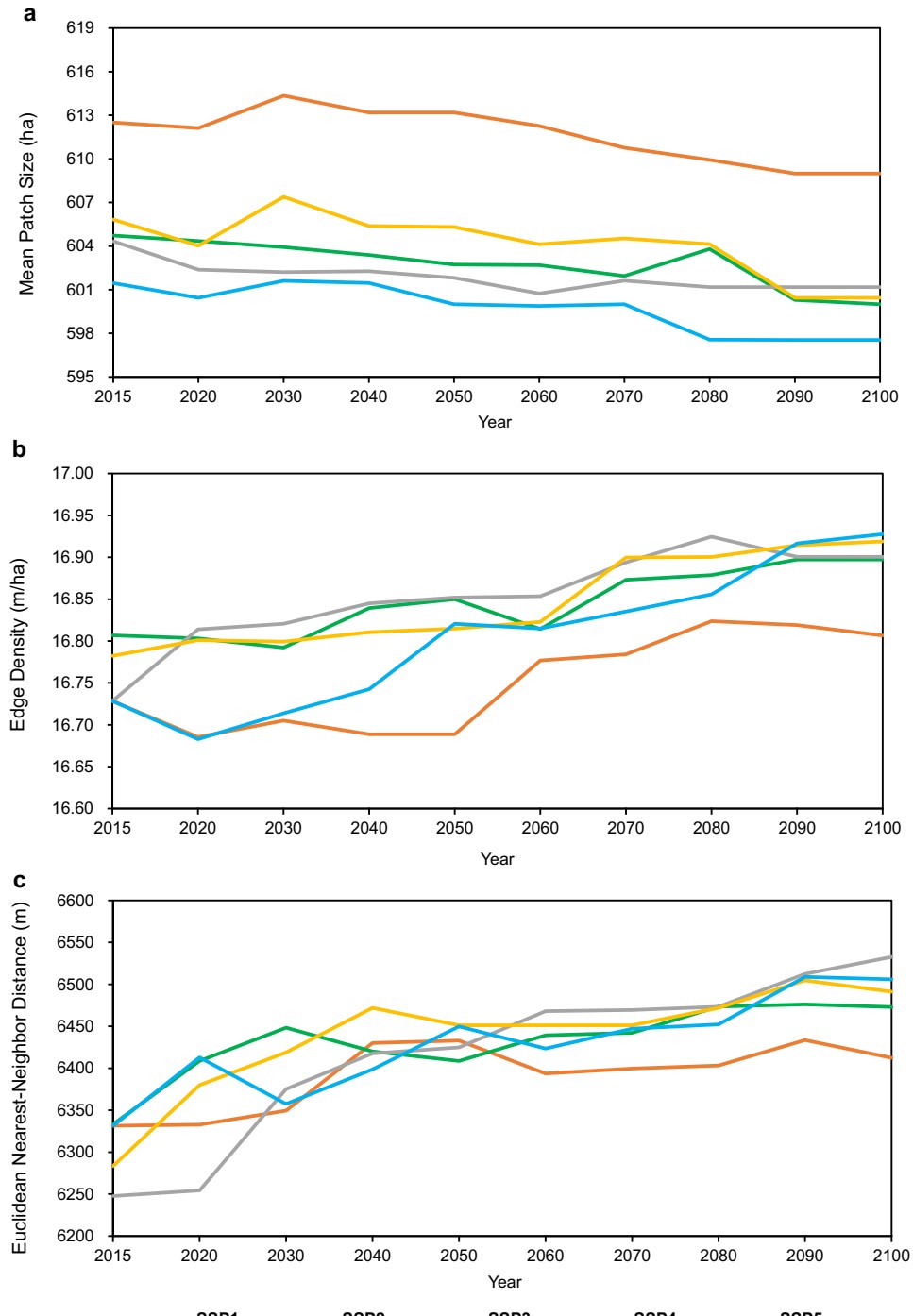

**Fig. 3 Future urban expansion effects on habitat fragmentation under SSP scenarios. a** Mean patch size (MPS), **b** edge density (ED), **c** mean Euclidean nearest-neighbor distance (ENN_MN).

according to several existing estimations[30,31]). Several other key findings in our study are worth mentioning, which we detail below.

First, the magnitude and spatial range of the impacts of future urban expansion on global terrestrial biodiversity are profound and thus should not be ignored. Specifically, natural habitat loss will increase by 694–1509% from 2020 to 2100 across the five SSP scenarios, and the loss in natural habitat showed substantial spatial heterogeneity across different habitat types, biodiversity hotspots, biomes, and ecoregions. Notably, different habitat types, biodiversity hotspots, biomes, and ecoregions have different

ecological functions and biodiversity values[29], which future conservation policies should take into account[9,19,20,32]. Moreover, the key biodiversity hotspots and ecologically vulnerable ecoregions that are currently less disturbed by humans will suffer the highest percentage of urban growth[15,20]. Yet, these areas are often located in less economically developed countries that cannot afford high expenses for biodiversity conservation. Thus, there is an urgent need to formulate effective conservation policies and conservation funding programs to formulate proactive land-use planning in these countries. Moreover, to effectively tackle future impacts on the global protected areas posed by urban expansion,

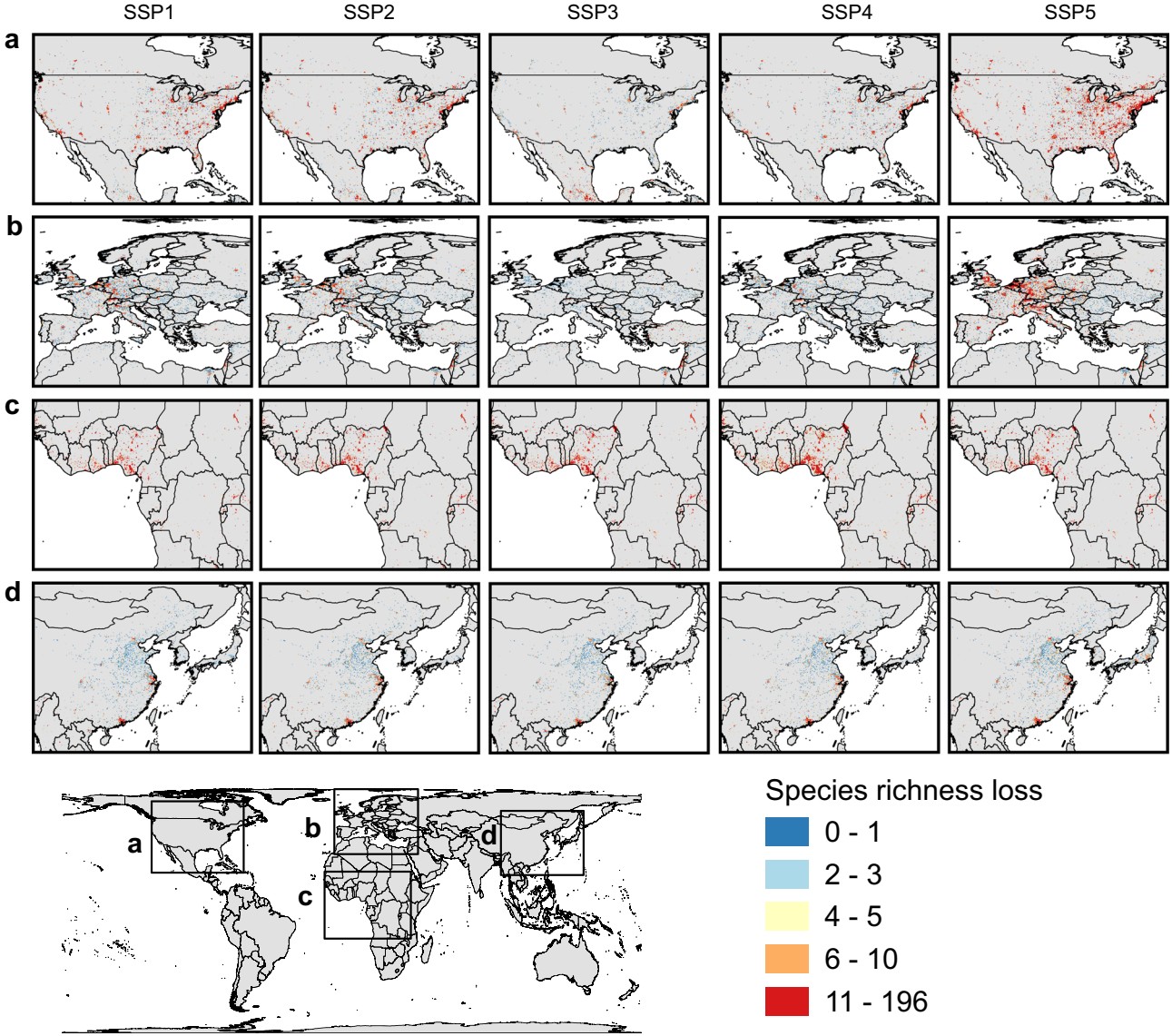

**Fig. 4 Potential biodiversity loss due to future urban expansion under SSP scenarios.** The biodiversity loss in terms of the number of terrestrial vertebrate species (amphibians, mammals, and birds) lost per 10 km grid cell in the North America (**a**), Europe (**b**), the Gulf of Guinea coast (**c**), and East Asia (**d**).

we need to extend protected areas to more vulnerable and critical biodiversity-rich areas, and to improve their durability and effectiveness[33,34].

Second, future urban expansion will disproportionately affect the natural habitat around the urban areas as urban areas get closer to the patch edges of natural habitat, and thus increase the risk of biodiversity loss[13]. In addition, natural habitats that are occupied by urban areas will show obvious fragmentation—including decreased MPS, increased mean ED, and increased isolation—due to the urban land barrier and destruction. Fragmentation has persistent and profound effects on biodiversity[13]. Negative impacts on biodiversity often result from the decrease in habitat connectivity and ecosystem integrity and the aggravation of edge effects[35,36]. These findings suggest that compact development and smart urban growth will continue to be key factors in reshaping urban morphology in the future. Moreover, establishing ecological corridors in fragmented areas caused by urban expansion may effectively improve habitat connectivity and facilitate species migration[15]. Notably, sustainable pathway SPP1 is an optimal choice, because it envisions a development path of

compact urban form[2], minimal urban sprawl, and urban de-concentration[37].

Third, although urban land conversion is on a smaller scale than cropland change, which is commonly considered as the leading driver of biodiversity loss[32], the largest percentage of loss in species richness and abundance often occurs in the urban land conversion process (i.e., conversion from a highly ecological land-use class to a human-dominated built environment)[16]. Moreover, the impact of urban expansion on species loss is not evenly distributed[20]. Consistent with this idea, our findings suggest that urban expansion will cause extensive biodiversity loss in many regions worldwide, but losses of threatened and small-ranged species due to future urban expansion will concentrate in fewer areas. This suggests that safeguarding habitat and species from urban expansion in a relatively small number of ecoregions (hotspots) could have a disproportionally large benefit in terms of avoiding species loss[20]. However, those regions with high species richness loss (particularly the low-income countries) are only partially covered by the current biodiversity prioritization schemes. Notably, the greatest challenge in global biodiversity

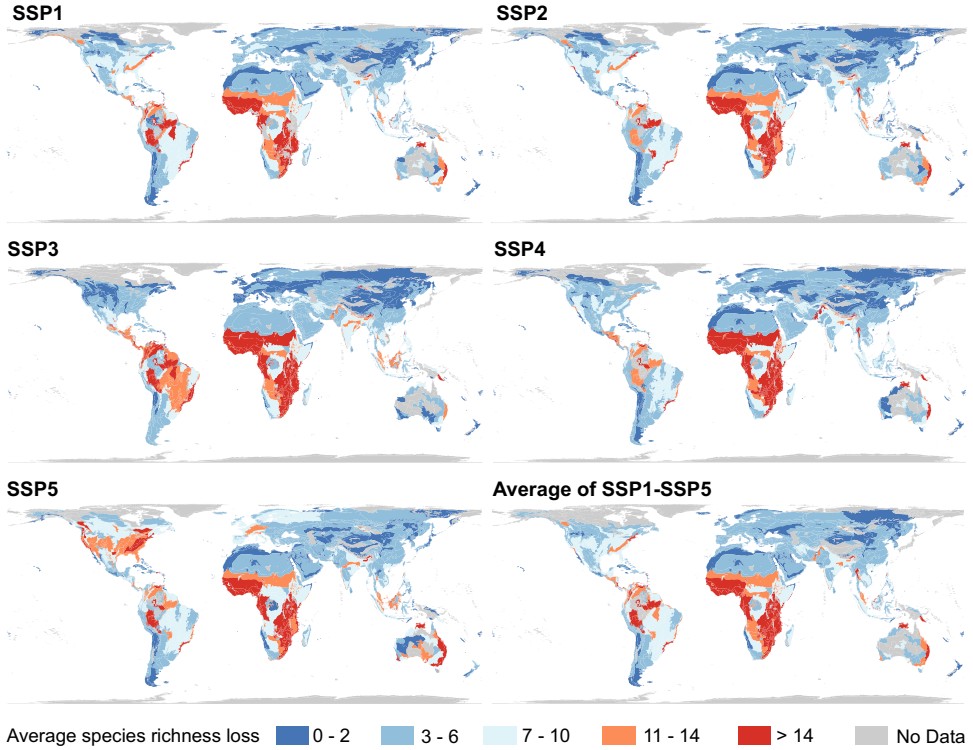

**Fig. 5 Average potential biodiversity loss per 10 km grid cell in ecoregions due to future urban expansion under SSP scenarios.** The mean potential biodiversity loss represents the average number of terrestrial vertebrate species (amphibians, mammals, and birds) lost per 10 km grid cell.

conservation lies in the least developed countries (mostly in Sub-Saharan Africa) that often have huge potential for urban expansion, greater risk of biodiversity loss, and higher investment gap in biodiversity conservation. Thus, the conservation priority areas must be reevaluated, updated, and better integrated into conservation management to avoid overlooking such areas with high risks of biodiversity loss[32]. Moreover, for these countries, choosing an appropriate urban development pathway may be the most urgent requirement for balancing urban expansion and biodiversity conservation.

Our study seeks to advance the field in several aspects. First, previous studies at the global scale mainly focused on how future urban land expansion affects natural habitats and biodiversity prioritization areas[14,15,19,20], and paid less attention to its impact on habitat fragmentation and losses of species richness and abundance[9,16]. We sought to address this understudied question by quantifying the effects of future urban expansion on habitat fragmentation. In addition, we combined land systems data, species distribution maps (consisting of three common vertebrate taxa, namely amphibians, mammals, and birds), and future urban expansion simulation data to assess losses of species richness and abundance due to future urban expansion. Second, previous studies on the effects of urban expansion often used datasets with lower resolution (e.g., 5 km) to simulate future urban land changes. Yet, low-resolution data can cause overestimation of future urban expansion[38]. To reduce overestimation, we used more recent datasets with higher resolution (1 km). Besides, we adopted a more advanced Future Land-Use Simulation (FLUS) model that can couple with the latest SSPs[17] to simulate future urban expansion[39]. This model can explicitly simulate the spatial trajectories of multiple land cover changes under different scenarios by coupling human-related and natural environmental impacts[39]. The characteristics and advantages of the FLUS model are the self-adaptive inertia mechanism and roulette selection

mechanism, which can reflect the complexity and uncertainty of land-use changes in the real world[39]. These advantages are not available in other models (e.g., SLEUTH model[23] and SELECT model[18]). In particular, these other models often set more priority to the edge growth transition rule or existing urban areas and thus are limited in simulating other urban development processes, such as the leapfrog development pattern[39]. Moreover, these other models often simplify the randomness and complexity of the urban expansion process and thus affect the simulation performance[17,39]. Compared to McDonald et al.[9], our estimated habitat loss due to future urban expansion (11–33 million ha by 2100) is smaller than their estimation (i.e., 29 million ha between 2000 and 2030). Notably, the projection of urban expansion used by McDonald et al.[9], which is up to 2030 derived from Seto et al.[15]. They might have overestimated urban growth (Supplementary Figs. 25–28). This may be due to coarse spatial resolution, the specific datasets they used, or their particular model configurations. Indeed, out of the 30 conservation priority ecoregions, we identified 19 ecoregions with high species number of vertebrates but high future urban growth potential, which were not covered by McDonald et al.[20]. In addition, the spatial allocation algorithm embedded in these models may not effectively capture urban population shrinkage (except for the study of Seto et al.[15], because their forecasts go out to 2030). For example, the simulation that our research was based on showed that a few regions (e.g., China and other Asian countries) will have a relatively low urban expansion rate or even a decline in urban land demand after 2050 (see Supplementary Figs. 7–11 for China).

Despite these strengths, some limitations and unaddressed issues in this research are noteworthy. First, we only examined how future urban expansion will directly affect terrestrial biodiversity without considering its indirect effects. Indeed, telecouplings[40,41] through supply chains (e.g., global trade[42], land grab[43]) and other human–nature interactions over distance can

also cause land-use change[9]. Thus, land-use displacement (particularly cropland displacement) induced by urban expansion may cause natural habitat loss in other areas[8], which further causes biodiversity loss in these other areas. Although it is currently difficult to accurately quantify such indirect effects of urban expansion, fine-scale trade-flows data (e.g., the world food trade) may facilitate us to indirectly analyze the embodied biodiversity loss. Future research may use environmentally extended input–output database and network analysis methods to measure these indirect effects. Second, due to limited technologies, the lack of some data makes it impossible to address some other potentially important questions. For instance, due to a lack of available future long-term and high-resolution global land cover change and land-systems simulation data, we could not accurately estimate the dynamic changes in habitat and biodiversity losses due to urban land conversion. Future research can address this issue by simulating future global land cover and land-systems change maps under the SSP framework. Meanwhile, because habitat fragmentation due to future urban expansion can threaten biodiversity, it is necessary to investigate the effects of habitat fragmentation caused by urban expansion on biodiversity degradation. However, because of the lack of global dynamic biodiversity data, we did not cover this aspect, although it is an important question proposed in previous research[13]. Third, we based our study on the global projections of urban expansion in previous research that used input data aggregated on 32 world regions[17]. Many of these macro-regions (e.g., EU15, LAM-H, and MEA-M) contain more than one country, which may lead to uncertainties in the future forecast of national-scale urban land demand and the specific locations of future urban land expansion. These uncertainties may affect the accuracy of our estimated impacts of future urban land expansion on biodiversity at the country and ecoregional levels. Nevertheless, we believe that future research can provide more accurate answers to these questions with the help of a more refined SSP database. Fourth, because urban expansion has occurred within protected areas in the past few decades[19], we assumed that this trend of urban expansion within protected areas will continue without restrictions in the future. If we assume no urban development in protected areas, then by 2100, urban expansion within protected areas is expected to decrease by 16,111–59,306 km$^2$ across SSP scenarios, which accounts for 4.59–8.02% of newly-added urban expansion between 2015 and 2100. Nevertheless, according to our model assumption, these urban areas will equivalently occur in other potential areas if they do not occur in protected areas. Thus, it is plausible that our work is subject to two caveats: (a) we may have overestimated the impacts of urban expansion on protected areas, especially on direct occupation, because the protected areas around the world vary in their enforcement effectiveness in preventing urban expansion and alleviating human pressure[33,44]. The enforcement effectiveness of protected areas depends on multiple factors, such as resources used to manage protected areas, law enforcement, and governance quality[45]. However, it is still difficult to accurately identify future enforcement effectiveness of protected areas, because protected area downgrading, downsizing, and degazettement (PADDD) are becoming increasingly prevalent in some developed countries (e.g., the United States)[34]. Even strictly protected areas (i.e., IUCN categories I and II) are subject to increasing human pressure, which suggests that the IUCN management category cannot inhibit the aggravation of human pressure[46]; (b) it should be noted that not all urban land within protected areas has negative consequences in an ecological sense, nor that all urban expansion in protected areas reflects a violation of the legal protection of protected areas. We should understand urban expansion in protected areas differently and seek potential solutions to sustain the harmonious

coexistence of human and nature in the future. Finally, it is plausible that both urban expansion and biodiversity change may interact with future climate change, which we did not investigate in our study. For instance, urban expansion can accelerate climate change (particularly the change in urban microclimate)[47], cause warming in urban and surrounding areas (urban heat island), and increase the intensity of precipitation and runoff in local areas[48]. Moreover, future climate change—such as more extreme weather events in urban areas and faster sea-level rise in most coastal urban areas[48]—can also affect urban development, urban environment, and urban expansion process[49,50]. In addition, climate change affects all aspects of life on Earth, perhaps with the most pervasive impact on species redistribution[51], such as poleward and elevational range shifts[51,52]. Climate-driven changes in species redistribution, which may be more profound in the future when climate change intensifies, will affect global biodiversity patterns and shape new hotspots. Thus, future research on the effect of urban expansion on biodiversity needs to take into account the effect of climate change in this process.

Our study also provides important practical implications for policy makers. First, our findings suggest that potential future biodiversity threats due to urban expansion should be incorporated in current and long-term biodiversity conservation schemes. We recommend that more attention should be given to urbanization when setting global biodiversity conservation goals for the post-2020 global biodiversity framework in the United Nations Biodiversity Conference (CBD COP 15). Second, policymakers should coordinate and solve the tradeoffs between multiple SDGs across the globe, including sustainable cities (SDG 11) and sustainably managing habitat and halting biodiversity loss (SDG 15). In addition, they should explore practical solutions to balance urban development and biodiversity targets, such as nature-based solutions (NBS), sustainable urban planning and design, low-impact green infrastructures, and establishing connected ecological corridors in which human and nature coexist. Third, because a large proportion of future urban expansion will occur at the regional scale—metropolitan areas, metropolitan area belts, large metropolitan area belts, and urban megalopolis[53]—rather than at the city scale, collaborative and comprehensive governance across regions and countries may help mitigate the disturbance of urban expansion on natural habitats and biodiversity. Fourth, to facilitate wider coordination, we propose to update the IUCN Protected Areas Management Category. This can be achieved by (1) assigning IUCN categories to protected areas with unknown or missing categories (roughly 30% of all protected areas) and increasing the strictness of protection for protected areas[46], and (2) developing a new category system of protected areas to represent the specific role of different protected areas in biodiversity conservation rather than for management purposes[54]. Of course, it is undeniable that governing conflicting demands of consumption on ecosystems and ensuring their integrity is a very challenging task that requires a joint effort from different stakeholders around the world.

## Methods
**Forecasting future urban expansion**. We base our study on the global projection of urban expansion dataset with five SSP scenarios (see detailed assumptions about urbanization patterns and urban planning for five SSPs in Supplementary Tables 8 and 9) and a 1 km resolution[17]. This dataset was developed using a panel data regression to estimate future urban land demand. First, Chen and colleagues built panel data regression using historical urban land data (obtained from the GHSL dataset[55] for the years 1975, 1990, 2000, and 2014) and statistical data to estimate per capita urban land demand from urbanization rate (i.e., urban population/total population) and per capita gross domestic product (GDP) (obtained from the World Bank[56] and United Nations[57]). Then, the established panel data regression model was used to predict, for each scenario, per capita urban land demand from the future per capita GDP and urbanization data obtained from the SSP database (https://tntcat.iiasa.ac.at/SspDb). Thus, the regional urban land demand in a future

year $t$ was obtained by multiplying the estimated urban land demand per capita at year $t$ by the projected total population in a region at year $t$. The data used in the regression are for the 32 macro regions that were created in the SSP database by aggregating the world's countries or regions. Subsequently, the FLUS model[39] was employed to allocate and simulate the spatially-explicit distribution of future urban expansion based on the forecasted urban land demand and the urban development potential for the 32 regions. It is assumed that even if the population shrinks, the urbanization rate and GDP can still grow, and there is still a certain urban land demand. Only when population, urbanization, and economic growth all stagnate, the urban land demand will stagnate, and then urban land may not continue to expand. Thus, the scenario of regional population shrinking but urban area growing (e.g., cities in Eastern Europe, see Supplementary Figs. 29 and 30) can be captured by this model framework. For regions with a decline in urban land demand, it was assumed that the land conversion from nonurban land to urban land is irreversible. In the spatial simulations, the substantial conversion of urban land to non-urban land will not occur. It is worth noting that in the urban growth simulation, if the estimated urban land area of a region in the future is smaller than its current urban area, then the existing urban lands will remain unchanged because future urban land demand can be met by the existing area[17].

**Habitat loss**. We applied spatial overlap analysis to examine the spatial overlap between future spatially-explicit urban expansion and natural habitat in 2015, to aggregate the natural habitat losses across different biological and geographical units. The land cover map for the year 2015 was based on the European Space Agency (ESA) Climate Change Initiative (CCI) Land Cover product[58]. This land cover map was reclassified into six aggregate classes (cropland, urban land, forest, shrubland, grassland, and other) based on an updated reclassification system (see Supplementary Table 1 for details). All land classes, except for cropland and urban land, are defined as natural habitats[8,9,59]. We used the Optimized Hot Spot Analysis in the ArcGIS Pro v2.5 to identify statistically significant spatial hot spots and cold spots of natural habitat loss due to urban expansion. To provide more ecologically meaningful results, we further quantified the spatial distribution of future natural habitat loss due to urban expansion across biomes and ecoregions[60] around the world. The world database on protected areas[61] was used to examine the spatial overlap between future urban expansion and the locations of these protected areas (see assumptions for protected areas under SSPs in Supplementary Table 9). Based on historical data analysis from 1992 to 2015 showing that urban expansions had occurred in protected areas, we assumed no strict constraints for the encroachment of urban expansion on protected areas (Supplementary Figs. 13, 14 and Supplementary Table 4).

**Habitat fragmentation**. To date, habitat fragmentation has been identified as the primary cause of global biodiversity decline[13,62]. The first manifestation of fragmentation is the proximity effect, namely the impact of urban expansion on adjacent habitats. We used the mean Euclidean distance from the patch edges of urban areas to the nearest patch edges of natural habitats in a 1 km cell to measure the changes in proximal effects caused by urban expansion (see detailed illustration in Supplementary Fig. 31). Specifically, we first analyzed the distance from the patch edges of urban areas to the nearest patch edges of natural habitats for the year 2015 and at ten years intervals throughout the period 2020–2100 under the five SSP scenarios of urban expansion. We then identified the inter-annual differences in mean distance using the distance at the latter time point to subtract the distance at the previous time point. A positive (negative) value means an increase (decrease) in the proximal effects due to urban expansion during the time in between.

In addition to the proximal effects, habitat fragmentation is often revealed in changes in landscape configuration such as a reduction in mean patch size (MPS), an increase in edge density (ED), and an increase in isolation of patches. Therefore, we selected three landscape metrics that reflect complementary aspects of habitat fragmentation: MPS, ED, and mean nearest neighbor distance between habitat patches (ENN_MN). The program FRAGSTATS[63] was used to calculate these three landscape metrics at the ecoregional scale. To examine the effects of urban land changes on habitat fragmentation, we established a 5 km buffer around urban land in 2100 across the SSP scenarios and measured the changes in landscape metrics of natural habitat within this buffer (see details in Supplementary Fig. 32).

**Land-systems data**. To examine the biodiversity loss due to future urban expansion, we first used an updated land-systems map[32] and projected urban expansion maps to characterize the conversions from non-urban land systems to urban land (peri-urban and villages were not considered). Compared with land cover data, land systems represent the interaction between major human activities and the eco-environment, which contain hierarchical categorical classifications that integrate more comprehensive metrics, including land cover, land-use intensity, and livestock density[64]. This updated land-systems map used the more recent land cover and land use datasets with finer spatial resolution (with 1 km resolution) than the original dataset (with ~9.5 km² resolution). However, the urban land data used in this updated map was obtained from the ESA CCI-Land Cover product for the period 2008–2012, which had different spatial extents than the urban land cover

map in 2015. Using this land-systems data directly may cause errors in the biodiversity loss assessment. Therefore, we replaced the urban land cover data used in the updated land-systems map with the urban land cover map for 2015. To address missing data or inconsistent data in some areas, we developed a new land-systems map following the decision-tree classification[64] and the same updated datasets of ref. [32].

**Terrestrial biodiversity data**. We used species richness and total abundance of three common terrestrial vertebrate taxa (amphibians, mammals, and birds) to represent terrestrial biodiversity. Species richness data, obtained from ref. [29] (https://biodiversitymapping.org/index.php/download/), was calculated through the spatial overlap of range maps for birds[65], mammals, and amphibians[66] (the three common vertebrate taxa) with an equal-area 10 km grid cell. This dataset also identified the spatial ranges of threatened amphibians, mammals, and birds and of small-ranged amphibians, mammals, and birds. The loss in threatened and small-ranged species is a key aspect of biodiversity loss. We considered the vulnerable, endangered, or critically endangered species in the IUCN Red List[66] as threatened species in our study. Here, small-ranged species refer to those living in a geographic range that is smaller than the median range size for that taxon.

**Estimating the terrestrial biodiversity loss due to urban expansion**. We investigated species responses to urban land conversion from other land-systems. This assessment was implemented based on the PREDICTS database[67] (Supplementary Note 9), which is a collaborative initiative that collects local-scale studies around the world and then uses meta-analysis to examine local terrestrial biodiversity responses to anthropogenic activities, such as habitat degradation, deforestation, pollution, hunting, invasive species, and overexploitation[67]. Using this database, a previous study[16] estimated variations in local species richness, rarefaction species richness, and total abundance percentage net change from a natural unimpacted baseline to different land use intensities (Supplementary Table 10). Therefore, this result can be used to estimate biodiversity loss per land-use intensity relative to a natural unaffected baseline.

To identify the spatial distribution of terrestrial biodiversity loss due to urban expansion under the five SSP scenarios, we extended the method of ref. [32] to match land-systems classes with different land-use intensities (i.e., high, medium, and low intensities; see ref. [16] and Supplementary Table 10 for a detailed description). Using these results, we can then calculate the mean biodiversity loss per land system (compared with the unimpacted baseline) by obtaining the average model estimates of biodiversity loss per land-use intensity class from earlier work[16]. We then estimated the mean rate of biodiversity change on a 1 km grid for each land system that will convert to urban land in the future. The biodiversity loss difference between the original land system and urban land was calculated accordingly to capture the relative biodiversity change. In fact, compared with the unimpacted baseline of the original PREDICTS estimates, we estimated the relative biodiversity change from each of the other land systems to urban land.

The original species richness data were based on 10 km grid, which could not be downscaled to 1 km grid because the calculation of species richness was based on 10 km grid. To further estimate the potential number of species lost in each 10 km grid cell, we first calculated the area-weighted mean value of percentage change in species across all land systems that will be occupied by future urban expansion between 2020 and 2100. We then multiplied this value by the number of species in each grid cell. Because spatially-explicit total abundance data at the global scale was not available, we did not calculate the absolute loss of abundance due to urban expansion.

**Reporting summary**. Further information on research design is available in the Nature Research Reporting Summary linked to this article.

## Data availability
The future urban expansion data can be obtained from https://doi.org/10.1594/PANGAEA.905890. The ESA CCI Land Cover product can be found at http://maps.elie.ucl.ac.be/CCI/viewer/download.php. The SSP database is available from https://tntcat.iiasa.ac.at/SspDb. The species richness data can be obtained from https://biodiversitymapping.org/index.php/download/. The updated land-systems map is publicly downloadable at https://box.hu-berlin.de/d/053f45f377/?p=%2FKehoe_et-al_2017_NatureEcoEvo&mode=list. Data on the boundaries of protected areas are retrieved from the World Database on Protected Areas (May 2020, https://www.protectedplanet.net/en). The biodiversity hotspots data can be retrieved from https://zenodo.org/record/3261807#.YToUlJ0zYuU. The Global 200 data is available from https://www.worldwildlife.org/publications/global-200. The Last of the Wild Areas data (version 2.0) is available from https://sedac.ciesin.columbia.edu/data/set/wildareas-v2-last-of-the-wild-geographic. The PREDICTS database is available from https://data.nhm.ac.uk/dataset/the-2016-release-of-the-predicts-database.

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

## Acknowledgements

We thank Shilong Piao and Junhui Wu for constructive comments on an earlier version of this paper. C.F. acknowledges support from the Science Fund for Creative Research Groups of the National Natural Science Foundation of China (42121001), G.L. acknowledges support from the National Natural Science Foundation of China (41971207), the Strategic Priority Research Program of Chinese Academy of Sciences, Pan-Third Pole Environment Study for a Green Silk Road (Pan-TPE) (XDA2004040), the program for "Kezhen-Bingwei" Excellent Talents in Institute of Geographic Sciences and Natural Resources Research and Young Innovator Association of Chinese Academy of Sciences (2020053).

## Author contributions

G.L., C.F., and X.L. conceived the study. G.L. and X.L. collected and analyzed the data. C.F. supervised the project. G.L. wrote the manuscript with support from Y. L., Z.W., S.S., S.H., W.Q., C.B., H.M., Y.F., and Y.F. All authors discussed the results and provided revisions on the manuscript.

## Competing interests

The authors declare no competing interests.
