## [Peer Review File · Nature Communications]

Reviewers' Comments:

Reviewer #1:

Remarks to the Author:

Overall, this is a well-written manuscript on an important topic. I have a few significant concerns, and then some minor text comments.

Significant concerns:

The protected area analysis seems to assume that urban growth can happen on top of currently gazetted protected areas, rather than occur around them. Certainly for IUCN categories I and II, this seems like an unlikely situation. I would rather this analysis focuses on the loss of habitat around protected areas, or their fragmentation from other protected areas (as indeed the authors do in a later analysis), rather than speculating that protected area boundaries will not be enforced. I would suggest dropping this analysis from the paper, unless the authors can defend this assumption.

The authors should clarify their urban shrinkage assumption. From some of their results, such as the relatively low urban area growth in China, I worry that the assumption is that if urban population decreases (which may be the case in China from 2050 to 2100) that urban area decreases. This does not happen in the real world, as shown in case studies from eastern Europe or the industrial belt of the US. As cities shrink in population the amount of impervious surface and disturbed area stays roughly constant, and the amount of natural habitat regenerated to ecologically functioning levels is rather small.

Emphasize throughout that you are talking about local within-site richness. Like Newbold, you could define an acronym and use this every time you talk about the percentage decline in local within-site species richness due to urbanization. This is important, because you do not want readers to confuse these results with a global decline in species richness.

I think the authors need to present more clearly the difference between a % decline in local within-site richness, and urban impacts on natural habitat that are globally significant. Areas in arid biomes, for instance, with a high loss in local within-site richness, many have few endemic species of global conservation concern. Specifically, their ecoregional results should be compared/contrasted with McDonald et al. 2018, which looked at habitat loss in ecoregions specifically focusing on vertebrate endemism, and which found a slightly different set of conservation priority ecoregions.

The estimated habitat loss (11-33 million ha by 2100) seems smaller than McDonald et al. 2019 Nature Sustainability (which had 290,000 km² between 2000-2030, or 29,000,000 million ha) given the longer timeframe. This deserves to be compared/contrasted with more fully in the discussion. Why is their result relatively smaller? There are perhaps good reasons (the Seto et al. urban growth forecasts used may have overestimated urban growth), but these deserve to be discussed.

Text edits/comments:

Abstract: Here and throughout the paper, rather than "~" you probably want to use "-" when indicating a range of numbers. Unless this journal's style guide requires the use of the tilde here. The tilde is usually reserved to mean "approximately" not to indicate a range.

Introduction:

In text citations do not need to be in blue, I don't think.

Line 47-49: I would rewrite this sentence to be clear that you are talking about the local (within site) richness. At a first read, it sounds like you mean urban expansion was responsible for 50% of the total global loss of species richness, and it was only when I saw the citation was it clear to me that you meant local richness.

Line 62: Suggest "spatially-explicit".

Results:

line 94: Wetlands have the largest loss only in proportional terms, right? In absolute terms (ha) the area of wetland loss is quite a bit less than the other habitat types. Be clear on this distinction, here and elsewhere in the text.

Figure 1. As currently designed, this figure is mostly showing the differences among SSPs. However, in the text more discussion is about the habitat types that will be lost the most. You might consider redesigning this figure to show differences in habitat types more clearly (this is hard to do now, since the Y-axes are different for each habitat type). For instance, you could have a bar chart showing habitat loss by 2100 for each habitat type, with error bars showing the range of values across all the SSPs.

lines 101-110: Be consistent about how you capitalize the regions. So, for instance, if you capitalize "Sub-Saharan Africa" you probably should capitalize "South Asia". However, when you have longer phrases like "as the northeastern, southern, and western coasts of the United States" it is quite alright not to capitalize them. Follow whatever style guide the journal prefers for this.

Figure 2- I find the lack of growth cross all SSP scenarios along the southern Chinese coast to be quite striking. Could you clarify how this is happening? How is it possible, for instance, that in SSP5 there is more urban growth in the Low Countries and Germany than in southern China, where there is expansion along the coast into natural habitat? Some of this is just scale, I would suggest you make the zoom in boxes approximately the same size (right now the US and China boxes are much bigger than the European or West African boxes).

lines 135-136: Are urban areas IN protected areas, or are they adjacent to protected areas? Please clarify the meaning here.

Lines 148-149: Please describe in a sentence how these hotspots were defined (they are one of several in common use in conservation biology, but just good to remind people which ones you are using and what they represent). Please contrast 0.9% with the fraction of the rest of the Earth's surface that is urbanized in 2015.

Lines 165-168: Is this ecologically meaningful, if it is less than 1% of the land in each ecoregion?

Lines 188-189: I do not understand what this sentence means, please clarify. I understand you measured change in the distance to the nearest edge, doesn't this by necessity involve changes to "the proximity of the habitat edges"?

Figure 3- Again, the layout of the figure makes visual comparison among the SSPs easiest, but that is not a focus of the discussion in the text. What ecologically do you want people to get from this figure?

Figure 4- I would suggest you make the zoom in boxes approximately the same size (right now the US and China boxes are much bigger than the European or West African boxes).

Figure 5- I would suggest that you do this analysis with ecoregions or some other more ecologically meaningful level of analysis, unless there is a policy reason to present national trends. The challenge is that countries vary widely in size, and considerable heterogeneity in urbanization impacts within the big countries like the US and China.

Discussion-

Overall, this section is long, without a clear narrative structure. It would be good to cut this down to a few paragraphs, each with a clear message they want readers to take away.

Lines 293-298: This paper only studied urban expansion, but obviously other land-use transitions have a larger biodiversity impact (e.g., logging, agriculture). Is it clear that these are also less under SSP1? If the authors did not study this, then they should avoid saying SSP1 is overall better for biodiversity.

Lines 308-310: Be careful to use "shrinkage" (an actual decline in the urban area or population) differently than just a reduction in the rate. For instance, "shrinking urban land expansion" might be better described as a "reduction in the rate of urban land expansion" in most places in the world.

Reviewer #2:

Remarks to the Author:

In their article, the authors analyse global scenarios of urbanization and the resulting impacts on biodiversity. To my knowledge, only few studies available address this topic on the global scale with such a high level of spatial detail. Therefore, I regard the underlying research as an important original contribution to the current scientific and societal debate on biodiversity conservation.

Overall, the paper is well written but needs substantial improvements and clarifications before it is ready for publication.

There are several issues regarding whole concept of the paper that need to be addressed.

(1)The authors did a good job in analysing the whole set of SSPs but they miss giving a proper description of main characteristics of the scenarios and the related policy implications regarding biodiversity impacts. Firstly, the authors should include a short description of the model used to generate urbanization pattern in the SI (even if it is already published). Secondly they need to explain the linkages between the underlying scenario assumptions and urbanization pattern, e.g. (a) What are the differences in settlement structures between the "sustainability" scenario and the "fossil fuel development" pathway"? (b) What are the assumptions regarding urban planning and nature conservation areas? (c) How do population densities develop? I think this information is crucial for a more detailed interpretation of the modelled implications on biodiversity and should be added to the manuscript (instead of just presenting ranges between very abstract scenario simulation results).

(2) Throughout the paper, settlement development and biodiversity loss is discussed on country-level or in terms of geographic hotspots. On the other hand, the land-use model uses input data aggregated on 32 world regions, i.e. each macro regions contains more than one country. Please explain the implications of this aggregation on uncertainties of the modelled locations of urban expansion and biodiversity loss.

(3)I like the idea to analyse biodiversity impacts by combining different indicators (habitat loss, fragmentation and species loss). Nevertheless, the authors need to explain in more detail why they have chosen this particular set of indicators and which different aspects of biodiversity impacts the aim to analyse – why is one indication not sufficient?

(4) The discussion should be reworked considerably. There are many statements that are vague and not supported by literature (see comments below).

In addition, I like to comment on some specific aspects of the paper:

Page 2: The authors should use consistent data regarding projections of future population. I think the SSPs that were analysed differ from the data cited in the first paragraph.

Page 4: Please explain why you have chosen this specific set of biodiversity indicators (see comment above). How are they interrelated?

Page 5, line 98: Are UK and Germany separate countries in your model framework or part of the "Western Europe" region? (see comment above).

Page 5, line 106ff: Urban area expansion is not only driven by population number but also by per capita area demand for housing, infrastructure etc. E.g. in Europe, we find regions with shrinking

population and growing urban areas. How is this effect captured in the scenarios?

Page 6ff: Avoid expression as "extremely large" or "even worse" when presenting results. Interpretation should be done in the discussion section.

Page 6ff: In this and the following sections you refer to CCI land-cover classes. But it seems that your model uses a much more aggregated classification (see SI Table 1). Please explain how you conducted this analysis?

Page 8: Please explain your assumptions regarding effectivity of protected areas within the different scenarios.

Page 10, line 190ff: Explain the differences between the SSPs.

Page 16, line 298: What are these opportunities? How are they related to the scenario storylines?

Page 16, line 301: In which sense and why does your study show significantly different results than the cited articles?

Page 18, line 326: There must be research on these issues. Please give references.

Page 18, line 332ff: Please add references that support your theses.

Page 19, line 360: Give evidence why it is understudied.

Page 25, line 500ff: I do not understand why you do an additional analysis on a coarser grid and how you combine the described data sets. Please explain in more detail.

Page 27: Here, more detailed information about PREDICTS and the matching process between land systems and intensity is required. A table with the coefficients used for the analysis should be added to the SI.

SI, figure 1: This type of plots is typically used to visualize uncertainty (e.g. a range of results between scenarios) rather than a range between different independent sets of simulation results – in this case 32 regions. Please use another graphic representation.

SI, figure 2: The colour scheme should be changed. I cannot recognize anything.

Reviewer #3:

Remarks to the Author:

The manuscript addresses an important topic. Especially I found the contribution to expected level of fragmentation -in addition to loss of habitat extents- of habitats due to future urban land expansion valuable. However, there are a few methodological and interpretation issues that need to be addressed. For example, the authors need to be measured in their claims. Their study is not an empirical one, in the sense that their findings are not verifiable -yet- by observation. Certain methodological details are vague (e.g., the nature of the urban land expansion forecasts) and important assumptions one needs to make in a study with such a long time horizon are not even discussed. Most importantly, I am not convinced that their findings lead to any novel insights beyond what is already reported in the literature by several studies. The manuscript is well-written overall though there are awkward phrases and typos throughout (e.g., (ln 66) "most of which", (ln 174) "slight"). I listed my detailed comments below:

1. (ln 30) State how much more instead of "much higher proportion".
2. (ln 44) Ref 3: There is a more recent study that reports similar findings but based on a larger number of works with more up-to-date and more detailed estimates on urban land expansion vs urban population growth. I suggest that the authors consider citing this more recent and more up-

to-date study: Güneralp et al. (2020)

3. (Ins 45) Consider writing "a major driver" instead of "the major driver" because there are other major drivers of land-use change globally including agricultural land expansion and deforestation. See Foley et al. (2005).
4. (Ins 53-54) "...more than 1.8–5.9 54 times urban land expansion from 2000 to 2100..." ◊
Confusing, please rephrase.
5. (Ins 71-73) To what extent does this assumption hold for cities across Eastern Europe where there has been widespread contraction across cities?
6. The authors make several statements about previous studies on forecasts of urban expansion or on their impacts on biodiversity; however, they do not cite the most relevant studies. For example, they do not support their claim "on only one aspect of impacts on biodiversity," on In 66. Which studies are these?
7. (In 98) How come countries with very little unaltered habitat left and with relatively slow urban expansion such as Germany and the UK are among those that are predicted to have the greatest amount of habitat loss to urban expansion?
8. (Ins 108-109) This does not make sense. So, China and India are expected to undergo population shrinkage from 2050 and 2100 but Europe's population will continue to increase such that it will be one of the regions with the most habitat loss to urban expansion? I know of no demographic projection scenario where Europe's population growth exceeds those of China and India by the end of the century.
9. (Ins 142-147) This information is not useful or can even be misleading. The urban forecasts did not account for protected status of the PAs or even their existence. Therefore, it does not mean much what proportion of urban forecasts occupy protected areas of which categories.
10. (Ins 145-146) What is the rationale to assume no category PAs as having intermediate status?
11. (In 179-In 480) The authors' use of the term "edge effects" is incorrect. Edge effect refers changes in community or population structure at the boundary of two or more habitats. What the authors refer to as edge effects appears to be proximal effects of urban areas on nearby habitats.
12. (Ins 213-215) Supporting citations?
13. (Ins 286-288) Empirical? What the authors did is taking urban expansion forecasts and several spatial biodiversity/habitat datasets to generate future projections on biodiversity/habitat impacts of urban expansion. Being a forward-looking study, there is nothing that is verifiable by observation in these analyses. The authors need to be measured in their claims.
14. (Ins 312-316) First, the sentence is too long. Second, it is crammed with catch phrases (e.g., urban development path, growth-oriented urban planning paradigm) with no insight (at least no novel insight) that we did not know from the previous studies. Third, there are no references cited to support any of its parts.
15. (Ins 326-331) Similar to the preceding comment, these comments amount to hand-waving repeating what is already well-known in conservation community, providing no new insights.
16. (Ins 354-355) How so? What evidence did the authors present apart from the focus of their study, biodiversity/habitat impacts?
17. (Ins 382-383) Please explain briefly how the two differ.
18. (Ins 386-389) Did the authors generate the urban expansion forecasts or used published forecasts? From what is stated under "Forecasting future urban expansion" in Methods, it is not clear if they used existing forecasts or created their own. Therefore, please clarify and rephrase.
19. (Ins 418-421) I am not convinced that the study has significant theoretical implications. As the authors state at the end of the paragraph, it suggests significant impacts on biodiversity/habitats from future urban expansion but does not advance any findings to inform theory, at least not any more than previous studies have already done.
20. (Ins 424-449) Here, the authors appear to put forward a few ideas although most of the paragraph repeats what is already known from previous studies. For example, one of these "update the IUCN Protected Areas Management Category" is interesting but needs further elaboration. Other two, "upgrade the management level of key biodiversity areas, and prevent development encroachment (urban expansion) on protected areas." have already been suggested. This also raises an important issue regarding the urban expansion forecasts. They were developed assuming there is no protection in the PAs from urban expansion. However, the enforcement varies widely around the world and none of these are addressed and discussed in the paper.
21. (Ins 452-463) So did the authors use the forecasts reported in Ref. 16 of generate their own forecasts?
22. (Ins 464-478) How about potential shifts in habitats and species ranges due to climate change?

In a study that has reports potential impacts of urban expansion by 2100, the lack of any discussion on these is not acceptable.

23. (Supplementary Fig.2 and Supplementary Fig.3) The captions are missing any information on what each map represents.

24. (Supplementary Fig.9) To which SSP forecasts do these maps belong?

References

Foley, J. A., R. DeFries, G. P. Asner, C. Barford, G. Bonan, S. R. Carpenter, F. S. Chapin, M. T. Coe, G. C. Daily, H. K. Gibbs, J. H. Helkowski, T. Holloway, E. A. Howard, C. J. Kucharik, C. Monfreda, J. A. Patz, I. C. Prentice, N. Ramankutty and P. K. Snyder (2005). "Global consequences of land use." *Science* 309(5734): 570-574.

Güneralp, B., M. Reba, B. U. Hales, E. A. Wentz and K. C. Seto (2020). "Trends in urban land expansion, density, and land transitions from 1970 to 2010: a global synthesis." *Environmental Research Letters* 15(4): 044015.

Reviewer #1

Overall, this is a well-written manuscript on an important topic. I have a few significant concerns, and then some minor text comments.

Response:

Thank you for your overall positive and constructive comments. We have seriously considered each of your comments and used them to further strengthen our manuscript (see our point-to-point responses below).

Significant concerns:

1. The protected area analysis seems to assume that urban growth can happen on top of currently gazetted protected areas, rather than occur around them. Certainly for IUCN categories I and II, this seems like an unlikely situation. I would rather this analysis focuses on the loss of habitat around protected areas, or their fragmentation from other protected areas (as indeed the authors do in a later analysis), rather than speculating that protected area boundaries will not be enforced. I would suggest dropping this analysis from the paper, unless the authors can defend this assumption.

Response:

Thank you for raising the question about the assumption that urban growth can even happen on top of currently gazetted protected areas. Intuitively, it seems unlikely for large-scale urban expansion to happen in protected areas with IUCN categories I and II. However, our analysis using historical data suggests that this is not the case. More specifically, we combined ESA CCI Land Cover data (from 1992 to 2015 with 300m resolution) with the World Database on Protected Areas (WDPA, May 2020 version, the number of protected areas with IUCN categories I and II was 18947) and found that urban growth indeed occurred on top of currently gazetted protected areas from 1992 to 2015: 579.31 km² urban areas occurred in 558 protected areas in 1992, and 1229.95 km² urban areas were distributed in 813 protected areas in 2015. This suggests that the gazetted protected areas were not strictly enforced, which may partly explain why the international community did not fully achieve any of the 20 Aichi biodiversity targets agreed in Japan in 2010 to slow the loss of the natural world. Thus, we expect this trend might continue in the future.

To address your concerns, we have now provided evidence from additional analyses to support our assumption. We have also made corresponding revisions in the paper:

- (a) We have specified this assumption in the Results section of the revised manuscript (p.8, line 136–141): “To reflect the potential encroachment of urban expansion on protected areas (Supplementary Note 5), the analyses presented here were based on the assumption that the encroachment of urban expansion on protected areas is not strictly restricted, and even occurs in the currently gazetted protected areas (IUCN categories I and II) (Supplementary Figs. 13 and 14).”
- (b) We have added an explanation for our assumption in the Method section (p.27, line 527–530): “Based on historical data analysis from 1992 to 2015 showing that urban expansions had occurred

in protected areas, we assumed no strict constraints for the encroachment of urban expansion on protected areas (Supplementary Figs. 13 and 14, Supplementary Table 4).”

- (c) We recognize that the enforcement of protected areas varies widely around the world, so we have also added the related discussions about this issue in the Discussion section (pp. 22–23, line 422–437): “Fourth, because urban expansion has occurred within protected areas in the past few decades¹⁹, we assume that this trend of urban encroachment within protected areas will continue without restrictions within protected areas in the future. However, our work is subject to one caveat. In particular, we may have overestimated the impacts of urban expansion on protected areas, especially on direct encroachment, because the protected areas around the world differ greatly in their enforcement effectiveness in preventing urban encroachment and human pressure^{33,44}. The enforcement effectiveness of protected areas depends on multiple factors, such as resources used to manage protected areas, law enforcement, and governance quality⁴⁵. However, it is still difficult to accurately identify future enforcement effectiveness of protected areas, because protected area downgrading, downsizing, and degazettement (PADDD) are becoming increasingly prevalent in some developed countries (e.g., the United States)³⁴. Even strict protected areas (i.e., IUCN categories I and II) are subject to increasing human pressure, which suggests that IUCN management category cannot inhibit the growth of human pressure⁴⁶.”

Supplementary Fig. 13 The areas of urban land in each protected area with IUCN categories I and II in 1992.

Supplementary Fig. 14 The areas of urban land in each protected area with IUCN categories I and II in 2015.

References:

1. Geldmann, J., Manica, A., Burgess, N. D., Coad, L. & Balmford, A. A global-level assessment of the effectiveness of protected areas at resisting anthropogenic pressures. *Proc Natl Acad Sci U S A* 116, 23209–23215 (2019).
2. Gray, C. L. et al. Local biodiversity is higher inside than outside terrestrial protected areas worldwide. *Nat Commun* 7, 12306 (2016).
3. Güneralp, B. & Seto, K. C. Futures of global urban expansion: uncertainties and implications for biodiversity conservation. *Environ Res Lett* 8, 014025 (2013).
4. Watson, J. E., Dudley, N., Segan, D. B. & Hockings, M. The performance and potential of protected areas. *Nature* 515, 67–73 (2014).
5. Golden Kroner, R. E. et al. The uncertain future of protected lands and waters. *Science* 364, 881–886 (2019).
6. Jones, K. R. et al. One-third of global protected land is under intense human pressure. *Science* 360, 788–791 (2018).

2. *The authors should clarify their urban shrinkage assumption. From some of their results, such as the relatively low urban area growth in China, I worry that the assumption is that if urban population decreases (which may be the case in China from 2050 to 2100) that urban area decreases. This does not happen in the real world, as shown in case studies from eastern Europe or the industrial belt of the US. As cities shrink in population the amount of impervious surface and disturbed area stays roughly constant, and the amount of natural habitat regenerated to ecologically functioning levels is rather small.*

Response:

Thank you for raising this good point. The assumption that population decline causes urban area to decrease is not necessarily true. We assume that a decrease in the population may still cause urban

area to remain the same or even increase”. Urban shrinkage is not just about a decline in population. Indeed, urban shrinkage is defined as “an urban area—a city, part of a city, an entire metropolitan area or a town—that has experienced population loss, economic downturn, employment decline, and social problems as symptoms of a structural crisis” (Martinez-Fernandez et al., 2012). Thus, urban shrinkage does not simply reflect the process of population shrinkage, but is a process involving economic, demographic, geographic, social, and physical dimensions, all of which can have effects on urban shrinkage (Martinez-Fernandez et al., 2012). Based on this definition, in the Method section we have now described our assumptions and forecasting future urban expansion in detail (pp. 25-26, line 486–513.)

As noted in the Method section, we base our study on the global projection of urban expansion dataset including five SSP scenarios with a 1-km resolution (Chen et al., 2020). To create the dataset of future urban expansion, Chen et al. (2020) built panel data regression using historical statistical data and urban land data to estimate future urban areas from urbanization rate (i.e., urban population/total population) and gross domestic product (GDP). The historical urban land data were obtained from the GHSL dataset, which covered the years 1975, 1990, 2000, and 2014. The concurrent statistical data of population, per capita GDP, and urbanization rate were collected from the World Bank and United Nations and aggregated at the macro regional level. The established regression equation was used to predict future per capita urban land demand using the scenario projections of per capita GDP and the urbanization rate provided by the SSPs database (<https://tntcat.iiasa.ac.at/SspDb>). Thus, the regional urban land demand in a future year t was obtained by multiplying the estimated urban land demand per capita at year t by the projected total regional population in the same year.

Thus, in our model framework, even if the population shrinks, but the urbanization rate and GDP per capita can still grow, then there is still new urban land demand. Only when population, urbanization, and economic growth all stagnate, the urban land demand will stagnate, and then urban land may not continue to expand. Thus, the assumption that population decline causes urban area to decrease is not the case.

To avoid confusion, we have also changed “shrinkage” to “decline in urban land demand” in the revised manuscript. Correspondingly, the relatively low urban area growth in China from 2050 to 2100 does not mean the reduction of existing urban land area. Instead, it means the reduction of urban land demand due to urban population decline. We have clarified our urban growth simulation assumption for this issue in the Methods section (p.26, line 507–513): “For regions with a decline in urban land demand, it was assumed that the land conversion from nonurban land to urban land is irreversible. In the spatial simulations, the substantial conversion of urban land to non-urban land will not occur. It is worth noting that in the urban growth simulation, if the estimated urban land area of a region in the future is smaller than its current urban area, then the existing urban lands will remain unchanged because future urban land demand can be met by the existing area¹⁷”

References:

1. Martinez-Fernandez C, Audirac I, Fol S, Cunningham-Sabot E. Shrinking cities: urban challenges of globalization. *Int J Urban Reg Res* 36(2):213–225 (2012).
2. Chen, G. et al. Global projections of future urban land expansion under shared socioeconomic

pathways. *Nature Communications* 11, 537 (2020).

3. *Emphasize throughout that you are talking about local within-site richness. Like Newbold, you could define an acronym and use this every time you talk about the percentage decline in local within-site species richness due to urbanization. This is important, because you do not want readers to confuse these results with a global decline in species richness.*

Response:

Thank you for your suggestion. Like Newbold et al. (2015), we have now defined an acronym when we first talk about local within-site species richness by saying “the local within-site species richness (hereafter ‘species richness’)” in this revision (see p.3, line 48).

4. *I think the authors need to present more clearly the difference between a % decline in local within-site richness, and urban impacts on natural habitat that are globally significant. Areas in arid biomes, for instance, with a high loss in local within-site richness, many have few endemic species of global conservation concern. Specifically, their ecoregional results should be compared/contrasted with McDonald et al. 2018, which looked at habitat loss in ecoregions specifically focusing on vertebrate endemism, and which found a slightly different set of conservation priority ecoregions.*

Response:

Thanks for raising this point. To address your concerns, we have performed additional analyses to compare our ecoregional results (i.e., the conservation priority ecoregions, see Supplementary Fig. 23) and those from McDonald et al. (2018). Supplementary Table 6 lists 30 priority ecoregions based on three indicators: projected mean urban growth area by 2100 (under SSP1-SSP5), projected mean urban growth rate by 2100 (SSP1-SSP5), and mean small-ranged species number of vertebrates. We define three thresholds according to the average values of the three indicators and the distribution of the data: (a) mean urban growth area by 2100 > 50km², (b) mean urban growth rate by 2100 > 20%, and (c) mean small-ranged species number of vertebrates > 35. We found that the 11 priority ecoregions highlighted in bold (Supplementary Table 6) were consistent with the results of McDonald et al. (2018). However, 19 priority ecoregions with high growth area, high urban growth rate, and high species number of vertebrates were not covered by McDonald et al. (2018). The main reason for this discrepancy is the difference in urban expansion data. The results of McDonald et al. (2018) were based on the urban growth forecasts by Seto et al. (2012), which tended to overestimate urban growth. For example, the Mount Cameroon and Bioko montane forests has only 10 km² of urban land area in 2015, and it will grow by 0.2 km² by 2100 based on our projection results. In contrast, Seto et al. (2012) forecasts that in this ecoregion, there will be over 66 km² of urban land area by 2030 (Supplementary Fig. 28). However, most of this ecoregion is not suitable for urban development. For instance, Dinerstein et al. (2017) found that 736 km², or 64%, of this ecoregion is in protected areas.

We have specified this analysis in the Results section of the revised manuscript (p.14, line 261–265):

“We have identified 30 conservation priority ecoregions with high risks of habitat loss and small-ranged species loss due to future urban expansion (Supplementary Table 6). These conservation priority ecoregions are all found in Latin America and Sub-Saharan Africa (Supplementary Fig. 23).”

In addition, we also compared our ecoregional results with McDonald et al. 2018 and explained the main reason for inconsistent result in the Discussion section (pp.20–21, line 379–391).

“The main reason may be that the projection of urban expansion used by McDonald et al.⁹ is up to 2030 derived from Seto et al.¹⁵ and this projection did not fully capture the dynamics of future urban land use due to coarse spatial resolution, and did not consider future population decline trend that may cause stagnation in urban land growth. For example, the simulation that our research was based on showed that most regions in the world will continue to experience urban expansion, but a few regions (e.g., China and other Asian countries) will have a relatively low urban expansion rate or even a decline in urban land demand after 2050 (see Supplementary Figs. 7–11 for China), which is different from Seto et al.¹⁵. Thus, they might overestimate urban growth (Supplementary Figs. 25–28). In addition, out of the 30 conservation priority ecoregions, we identified 19 ecoregions with high species number of vertebrates but high future urban growth potential, which were not covered by McDonald et al.²⁰.”

Supplementary Table 6. Conservation priority ecoregions for future urban expansion-caused habitat and species loss. These priority ecoregions were identified based on three indicators: projected mean urban growth area by 2100 (under SSP1–SSP5), projected mean urban growth rate by 2100 (SSP1–SSP5), and mean small-ranged species number of vertebrates. We define three thresholds according to the average values of the three indicators and the distribution of the data: (a) mean urban growth area by 2100 > 50km², (b) mean urban growth rate by 2100 > 20%, and (c) mean small-ranged species number of vertebrates > 35.

Ecoregions Name	Urban area in 2015 (km ²)	Projected mean urban growth area by 2100 (km ² , SSP1-SSP5)	Projected mean urban growth rate by 2100 (SSP1-SSP5)	Mean small-ranged species number of vertebrates
Albertine Rift montane forests	219	59.80	27	63
Cameroonian Highlands forests	133	142.80	107	42
Cauca Valley montane forests	145	149.20	103	127
Central American dry forests	635	213.80	34	40
Central American montane forests	142	175.00	123	76
Central American pine-oak forests	993	1101.00	111	61
Central Andean wet puna	431	134.20	31	61
Cordillera La Costa montane forests	252	113.60	45	71
Costa Rican seasonal moist forests	384	171.60	45	56
East African montane forests	51	145.20	285	43
Eastern Cordillera real montane forests	344	77.20	22	153
Hispaniolan moist forests	622	166.40	27	43
Isthmian-Atlantic moist forests	223	60.80	27	94
La Costa xeric shrublands	839	480.80	57	34
Madagascar subhumid forests	186	363.00	195	41
Magdalena Valley dry forests	61	51.20	84	62
Magdalena Valley montane forests	920	407.60	44	131

Northwestern Andean montane forests	519	146.20	28	170
Paraguana xeric scrub	223	144.00	65	45
Peruvian Yungas	221	49.80	23	123
Petén-Veracruz moist forests	397	190.80	48	52
Puerto Rican dry forests	138	131.80	96	38
Puerto Rican moist forests	978	853.00	87	45
Serra do Mar coastal forests	5226	1097.46	21	53
Sierra Madre de Chiapas moist forests	77	122.40	159	65
Southern Atlantic mangroves	746	222.60	30	38
Southern Pacific dry forests	437	129.60	30	49
Trans-Mexican Volcanic Belt pine-oak forests	1145	250.60	22	45
Tumbes-Piura dry forests	110	65.20	59	73
Venezuelan Andes montane forests	188	59.20	31	120

Supplementary Fig. 23 Conservation priority ecoregions for species loss caused by future urban growth.

Supplementary Fig. 28 The difference in projected urban growth by Seto et al. (2012) and ours (based on Chen et al., 2020) in the Mount Cameroon and Bioko montane forests. White lines indicate the boundary of the ecoregion.

Reference

Dinerstein, et al. An ecoregion-based approach to protecting half the terrestrial realm, *BioScience* 67(6): 534–545 (2017).

5. *The estimated habitat loss (11-33 million ha by 2100) seems smaller than McDonald et al. 2019 Nature Sustainability (which had 290,000 km² between 2000-2030, or 29,000,000 million ha) given the longer timeframe. This deserves to be compare/contrasted with more fully in the discussion. Why is their result relatively smaller? There are perhaps good reasons (the Seto et al. urban growth forecasts used may have overestimated urban growth), but these deserve to be discussed.*

Response:

Thanks for raising this point. Our estimated habitat loss (11–33 million ha by 2100) is smaller than McDonald et al. 2019 *Nature Sustainability* (which had 290,000 km² of habitat loss between 2000 and 2030, or 29 million ha). This may be partly because McDonald et al. (2019) used the urban growth forecasts by Seto et al. (2012), which tended to overestimate urban growth. We have now made several revisions in the manuscript:

Frist, we have now discussed about the possible reasons in the Discussion section (pp. 20–21, line 377–389):

“Compared to McDonald et al.⁹, our estimated habitat loss due to future urban expansion (11–33 million ha by 2100) is smaller than theirs (which had 29 million ha between 2000 and 2030). The main reason may be that the projection of urban expansion used by McDonald et al.⁹ is up to 2030 derived from Seto et al.¹⁵ and this projection did not fully capture the dynamics of future urban land use due to coarse spatial resolution, and did not consider future population decline trend that may cause stagnation in urban land growth. For example, the simulation that our research was based on showed that most regions in the world will continue to experience urban expansion, but a few regions (e.g., China and other Asian countries) will have a relatively low urban expansion rate or even a decline in urban land demand after 2050 (see Supplementary Figs. 7–11 for China), which is different from Seto et al.¹⁵. Thus, they might overestimate urban growth (Supplementary Figs. 25–28).”

Second, we have also illustrated some of the details in the Supplementary Figs 25–28. The overestimations of urban growth by Seto et al. (2012) in Asia, Africa, and South America are large. Among them, the urban growth in 63% of the ecoregions were overestimated. There are more than 4 times differences in the estimated urban growth for 14% of the ecoregions between Seto et al. (2012) and the projections our study was based on.

Supplementary Fig. 25 Global comparison between urban expansion forecasts of Seto et al. (2012) and our projection results under SSP2. (A) China, (B) India, (C) East Africa, and (D) West Africa.

Difference ratio 0.00 0.01 - 1.00 1.01 - 3.00 3.01 - 5.00 5.01 - 75.00

Forecasts of urban expansion to 2030 (Seto et al. 2012)/Our forecasts of urban expansion to 2030 under SSP2

Supplementary Fig. 26 Difference ratio of urban expansion forecasts between Seto et al. 2012 and our SSP2 projection results (based on Chen et al., 2020).

Supplementary Fig. 27. Differences in urban details for some metropolitan areas around the world for the year 2030 using 1-km resolution (our results based on Chen et al., 2020) and 5-km resolution (Seto et al., 2012).

Supplementary Fig. 28. The difference in projected urban growth by Seto et al. (2012) and ours (based on Chen et al., 2020) in the Mount Cameroon and Bioko montane forests. White lines indicate the boundary of the ecoregion.

Text edits/comments:

1. Abstract: Here and throughout the paper, rather than "~" you probably want to use "-" when indicating a range of numbers. Unless this journal's style guide requires the use of the tilde here. The tilde is usually reserved to mean "approximately" not to indicate a range.

Response:

Thank you for pointing this out. We have now changed "~" into "-" across our manuscript when we indicate a range in the numbers.

2. Introduction: In text citations do not need to be in blue, I don't think.

Response:

We have removed the blue colors for all in-text citations.

3. Line 47-49: I would rewrite this sentence to be clear that you are talking about the local (within site) richness. At a first read, it sounds like you mean urban expansion was responsible for 50% of the total global loss of species richness, and it was only when I saw the citation was it clear to me that you meant local richness.

Response:

We have revised this sentence into “Global assessments show that urban expansion has caused about 50% loss of local within-site species richness (hereafter ‘species richness’) and 38% loss of local abundance in intensely used urbanized areas as compared to a naturally unimpacted baseline¹⁵” (p.3, line 47–50).

4. Line 62: Suggest "spatially-explicit".

Response:

Following your suggestion, we have now changed “spatially explicit” into “spatially-explicit” (p.3, line 62).

5. Results: line 94: Wetlands have the largest loss only in proportional terms, right? In absolute terms (ha) the area of wetland loss is quite a bit less than the other habitat types. Be clear on this distinction, here and elsewhere in the text.

Response:

Sorry for causing this confusion. To clarify, we want to show that SSP4 scenario will face the greatest loss of wetlands compared to the other scenarios. We have changed this sentence to “In terms of urban encroachment on wetlands, wetland will undergo the largest loss in area under scenario SSP4 than under other scenarios.” (p.5, line 94–96).

6. Figure 1. As currently designed, this figure is mostly showing the differences among SSPs. However, in the text more discussion is about the habitat types that will be lost the most. You might consider redesigning this figure to show differences in habitat types more clearly (this is hard to do now, since the Y-axes are different for each habitat type). For instance, you could have a bar chart showing habitat loss by 2100 for each habitat type, with error bars showing the range of values across all the SSPs.

Response:

Thank you for your suggestion. In addition to the original figures showing the differences among SSPs, we have also added a new figure displaying the habitat loss by 2100 for each habitat type (see Fig. 1a and 1b; p. 6).

Fig. 1 Future direct habitat loss due to urban expansion under SSP scenarios. (a) The habitat loss by 2100 for each habitat type. Error bars represent 95% confidence intervals. **(b)** The losses in total area, forest, shrubland, grassland, wetland, and other land.

7. lines 101-110: Be consistent about how you capitalize the regions. So, for instance, if you capitalize "Sub-Saharan Africa" you probably should capitalize "South Asia". However, when you have longer phrases like "as the northeastern, southern, and western coasts of the United States" it is quite alright not to capitalize them. Follow whatever style guide the journal prefers for this.

Response:

Thank you for your suggestion. We have now capitalized "south Asia" as "South Asia" (p. 5, line 110).

8. Figure 2- I find the lack of growth cross all SSP scenarios along the southern Chinese coast to be quite striking. Could you clarify how this is happening? How is it possible, for instance, that in SSP5

there is more urban growth in the Low Countries and Germany than in southern China, where there is expansion along the coast into natural habitat? Some of this is just scale, I would suggest you make the zoom in boxes approximately the same size (right now the US and China boxes are much bigger than the European or West African boxes).

Response:

To clarify, China is not the hotspot of global urban land expansion in the future, especially after the population peak in 2030. We found that this was mainly due to population shrinkage that will occur in China based on the estimation of SSP Database (Shared Socioeconomic Pathways, Version 2.0) (see Supplementary Fig. 7). In addition, this population shrinkage of China in the future was also confirmed by the World Population Prospects 2019 (United Nations) (Supplementary Fig. 8). It is worth noting that although China’s urbanization rate and per capita GDP will continue to increase in the future (Supplementary Figs. 9–10), the urban population will also shrink obviously after around 2050 due to the decline of the total population (Supplementary Fig. 11). The shrinkage of the urban population will lead to a reduction in the demand for urban land.

In SSP5, we do find more urban growth in the Low Countries and Germany than in southern China (see Fig. 2), but there is no expansion along the coast into nature habitat. One explanation for this pattern of results is the changes in future urban population. That is, compared with southern Chinese coast, the Low Countries and Germany will experience urban population growth under the SSP5 scenario, and also have a certain amount of suitable land for urban development.

Following your suggestion, we have revised Fig. 2 to make the zoom in boxes approximately the same size (see Fig. 2).

Supplementary Fig. 7 The changes in total population (in million) in China from 2010 to 2100 based on the SSP Database version 2.0.

Supplementary Fig. 8 The changes in total population (in billions) in China from 1950 to 2100 based on the World Population Prospects 2019, UN.

Supplementary Fig. 9 Future urbanization estimation in China from 2010 to 2100 based on the SSP Database version 2.0.

Supplementary Fig. 10. Future per capita GDP estimation in China from 2010 to 2100 based on the SSP Database version 2.0.

Supplementary Fig. 11 Future urban population estimation in China from 2010 to 2100 based on the SSP Database version 2.0.

Fig. 2 Future hot spots and cold spots of habitat loss due to urban expansion under SSP scenarios by 2100. Figures for the United States (a), Europe (b), Africa (c), and China (d) are presented separately.

9. lines 135-136: Are urban areas IN protected areas, or are they adjacent to protected areas? Please clarify the meaning here.

Response:

Our spatial overlay analysis shows that urban areas with 30,594 km² were distributed in 28,152 protected areas, accounting for 12.6% of all protected areas in 2015. This suggests that there are fairly high proportions of urban areas located in protected areas. There are three possible reasons for this phenomenon. First, despite the various types of protected areas around the world, the primary purpose of many protected areas is not to protect the natural ecosystem. Many protected areas receive protection because of their recognized cultural values, such as the World Heritage Site (mixed). Second, many protected areas are established in complicated and large areas, and cannot avoid the presence of urban areas and other human activities, especially in areas with high population density and land use intensity. Third, the global urban areas data obtained by remote sensing often fail to distinguish between urban areas, towns, and villages, leading to the identification of some land of villages and towns as urban areas. We find that these overlaying areas were mainly located in Europe and Japan (Supplementary Fig. 15). Supplementary Fig. 16 shows a typical example for Kamianets-Podilskyi, a city within Podolskie Tovtry National Park in Ukraine. In addition, our results were consistent with the global study by Güneralp and Seto (2013) and Güneralp et al.'s (2017) study of Africa. Globally, Güneralp and Seto (2013) found that 32,000 km² of protected areas were already urbanized circa 2000, corresponding to 5% of global urban land (Güneralp and Seto, 2013). Güneralp et al. (2017) found that about 500 km² of urban land in Africa were located within its protected areas in 2000.

Supplementary Fig. 15 Urban areas were distributed in protected areas for the year 2015.

Supplementary Fig. 16 Kamianets-Podilskyi, a city within Podolskie Tovtry National Park, Ukraine.

References:

1. Güneralp, B. & Seto, K. C. Futures of global urban expansion: uncertainties and implications for biodiversity conservation. *Environ Res Lett* 8, 014025 (2013).
2. Güneralp, B., Lwasa, S., Masundire, H., Parnell, S. & Seto, K. C. Urbanization in Africa: challenges and opportunities for conservation. *Environ Res Lett* 13, 015002 (2017).

10. Lines 148-149: Please describe in a sentence how these hotspots were defined (they are one of several in common use in conservation biology, but just good to remind people which ones you are using and what they represent). Please contrast 0.9% with the fraction of the rest of the Earth's surface that is urbanized in 2015.

Response:

We have made several revisions to address your comments:

First, we have added a detailed description of the definition of biodiversity hotspot in the Supplementary Information (p.9):

“Supplementary Note 6. Biodiversity Hotspots Data

Biodiversity hotspots are places on Earth that are rich in biological resources but deeply threatened. To be a biodiversity hotspot, an area must (a) have at least 1,500 endemic vascular plant species that are found nowhere else on the planet, and (b) have 30% or less of its original natural vegetation (i.e., threatened)¹⁵. Based on these two criteria, 36 ecoregions have been identified as biodiversity hotspots¹⁶, and the success of species conservation in these ecoregions has greatly impact in protecting global biodiversity.”

Second, we have added detailed descriptions of other biodiversity prioritization schemes, Protected Area Data, Global 200 Data, and Last of the Wild Areas Data in the Supplementary Information (see pp. 8, 10–11).

Third, we have added information about the fraction of the rest of the Earth’s surface that is urbanized in 2015 (p. 8, line 151–152):

“And this proportion (0.90%) is higher than that located in the rest of the Earth’s surface

(0.51%) in 2015.”

References:

1. Myers, N., Mittermeier, R. A., Mittermeier, C. G., da Fonseca, G. A. B. & Kent, J. Biodiversity hotspots for conservation priorities. *Nature* 403, 853–858 (2000).
2. Hoffman, M., Koenig, K., Bunting, G., Costanza, J. & Williams, K. J. in *Biodiversity Hotspots* (version 2016.1). Zenodo. <https://doi.org/10.5281/zenodo.3261807> (2016).

11. Lines 165-168: Is this ecologically meaningful, if it is less than 1% of the land in each ecoregion?

Response:

We found that the global urbanized areas only account for about 0.51% of the Earth’s surface in 2015. Although the proportion of urban land in each ecoregion (0.61–0.79%) is relatively low in the Global 200, it is still higher than the global proportion of urbanized areas (0.51%) in 2015. Meanwhile, these regions have irreplaceability and distinctiveness for global biodiversity protection. The distribution of future urban areas in these ecoregions is not uniform. By 2100, the urban areas of four key ecoregions will increase by 877–9955%, 527–646%, 18–902%, and 500–1037%, respectively. For these ecoregions, the future ecological risks are extremely high. In addition, the effects of urban areas on biodiversity were not just limited within the local areas. Therefore, although it is less than 1% of the land, we firmly believe that it still has substantial ecological impact.

12. Lines 188-189: I do not understand what this sentence means, please clarify. I understand you measured change in the distance to the nearest edge, doesn't this by necessity involve changes to "the proximity of the habitat edges"?

Response:

To clarify, we have analyzed the impact of future urban expansion on the natural habitat edges. However, our previous results show that not all urban expansion leads to a shorter distance between natural habitat and urban areas: < 1% of natural habitat will be far from the natural habitat edge due to future urban expansion. We have rechecked the results of our analyses and find that these areas are mainly located in Greenland. Thus, it may be unreasonable to discuss the impact of urban expansion on natural habitats in these areas. To avoid confusion, we have revised the original sentence “More importantly, 99% of these changes involve the proximity of the habitat edges, which will inevitably increase the risk of biodiversity decline” into “The future urban expansion will make 34–40 Mha natural habitat much closer to urban areas, which will inevitably threaten the natural habitat and increase the risk of biodiversity decline.” (p.10, line 185–188).

13. Figure 3- Again, the layout of the figure makes visual comparison among the SSPs easiest, but that is not a focus of the discussion in the text. What ecologically do you want people to get from this figure?

Response:

To clarify, we use Fig. 3 to show the future trends in the impact of urban expansion on habitat fragmentation, as well as the impact differences across different SSP scenarios. To avoid confusion, we have made some revisions to more clearly interpret the patterns of results in Fig. 3 (p.12, line 211–227):

“The median of MPS is the largest under scenario SSP1, followed by SSP4, SPP2, and SSP3

with some fluctuations in between, and the smallest MPS is found with the most fragmented landscape under scenario SSP5. A smaller patch size indicates that the inner parts of the habitat are subject to higher risk of being influenced by external disturbance. Future urban expansion also tends to cause an increase in the edge density of natural habitat, which is often linked with smaller patches or more irregular shapes and poses a threat to biodiversity that influences many ecological processes (e.g., the spread of dispersal and predation)^{13,27,28}. Scenario SSP1 shows the best performance in maintaining a low habitat edge density and a high level of biodiversity conservation. However, under scenario SSP5, edge density will experience a rapid increase in the second half of the 21st century. Meanwhile, the ENN_MN will increase substantially in the future, suggesting that areas with the same habitat type will become increasingly isolated, irregular, dispersed, or unevenly distributed due to the barrier of urban land. This will affect the speed of dispersal and patch recolonization. Scenario SSP1 is also most conducive to maintaining the proximity of areas of natural habitats with the same habitat type. Other scenarios show similar performance.”

14. Figure 4- I would suggest you make the zoom in boxes approximately the same size (right now the US and China boxes are much bigger than the European or West African boxes).

Response:

Thanks for your suggestion. We have revised Fig. 4 to make the zoom in boxes approximately the same size (see Fig. 4).

Fig. 4 Potential biodiversity loss due to future urban expansion under SSP scenarios. The biodiversity loss in terms of the number of terrestrial vertebrate species (amphibians, mammals, and birds) lost per 10-km grid cell in the North America (a), Europe (b), the Gulf of Guinea coast (c), and East Asia (d).

15. Figure 5- I would suggest that you do this analysis with ecoregions or some other more ecologically meaningful level of analysis, unless there is a policy reason to present national trends. The challenge is that countries vary widely in size, and considerable heterogeneity in urbanization impacts within the big countries like the US and China.

Response:

Thanks for your suggestion. We have revised Fig. 5 in the revised manuscript to present the trends with ecoregions (see Fig. 5). Accordingly, the original Fig. 5 was moved to Supplementary Information (see Supplementary Fig. 24).

Fig. 5 Average potential biodiversity loss per 10-km grid cell in ecoregions due to future urban expansion under SSP scenarios. The mean potential biodiversity loss in terms of average number of the estimate of terrestrial vertebrate species (amphibians, mammals, and birds) lost per 10-km grid cell. SR= Species Richness. Gray areas were not considered in this analysis.

16. Discussion-Overall, this section is long, without a clear narrative structure. It would be good to cut this down to a few paragraphs, each with a clear message they want readers to take away.

Response:

Thanks for your suggestion. We have now re-structured the Discussion section, cut it down to fewer paragraphs, and substantially revised it to be more concise with a clear message in each paragraph (pp.16–25). We hope you agree that the Discussion is substantially improved.

17. Lines 293-298: This paper only studied urban expansion, but obviously other land-use transitions have a larger biodiversity impact (e.g., logging, agriculture). Is it clear that these are also less under SSP1? If the authors did not study this, then they should avoid saying SSP1 is overall better for biodiversity.

Response:

Thanks for pointing this out. We reviewed the relevant literature on this issue and find that the scenario SSP1 is overall better for biodiversity. In SSP1, with the lowest demands for agricultural goods and high intensification of agricultural production, agricultural land decreases and is

significantly lower than in the other SSPs. (Popp et al., 2017). One general conclusion is that low demand for agricultural commodities, rapid growth in agricultural productivity and globalized trade, and largest forest area are most pronounced in SSP1 scenario, and have the potential to maintain natural ecosystems and conserve biodiversity. So, in general, we expect that the SSP1 has the most positive effects for sustainable development, including sustainable food consumption, rapid growth in agricultural productivity, and forest conservation. According to the estimation of Popp et al. (2017), cropland area is the lowest in SSP1 and highest in SSP3, leading to higher forest area in SSP1 and lower forest area in SSP3, and cumulative CO₂ emissions from land use change are lowest in SSP1 and highest in SSP3. We have added the explanation in the Discussion section (p.17, line 299–304):

“In particular, if the sustainable pathway (i.e., scenario SSP1) of urban expansion is properly implemented, humans will be able to maintain a relatively low natural habitat loss, low habitat fragmentation, and a high level of species conservation (SSP1 with lowest cropland expansion and higher forest area is also generally more beneficial for biodiversity conservation according to several existing estimations^{30,31}).”

References:

1. Popp A, Calvin K, Fujimori S, et al. Land-use futures in the shared socio-economic pathways. *Global Environmental Change*, 42:331-345 (2017).
2. Schipper, A. M. et al. Projecting terrestrial biodiversity intactness with GLOBIO 4. *Glob Change Biol* 26, 760-771 (2020).

18. Lines 308-310: Be careful to use "shrinkage" (an actual decline in the urban area or population) differently than just a reduction in the rate. For instance, "shrinking urban land expansion" might be better described as a "reduction in the rate of urban land expansion" in most places in the world.

Response:

Thanks for pointing this out. Indeed, an actual decline in urban area is rarely seen in reality. To avoid confusion, we have clearly described “shrinkage” as “the decline in urban land demand”.

Reviewer #2

In their article, the authors analyse global scenarios of urbanization and the resulting impacts on biodiversity. To my knowledge, only few studies available address this topic on the global scale with such a high level of spatial detail. Therefore, I regard the underlying research as an important original contribution to the current scientific and societal debate on biodiversity conservation.

Overall, the paper is well written but needs substantial improvements and clarifications before it is ready for publication.

Response:

Thank you for your positive and constructive comments. We have substantially revised our manuscript based on your comments (see our point-to-point responses below). We hope you agree that this is a much-improved manuscript.

There are several issues regarding whole concept of the paper that need to be addressed.

1. The authors did a good job in analysing the whole set of SSPs but they miss giving a proper description of main characteristics of the scenarios and the related policy implications regarding biodiversity impacts. Firstly, the authors should include a short description of the model used to generate urbanization pattern in the SI (even if it is already published). Secondly they need to explain the linkages between the underlying scenario assumptions and urbanization pattern, e.g. (a) What are the differences in settlement structures between the “sustainability” scenario and the “fossil fuel development” pathway”? (b) What are the assumptions regarding urban planning and nature conservation areas? (c) How do population densities develop? I think this information is crucial for a more detailed interpretation of the modelled implications on biodiversity and should be added to the manuscript (instead of just presenting ranges between very abstract scenario simulation results).

Response:

We have made corresponding revisions to address your concerns.

First, we have added a short description of the model used to generate urbanization pattern in the Supplementary Information (pp. 3–5; see also below).

Full SSP description of urbanization pattern

The five shared socioeconomic pathways (SSPs) characterize a wide range of possible future development pathways with different trends in various domains (e.g., rate of urbanization).

Below, we describe the different pathways in detail:

SSP1 (Sustainability): SSP1 envisions a development path of rapid urbanization for all country groups (i.e., high-, middle-, and low-income countries), coupled with high-income growth. Urbanization is partly driven by a desire to promote environment-friendly living conditions, and compact urban form contributes to resource efficiency¹⁻³. Rural-to-urban migration is moderate. Urbanization is well managed to minimize urban sprawl and urban de-concentration⁴. Cities become stable incubators and enablers of sustainable practices⁵. Global urbanization rate is high and is expected to reach 92.6% by 2100.

SSP2 (Middle of the road): SSP2 envisions a development path of moderate urbanization and moderate income growth for all country groups¹. Urbanization growth trends vary by region and over time, but on average they are closer to the center of expectations for future outcomes than to the upper or lower bounds of possibilities⁴. Urbanization has been particularly transformative in East and South Asia and Sub-Saharan Africa. As a result of sustainable energy technologies and related designs, the transformation of cities has proceeded at different rates, with the highest rates in developed or rapidly developing cities⁵. Global urbanization rate is moderate and is expected to reach 79.7% by 2100.

SSP3 (Regional rivalry): SSP3 envisions a development path with slow urbanization for all country groups. Slow economic growth limits employment opportunities and cross-regional mobility, thus constraining the process of urbanization. Moreover, poor urban planning reduces the attractiveness of urban areas as destinations^{1,6}. Urbanization faces greater challenges in developing countries, where inequality and fragmentation lead to a mixed pattern of urban

change (e.g., pockets of wealthier, dispersed settlements along with more concentrated slum-type growth)⁴. Disadvantaged populations, however, continue to migrate to poorly planned settlements around large urban areas, particularly in low-income countries⁵. Global urbanization rate is very low and is expected to reach 58.4% by 2100.

SSP4 (Inequality): SSP4 envisions a development path with medium urbanization in high-income countries, and fast urbanization in medium- and low-income countries. Moderate economic growth and attractive urban conditions in cities with high concentrations of the elite have supported urbanization in high-income countries, but rapid aging due to low fertility has moderated rural-urban migration⁷. In contrast, high fertility in medium- and low-income countries produces age structures conducive to migration. In medium-income countries, the assumption of medium economic growth is associated with the development of cities as manufacturing centers and engines of economic growth; therefore, urbanization proceeds rapidly⁸. In low-income countries, rapid population growth and the pressure of shrinking land and other resources have stimulated rural out-migration⁹. Meanwhile, large income disparities are assumed to apply particularly between rural and urban areas, and thus cause large flows of migration to urban areas¹⁰. Cities are affected by high inequality, providing urban amenities for the elite but poor housing and infrastructure for the rest of the population, leading to massive slum expansion and high unemployment rates^{1,11}. Spatial development patterns vary across cities, with some cities dominated by urban sprawl, whereas better planning in cities that are predominantly inhabited by the higher income classes leads to more concentrated development⁴. Urbanization rate is high and is expected to reach 91.7% by 2100.

SSP5 (Fossil-fueled development): SSP5 envisions a development path of rapid urbanization for all country groups. Urban areas have become attractive destinations due to rapid economic growth and technological change, which has enabled large engineering projects to develop desirable housing. Even if population growth rates decline, increases in agricultural productivity and wealth growth will lead to increased migration to cities and growth in the urban labor force^{1,6}. Unlike SSP1, however, urban planning and land use management has difficulty to keep up with the rapid pace of urbanization in the first decades of this century, and sprawling patterns of development dominate⁴. Over time, the pace of urbanization has converged, and urban structures and forms have evolved in different parts of the world to reflect historical patterns and prevailing local and national policies. This includes densely populated megacities in densely populated countries, as well as metropolitan areas with significant urban sprawl in other parts of the world⁵. Urbanization rate is high and is expected to reach 93.0% by 2100.

Supplementary Table 8 Summary of assumptions on urbanization patterns for five SSPs.

Elements	SSP1	SSP2	SPP3	SSP4	SSP5
Urbanization in high-income countries	Fast	Central	Slow	Central	Fast
Urbanization in medium-income countries	Fast	Central	Slow	Fast	Fast
Urbanization in low-	Fast	Central	Slow	Fast	Fast

income countries					
Urbanization rate by 2100	92.6%	79.7%	58.4%	91.7%	93.0%
Spatial pattern	Concentrated	Historical patterns	Mixed	Mixed	Sprawl
Migration	Moderate	Intermediate	Low	Mixed	Fast

Second, to answer your question about the linkages between the underlying scenario assumptions and urbanization pattern, there are great differences in settlement structures between the “sustainability” scenario (SSP1) and the “fossil fuel development” pathway (SSP5). Under SSP1 (sustainability), compact urban form contributes to high resource efficiency. Meanwhile, urban sprawl and urban de-concentration are minimized. In contrast, under SSP5 (fossil fuel development), urban sprawl pattern dominates the development process. Urban structures and forms have developed in different regions of the world to reflect historical patterns and prevailing local and national policies. This includes densely populated megacities in densely populated countries, as well as metropolitan areas with considerable urban sprawl in other parts of the world. These differences in urbanization patterns have been articulated in the full SSP description. We have added this information in the Supplementary Information (pp. 3–5).

Third, we have included a table that summarizes the assumptions regarding urban planning and nature conservation areas of the SSPs in the Supplementary Information (see Table 9; pp. 59–60). However, current spatial simulation models of urban expansion do not fully reflect these assumptions. These assumptions can only be partially reflected in the forecasts of urban land demand at the regional scale.

Supplementary Table 9 Assumptions for urban planning and protected areas under SSPs.

Scenarios	Urban planning assumptions	Protected areas assumptions (Environment assumptions)
SSP1	Urbanization is well managed, and urban planning is promoted in tandem with high urbanization rates.	It is an environment-friendly development pattern to strengthen the protection of fragile ecosystems and regions such as protected areas. Land use is strictly regulated. Urban expansion has barely encroached on protected areas. Protected areas are effective. Protected areas are on track to meet Aichi’s 17% target due to strong land-use change regulation.
SSP2	Moderate urban planning regulation.	Growing energy demand has led to the continuous environmental deterioration. Moderate regulation of land use leads to a slow decline in deforestation rates. Moderate land use regulation makes the effectiveness of protected areas in the middle level. Protected areas are being moderately encroached upon. Protected areas are expected to meet the Aichi target of 17% of land area due to moderate land use change regulation gradually implemented from 2010–2050.
SSP3	Urban settlements are poorly planned, particularly in developing countries where	Not enough attention has been paid to solving environmental problems, resulting in serious environmental degradation in some areas.

	inequality and fragmentation cause mixed pattern of urban change.	Deforestation continues because of a lack of regulation, competition for land and the rapid expansion of agriculture. Poor land use regulation leads to low effectiveness of protected areas. Urban expansion has encroached heavily on protected areas. Protected areas are under serious threat.
SSP4	Spatial development pattern varies across cities, with urban sprawl dominating in some cities, whereas better planning in cities that are predominantly inhabited by the higher-income groups leads to more concentrated development.	There are significant differences in environmental conditions. On the one hand, there are some areas of world concern, close to the places where middle- and high-income groups live and vacation, which are well managed. On the other hand, resource and production areas and many other out of sight places are neglected and become deteriorated. Conservation of protected areas is also divided, with highly regulated and well-managed areas in middle- and high-income countries, but largely unmanaged and deteriorating areas in low-income countries.
SSP5	It is difficult for urban planning to keep up with high urbanization rates, and the sprawling pattern of development is dominant.	Regulations are imperfect, and many protected areas are not effectively protected. Protected areas are under serious threat.

Finally, to answer your last question about population density, it is clear that with the changes in the total population and built-up areas across SSP scenarios, the population density will also change. SSP data show that future population density will decrease in all scenarios except for SSP3. Among these scenarios, population density will be the lowest under SSP5 scenario, followed by the SSP1 scenario. The change paths of SSP3 and SSP4 scenarios are very similar. Only under the SSP3 scenario, population density will increase after 2050 (see Fig. 1 below).

Fig. 1 Future population density change trends under SSP scenarios.

References:

1. Jiang, L. & O'Neill, B. C. Global urbanization projections for the Shared Socioeconomic Pathways. *Global Environmental Change* 42, 193–199 (2017).
2. Gossop, C. Low carbon cities: An introduction to the special issue. *Cities* 28, 495–497 (2011).
3. Ewing, R. & Cervero, R. Travel and the Built Environment. *J Am Plann Assoc* 76, 265–294 (2010).
4. Jones, B. & O'Neill, B. C. Spatially explicit global population scenarios consistent with the Shared Socioeconomic Pathways. *Environ Res Lett* 11, 084003 (2016).
5. O'Neill, B. C. et al. The roads ahead: Narratives for shared socioeconomic pathways describing world futures in the 21st century. *Global Environmental Change* 42, 169–180 (2017).
6. Mohan, R. The effect of population growth, the pattern of demand and of technology on the process of urbanization. *J Urban Econ* 15, 125–156 (1984).
7. Kelley, A. C. & Williamson, J. G. Population Growth, Industrial Revolutions, and the Urban Transition. *Popul Dev Rev* 10, 419–441 (1984).
8. Ledent, J. Rural-Urban Migration, Urbanization, and Economic Development. *30*, 507–538 (1982).
9. Abdel-Rahman, A. N., Safarzadeh, M. R. & Bottomley, M. B. Economic growth and urbanization: A cross-section and time-series analysis of thirty-five developing countries. *International Review of Economics and Business* 53, 334–348 (2006).
10. Jiang, L. Internal consistency of demographic assumptions in the shared socioeconomic pathways. *Popul Environ* 35, 261–285 (2014).
11. Fay, M. & Opal, C. Urbanization without growth: A not so uncommon phenomenon. Vol. 2412 (World Bank Publications, 2000).

2. Throughout the paper, settlement development and biodiversity loss is discussed on country-level or in terms of geographic hotspots. On the other hand, the land-use model uses input data aggregated on 32 world regions, i.e. each macro regions contains more than one country. Please explain the implications of this aggregation on uncertainties of the modelled locations of urban expansion and biodiversity loss.

Response:

Thank you for your suggestion. To clarify, we based our study on the global projection of urban expansion dataset in previous research (Chen et al., 2020), which used input data aggregated on 32 world regions. Many of these macro regions (e.g., EU15, LAM-H, and MEA-M) contain more than one country, which may lead to uncertainties in the future forecast of national-scale urban land demand. In building the global projection of urban expansion dataset, Chen et al. (2020) used the FLUS model to simulate future urban land expansion at a 1-km resolution. This model assumes that the probability of a non-urban grid being converted into an urban grid is a product of the urban development potential, neighborhood effect, development restriction, and adjustment factor. To deal with uncertainties in the modelled locations, Chen et al. (2020) quantified the parameter uncertainties in the projections of urban land demand. In order to deal with the spatial heterogeneity, Chen et al. (2020) assigned different intercepts to different regions in the panel data regression, implementing spatial simulations in different regions separately. They then performed 100 spatial simulations to understand the stochastic uncertainty of the model.

We agree with the reviewer that uncertainty in future urban land demand at the national scale may lead to uncertainties in the locations of future urban land expansion. In addition, the uncertainties in future urban land demand and location may lead to uncertainties in the impacts of future urban land expansion on biodiversity at the country and ecoregional level. However, these uncertainties can be mainly attributed to the selection of data scale. The conditions of geographic location and income level (i.e., high-income, mid-income and low-income levels) of different countries were taken into account in the definition of SSP database (<https://tntcat.iiasa.ac.at/SspDb>) for 32 macro regions.

Urban development policies usually operate at the national level, but in this study, we mainly provide urban land demand at the regional scale due to the lack of scenario data at the national level. We have now acknowledged this limitation in the Discussion section and proposed that future research can use more data to obtain relevant country-level information for different scenarios (p.22, line 414–422):

“Third, we based our study on the global projections of urban expansion in previous research that used input data aggregated on 32 world regions¹⁷. Many of these macro regions (e.g., EU15, LAM-H, and MEA-M) contain more than one country, which may lead to uncertainties in future forecast of national-scale urban land demand and the specific locations of future urban land expansion. These uncertainties may affect the accuracy of our estimated impacts of future urban land expansion on biodiversity at the country and ecoregional levels. Nevertheless, we believe that future research can provide more accurate answers to these questions with the help of more refined SSP database.”

3. I like the idea to analyse biodiversity impacts by combining different indicators (habitat loss, fragmentation and species loss). Nevertheless, the authors need to explain in more detail why they have chosen this particular set of indicators and which different aspects of biodiversity impacts the aim to analyse – why is one indication not sufficient?

Response:

Thank you for pointing this issue out. Due to limited space in the main text, we have now explained why we used multiple indicators for biodiversity in the Supplementary Information (p. 2).

“Biodiversity refers to the variety of organisms from all sources, including inter alia, terrestrial, marine, and other aquatic ecosystems and the ecological complexes they comprise. Thus, biodiversity is an ecological complex that includes not only the diversity of species but also the diversity of ecosystems. Accordingly, biodiversity conservation is a multi-dimensional process that requires us not only to protect species, but also to protect their habitat and the surrounding environment. In this study, we combined different indicators for biodiversity (i.e., habitat loss, habitat fragmentation, and species loss) to analyze how biodiversity is affected by future urban expansion. We considered habitat loss and fragmentation mainly because habitat changes are closely related to species changes. Indeed, species rely on habitat to survive, and habitat loss and fragmentation are major causes of species loss. Thus, we first examined how future urban land expansion affects direct loss of natural habitat. To clarify the impacts of future urban growth on habitat loss in key and hotspot biodiversity areas, i.e., the world’s most

important places for species and their habitats (e.g., protected areas, biodiversity hotspots, Global 200, and the Last of the Wild areas), we also investigated the impacts of future urban expansion on biodiversity prioritization schemes. Second, we examined habitat fragmentation that is captured by edge proximity, edge density, and isolation. Finally, we focus on the species and examined the effects of future urban expansion on species richness, species abundance, and number of species. We believe that by using multiple indicators of biodiversity, we can comprehensively examine the how future urban expansion will affect biodiversity. Given the multiple dimensions of biodiversity, one indicator is not sufficient for providing a broader picture.”

4. The discussion should be reworked considerably. There are many statements that are vague and not supported by literature (see comments below).

Response:

We have substantially revised the Discussion to make it more compact in structure and clear in content (pp. 16–25). For instance, we have restructured the Discussion, cut some redundant paragraphs, revised all vague statements, and added extra discussions about some key points raised by you and other reviewers. We hope you agree that this is a much-improved Discussion section.

In addition, I like to comment on some specific aspects of the paper:

1. Page 2: The authors should use consistent data regarding projections of future population. I think the SSPs that were analysed differ from the data cited in the first paragraph.

Response:

Thank you for pointing this issue out. Indeed, the projections of future population of the United Nations are different from the SSP database that was used for creating database for future urban expansion in the current study. To be consistent across the paper, we have now referred to SSP population projections data in the Introduction. Notably, according to the SSP projections, the world’s population is projected to reach 6.9–12.6 billion by 2100 across SSPs scenarios. Under SSP1 and SSP5, the future population is shrinking. To avoid confusion and be consistent across our paper, we report future population data up to 2050. The first sentence has been revised into “The world’s population is projected to reach 8.5–9.9 billion by 2050¹, with 55–78% living in urban areas ².” (p. 2, line 40–41)

References:

1. Kc, S., Lutz, W., 2017. The human core of the shared socioeconomic pathways: Population scenarios by age, sex and level of education for all countries to 2100. *Global Environmental Change* 42, 181–192.
2. Jiang, L., O’Neill, B.C., 2017. Global urbanization projections for the Shared Socioeconomic Pathways. *Global Environmental Change* 42, 193–199.

2. Page 4: Please explain why you have chosen this specific set of biodiversity indicators (see comment above). How are they interrelated?

Response:

Thank you for pointing this issue out. Due to limited space in the main text, we have now explained why we used multiple indicators for biodiversity in the Supplementary Information (p. 2).

“Biodiversity refers to the variety of organisms from all sources, including inter alia, terrestrial, marine, and other aquatic ecosystems and the ecological complexes they comprise. Thus, biodiversity is an ecological complex that includes not only the diversity of species but also the diversity of ecosystems. Accordingly, biodiversity conservation is a multi-dimensional process that requires us not only to protect species, but also to protect their habitat and the surrounding environment. In this study, we combined different indicators for biodiversity (i.e., habitat loss, habitat fragmentation, and species loss) to analyze how biodiversity is affected by future urban expansion. We considered habitat loss and fragmentation mainly because habitat changes are closely related to species changes. Indeed, species rely on habitat to survive, and habitat loss and fragmentation are major causes of species loss. Thus, we first examined how future urban land expansion affects direct loss of natural habitat. To clarify the impacts of future urban growth on habitat loss in key and hotspot biodiversity areas, i.e., the world’s most important places for species and their habitats (e.g., protected areas, biodiversity hotspots, Global 200, and the Last of the Wild areas), we also investigated the impacts of future urban expansion on biodiversity prioritization schemes. Second, we examined habitat fragmentation that is captured by edge proximity, edge density, and isolation. Finally, we focus on the species and examined the effects of future urban expansion on species richness, species abundance, and number of species. We believe that by using multiple indicators of biodiversity, we can comprehensively examine the how future urban expansion will affect biodiversity. Given the multiple dimensions of biodiversity, one indicator is not sufficient for providing a broader picture.”

3. Page 5, line 98: *Are UK and Germany separate countries in your model framework or part of the “Western Europe” region? (see comment above).*

Response:

To clarify, UK and Germany are not separate countries in our model framework. Our urban expansion model framework was based on the 32 macro regions of SSP definition (<https://tntcat.iiasa.ac.at/SspDb/dsd?Action=htmlpage&page=10>). UK and Germany are part of the EU15 region. This region includes European Union member states that joined in EU prior to 2004, namely Austria, Belgium, Denmark, Finland, France, Germany, Greece, Ireland, Italy, Luxembourg, Netherlands, Portugal, Spain, Sweden, and United Kingdom. The uncertainty caused by region definition or aggregation in our model framework has now been discussed in the Discussion section (p.22, line 414–422).

4. Page 5, line 106ff: *Urban area expansion is not only driven by population number but also by per capita area demand for housing, infrastructure etc. E.g. in Europe, we find regions with shrinking population and growing urban areas. How is this effect captured in the scenarios?*

Response:

To clarify, we built our analysis on the future urban expansion datasets from previous research (Chen

et al., 2020). Chen et al. (2020) used panel data regression to estimate future urban land areas based on the factors of population, urbanization rate (i.e., percentage of urban population to total population) and gross domestic product (GDP). Specifically, they used historical statistical data and urban land data to build regression models in which urban land demand per capita was predicted by GDP per capita and urbanization rate at the macro regional level. The historical urban land data were retrieved from the Global Human Settlement Layer (GHSL) dataset that covers the years 1975, 1990, 2000 and 2014. The concurrent statistical data of population, GDP per capita, and urbanization rate were collected from the World Bank and United Nations and aggregated at the macro regional level. The established regression models can then be applied to predict future urban land demand per capita using the scenario projections of GDP per capita and the urbanization rate in the SSPs database (<https://tntcat.iiasa.ac.at/SspDb>). The regional urban land demand in a future year t was thus obtained through multiplying the estimated urban land demand per capita in year t by the projected total regional population in the same year.

Thus, in this model framework, regions with shrinking populations and growing urban areas in Europe, as you mentioned, were also captured in the SSP scenarios. It is assumed that even if the population shrinks, the urbanization rate and GDP can still grow, and there is still a certain urban land demand. Only when population, urbanization, and economic growth all stagnate, the urban land demand will stagnate, and then urban land may not continue to expand. We have now discussed about the link between population shrinkage and urban expansion in the Methods section (p.26, line 502–507).

“It is assumed that even if the population shrinks, the urbanization rate and GDP can still grow, and there is still a certain urban land demand. Only when population, urbanization, and economic growth all stagnate, the urban land demand will stagnate, and then urban land may not continue to expand. Thus, the scenario of regional population shrinking but urban area growing can be captured by this model framework (for example, cities in Eastern Europe, see Supplementary Figs. 29 and 30).”

5. Page 6ff: Avoid expression as “extremely large” or “even worse” when presenting results. Interpretation should be done in the discussion section.

Response:

We have avoided using “extremely large” or “even worse” when presenting the results (p.7, line125–126).

6. Page 6ff: In this and the following sections you refer to CCI land-cover classes. But it seems that your model uses a much more aggregated classification (see SI Table 1). Please explain how you conducted this analysis?

Response:

To clarify, we used a more aggregated classification for ease of calculation and presentation of results. Compared to the original CCI land cover data that includes 22 classes, the aggregate six classes (i.e., forest, shrubland, grassland, wetland, urban, and other land) are more general land cover classes and

are easier for readers to understand. We have articulated how we have reclassified ESA-CCI land cover classes in this study in Supplementary Table 1 (p. 45). This reclassification system was based on the Land Cover CCI Product User Guide version 2.0 (van Vliet, 2019) and EUNIS Habitat Classification Revised 2004. Specifically, we reclassified the detailed land cover classes as general land classes in ArcGIS Pro V2.4. For example, we merged the detailed land cover category—10-Cropland, rainfed, 20-Cropland, irrigated or post-flooding, 30-Mosaic cropland (>50%) / natural vegetation (tree, shrub, herbaceous cover) (<50%), and 40-Mosaic natural vegetation (tree, shrub, herbaceous cover) (>50%) / cropland (<50%)—into a single classification “cropland”.

References:

van Vliet, J. Direct and indirect loss of natural area from urban expansion. *Nature Sustainability* 2, 755–763 (2019).

7. Page 8: Please explain your assumptions regarding effectivity of protected areas within the different scenarios.

Response:

Thank you for pointing this issue out. This issue and the urban planning assumptions were also raised by other reviewers. Based on the SSP environmental narratives, we have now summarized the assumptions regarding effectivity of protected areas for the five SSP scenarios in the Supplementary Information Table 9 (pp. 59–60; see also below).

Supplementary Table 9. Assumptions for urban planning and protected areas under SSPs.

Scenarios	Urban planning assumptions	Protected areas assumptions (Environment assumptions)
SSP1	Urbanization is well managed, and urban planning is promoted in tandem with high urbanization rates.	It is an environment-friendly development pattern to strengthen the protection of fragile ecosystems and regions such as protected areas. Land use is strictly regulated. Urban expansion has barely encroached on protected areas. Protected areas are effective. Protected areas are on track to meet Aichi’s 17% target due to strong land-use change regulation.
SSP2	Moderate urban planning regulation.	Growing energy demand has led to the continuous environmental deterioration. Moderate regulation of land use leads to a slow decline in deforestation rates. Moderate land use regulation makes the effectiveness of protected areas in the middle level. Protected areas are being moderately encroached upon. Protected areas are expected to meet the Aichi target of 17% of land area due to moderate land use change regulation gradually implemented from 2010–2050.
SSP3	Urban settlements are poorly planned, particularly in developing countries where	Not enough attention has been paid to solving environmental problems, resulting in serious environmental degradation in some areas.

	inequality and fragmentation cause mixed pattern of urban change.	Deforestation continues because of a lack of regulation, competition for land and the rapid expansion of agriculture. Poor land use regulation leads to low effectiveness of protected areas. Urban expansion has encroached heavily on protected areas. Protected areas are under serious threat.
SSP4	Spatial development pattern varies across cities, with urban sprawl dominating in some cities, whereas better planning in cities that are predominantly inhabited by the higher-income groups leads to more concentrated development.	There are significant differences in environmental conditions. On the one hand, there are some areas of world concern, close to the places where middle- and high-income groups live and vacation, which are well managed. On the other hand, resource and production areas and many other out of sight places are neglected and become deteriorated. Conservation of protected areas is also divided, with highly regulated and well-managed areas in middle- and high-income countries, but largely unmanaged and deteriorating areas in low-income countries.
SSP5	It is difficult for urban planning to keep up with high urbanization rates, and the sprawling pattern of development is dominant.	Regulations are imperfect, and many protected areas are not effectively protected. Protected areas are under serious threat.

8. Page 10, line 190ff: Explain the differences between the SSPs.

Response:

We have now explained the specific differences in the habitat edge areas affected by urban expansion across the five SSPs in the paper (p.10, line 188–192).

“The effects of urban expansion on habitat edge are remarkably different across different scenarios. Specifically, the area of affected habitat edges is expected to be 38.45 Mha, 34.24 Mha, 40.31 Mha, 37.84 Mha, and 39.42 Mha from SSP1 to SSP5 by 2100, with the smallest effect under scenario SSP2, and the largest effect under scenario SSP3.”

9. Page 16, line 298: What are these opportunities? How are they related to the scenario storylines?

Response:

To clarify, our results imply that countries around the world should take strict measures to adjust their current urban development trajectories and urbanization patterns to pursue SSP1 sustainable development path. Such measures include, but are not limited to, environment-friendly living arrangements, compact urban form planning, enhancing resource efficiency, and protecting vulnerable ecosystems and regions. These measures are directly related to the development path envisioned by SSP1 scenario (see the detailed description of all SSP scenarios in the Supplementary Information, pp. 3–5).

We have revised this sentence to specify the opportunities in the paper (p.17, line 297–304).

“Nevertheless, there are still opportunities to adjust future urban development trajectories and to intervene against the long-term negative impacts of urbanization on biodiversity. In particular, if the sustainable pathway (i.e., scenario SSP1) of urban expansion is properly implemented, humans will be able to maintain a relatively low natural habitat loss, low habitat fragmentation, and a high level of species conservation (SSP1 with lowest cropland expansion and higher forest area is also generally more beneficial for biodiversity conservation according to several existing estimations^{30,31}).”

10. Page 16, line 301: In which sense and why does your study show significantly different results than the cited articles?

Response:

We found significantly different results from previous studies (Güneralp et al., 2013; Seto et al., 2012; Zhou et al., 2019) mainly because these early studies did not take into account the issue of future population shrinkage, whereas we considered this issue based on the SSP database. In particular, Seto et al. (2012) and Güneralp et al. (2013) only forecasted global urban expansion till 2030, but many Asian countries will not experience evident population shrinkage until 2030. In addition, different sources of data for analysis may be another reason. For instance, Seto et al. (2012) and Güneralp et al. (2013) used five sources of data to forecast urban expansion: (a) global urban extent circa 2000 from National Aeronautics and Space Administration’s Moderate Resolution Imaging Spectroradiometer, (b) urban population projections to 2030 from the United Nations, (c) population projection uncertainty ranges from the US National Research Council, (d) population density estimates from the Global Rural-Urban Mapping Project, and (e) country-level GDP projections by the Intergovernmental Panel on Climate Change Special Reports on Emissions Scenarios. In contrast, we used the latest recognized scenarios of the SSPs for simulating future urban land dynamics, which enabled us to better connect with existing climate research and promote future climate research. Overall, we used the SSP database (GDP per capita and the urbanization rate), GHSL dataset (the historical urban land areas), the World Bank and United Nations dataset (statistical data of population, urbanization rate and GDP) to conduct our analyses. Our datasets are obviously different from the datasets used by Seto et al. (2012) and Güneralp et al. (2013) in terms of data source, data resolution (1 km vs. 5 km), and time span (2015–2100 vs. 2000–2030).

Zhou et al. (2019) used SLEUTH model to simulate future land changes based on the historical trends captured by a series of historical land maps. However, their simulation used only a single scenario based on historical trajectories, and did not consider the potential pathways and uncertainties of future socio-economic factors and was therefore incompatible with the recent IPCC framework. In addition, the SLEUTH model can only have five spatial driving layers (slope, excluded, urban, transportation, hillshade) for the urban land simulation, which correspond to the five coefficients that control the behavior of the system. These coefficients provide convenience for explaining the mechanism of the model, but could not consider more factors in the simulation process.

Different from Zhou et al. (2019), the projections of future urban expansion that our research was based on (Chen et al., 2020) were generated using FLUS model coupled with the latest SSPs. SSPs

describe potential pathways for policy assumptions and the socio-economic narratives in the 21 century and is consistent with the recent IPCC framework. Thus, these projections offer the potential to better support research in relevant areas, such as ecological protection, water security, urban climate, and global climate change.

We have specified this difference in datasets and models between our study and previous studies in the Discussion (pp. 20–21, line 368–388).

“Second, previous studies on the effects of urban expansion often used datasets with lower resolution (e.g., 5 km) and shorter time span (e.g., 2000–2030)^{15,19}, or used SLEUTH model²³ or SELECT model¹⁸ to simulate future urban land changes. Yet, low-resolution data can cause overestimation of future urban expansion³⁸. To reduce overestimation, we used more recent datasets with higher resolution (1 km) and longer time span (2015–2100). Besides, we adopted a more advanced Future Land-Use Simulation (FLUS) model that can couple with the latest SSPs¹⁷ to simulate future urban expansion³⁹. This model has been proved to be able to explicitly simulate the spatial trajectories of multiple land cover changes under different scenarios by coupling human-related and natural environmental impacts³⁹. Compared to McDonald et al.⁹, our estimated habitat loss due to future urban expansion (11–33 million ha by 2100) is smaller than theirs (which had 29 million ha between 2000 and 2030). The main reason may be that the projection of urban expansion used by McDonald et al.⁹ is up to 2030 derived from Seto et al.¹⁵ and this projection did not fully capture the dynamics of future urban land use due to coarse spatial resolution, and did not consider future population decline trend that may cause stagnation in urban land growth. For example, the simulation that our research was based on showed that most regions in the world will continue to experience urban expansion, but a few regions (e.g., China and other Asian countries) will have a relatively low urban expansion rate or even a decline in urban land demand after 2050 (see Supplementary Figs. 7–11 for China), which is different from Seto et al.¹⁵.”

References:

1. Seto, K. C., Güneralp, B. & Hutyrá, L. R. Global forecasts of urban expansion to 2030 and direct impacts on biodiversity and carbon pools. *Proc. Natl. Acad. Sci. U. S. A.* 109, 16083–16088 (2012).
2. Zhou, Y., Varquez, A. C. G. & Kanda, M. High-resolution global urban growth projection based on multiple applications of the SLEUTH urban growth model. *Sci. Data* 6, 34 (2019).
3. Güneralp, B. & Seto, K. C. Futures of global urban expansion: uncertainties and implications for biodiversity conservation. *Environ. Res. Lett.* 8, 014025 (2013).

11. Page 18, line 326: *The must be research on these issues. Please give references.*

Response:

We have now cited relevant references to support our point of view (p.17, line 307–313).

“Specifically, natural habitat loss will increase by 694–1509% from 2020 to 2100 across the five SSP scenarios, and the loss in natural habitat showed substantial spatial heterogeneity across different habitat types, biodiversity hotspots, biomes, and ecoregions. Notably, different habitat types, biodiversity hotspots, biomes, and ecoregions have different ecological functions

and biodiversity values²⁹, which future conservation policies should take into account ^{9,19,20,32}.”

References:

1. McDonald, R.I., Güneralp, B., Huang, C.-W., Seto, K.C., You, M., 2018. Conservation priorities to protect vertebrate endemics from global urban expansion. *Biological Conservation*, 224, 290–299.
2. McDonald, R.I., Mansur, A.V., Ascensão, F., Colbert, M.I., Crossman, K., Elmqvist, T., Gonzalez, A., Güneralp, B., Haase, D., Hamann, M., Hillel, O., Huang, K., Kahnt, B., Maddox, D., Pacheco, A., Pereira, H.M., Seto, K.C., Simkin, R., Walsh, B., Werner, A.S., Ziter, C., 2019. Research gaps in knowledge of the impact of urban growth on biodiversity. *Nature Sustainability*, 3, 16–24.
3. Kehoe, L., Romero-Munoz, A., Polaina, E., Estes, L., Kreft, H., Kuemmerle, T., 2017. Biodiversity at risk under future cropland expansion and intensification. *Nature Ecology & Evolution*, 1, 1129–1135.
4. Güneralp, B., Seto, K.C., 2013. Futures of global urban expansion: uncertainties and implications for biodiversity conservation. *Environmental Research Letters*, 8, 014025.
5. Jenkins, C.N., Pimm, S.L., Joppa, L.N., 2013. Global patterns of terrestrial vertebrate diversity and conservation. *Proceedings of the National Academy of Sciences of the United States of America*, 110, E2602–2610.

12. Page 18, line 332ff: *Please add references that support your theses.*

Response:

We have now added references that support our findings (p. 17, line 313–315).

“Moreover, the key biodiversity hotspots and ecologically vulnerable ecoregions that are currently less disturbed by humans will suffer the highest percentage of urban growth^{15,20}.”

References:

1. Seto, K.C., Güneralp, B., Hutyrá, L.R., 2012. Global forecasts of urban expansion to 2030 and direct impacts on biodiversity and carbon pools. *Proceedings of the National Academy of Sciences*, 109, 16083–16088.
2. McDonald, R.I., Güneralp, B., Huang, C.-W., Seto, K.C., You, M., 2018. Conservation priorities to protect vertebrate endemics from global urban expansion. *Biological Conservation*, 224, 290–299.

13. Page 19, line 360: *Give evidence why it is understudied.*

Response:

A careful review of previous research suggests that studies on the global scale have mainly focused on the impact of future urban land expansion on natural habitats, biodiversity hotspots, and protected areas^{13,28,30,35}. For instance, McDonald et al. (2018) combined spatially explicit global forecasts of urban expansion, information on terrestrial vertebrate endemism, and data on current land cover and protected areas to define conservation priorities. However, this study is ecoregion-based and does not characterize the specific effects of future urban expansion on species abundance and richness at a fine-grained scale. Notably, Newbold et al. (2015) estimated within-sample (local) species richness and abundance as a function of human pressures, which opened up a new research field on the

relationship between human pressures and local species richness and abundance. Yet, Newbold et al. (2015) did not examine how future urban expansion affects losses of species richness and abundance. Moreover, in a recent review on current research advances on urban growth and biodiversity, McDonald et al. (2019) also suggested that the effects of future urban expansion on global losses of species richness and abundance have been understudied.

We have now revised the original text in the paper to clarify this issue (p. 19–20, line 359–363).

“First, previous studies on the global scale mainly focused on how future urban land expansion affects natural habitats and biodiversity prioritization areas^{14,15,19,20}, and less attention has been paid to its impact on habitat fragmentation and losses of species richness and abundance^{9,16}.”

References:

1. McDonald, R.I., Güneralp, B., Huang, C.-W., Seto, K.C., You, M. Conservation priorities to protect vertebrate endemics from global urban expansion. *Biological Conservation*, 224, 290–299 (2018).
2. Newbold, T. et al. Global effects of land use on local terrestrial biodiversity. *Nature*, 520, 45–50 (2015).
3. McDonald, R.I., et al. Research gaps in knowledge of the impact of urban growth on biodiversity. *Nature Sustainability*, 3, 16–24 (2019).
4. McKinney, M.L. Urbanization, Biodiversity, and Conservation. *Bioscience* 52, 883-890 (2002).
5. Seto, K.C., Güneralp, B., Hutyra, L.R. Global forecasts of urban expansion to 2030 and direct impacts on biodiversity and carbon pools. *Proceedings of the National Academy of Sciences*, 109, 16083–16088 (2012).
6. Güneralp, B. & Seto, K. C. Futures of global urban expansion: uncertainties and implications for biodiversity conservation. *Environ Res Lett* 8, 014025 (2013).

14. Page 25, line 500ff: I do not understand why you do an additional analysis on a coarser grid and how you combine the described data sets. Please explain in more detail.

Response:

In our study, we used many datasets with various levels of resolution to analyze the impacts of future urban expansion on biodiversity. Specifically, we use land-system data with 1-km resolution rather than CC-LC data with a finer 300-m resolution because CCI-LC data did not have information for land use intensity. We used two indicators to describe the impacts of future urban expansion on biodiversity, namely the percentage of the mean biodiversity loss with 1-km grid and the potential number of species loss with 10-km grid. Analysis of the percentage of mean biodiversity loss was based on 1-km grid in order to match the future urban expansion data. However, the original species richness data were based on 10-km grid, which could not be downscaled to 1-km grid because species richness calculation was based on 10-km grid. Thus, we estimated the potential number of species loss based on a 10-km grid.

15. Page 27: Here, more detailed information about PREDICTS and the matching process between land systems and intensity is required. A table with the coefficients used for the analysis should be

added to the SI.

Response:

To address your comments, we have added detailed information about PREDICTS (see Supplementary Note 4, p. 7) and a table with the coefficients used for the analysis (see Supplementary Table 10, pp. 61–62).

16. SI, figure 1: This type of plots is typically used to visualize uncertainty (e.g. a range of results between scenarios) rather than a range between different independent sets of simulation results – in this case 32 regions. Please use another graphic representation.

Response:

We have changed the graphic representation based on your suggestion. The new Supplementary Fig. 1 now represents the range of urban expansion simulation results across five SSP scenarios.

Supplementary Fig. 1 Urban expansion projections by 2100 under SSP scenarios. These 32 macro regions are defined as follows: ANUZ = Australia and New Zealand. BRA = Brazil. CAN = Canada. CAS = countries in Central Asia, including Armenia, Azerbaijan, Georgia, Kazakhstan, Kyrgyzstan, Tajikistan, Turkmenistan, and Uzbekistan. CHN = China (Mainland, Hongkong, Macao; excl. Taiwan), including China, Hong Kong SAR (China), Macao SAR (China). EEU = Eastern Europe (excl. former Soviet Union and EU member states), including Albania, Bosnia and Herzegovina, Croatia, Montenegro, Serbia, The former Yugoslav Republic of Macedonia. EEU-FSU = Eastern Europe, former Soviet Union (excl. Russia and EU members). Belarus, Republic of Moldova, Ukraine. EFTA = Iceland, Norway, and Switzerland. EU12-H = New EU member states that joined as of 2004 - high income. Cyprus, Czech Republic, Estonia, Hungary, Malta, Poland, Slovakia, Slovenia. EU12-M = medium-income New EU member states that joined as of 2004, including Bulgaria, Latvia, Lithuania, and Romania. EU15 = European Union member states that joined prior to 2004, including Austria, Belgium, Denmark, Finland, France, Germany, Greece, Ireland, Italy, Luxembourg, Netherlands, Portugal, Spain, Sweden, and United Kingdom. IDN = Indonesia. IND = India. JPN = Japan. KOR = Republic of Korea. LAM-L = low-income countries in Latin America (excl. Brazil,

Mexico), including Belize, Guatemala, Haiti, Honduras, and Nicaragua. LAM-M = medium- and high-income countries in Latin America (excl. Brazil, Mexico), including Antigua and Barbuda, Argentina, Bahamas, Barbados, Bermuda, Bolivia (Plurinational State of), Chile, Colombia, Costa Rica, Cuba, Dominica, Dominican Republic, Ecuador, El Salvador, French Guiana, Grenada, Guadeloupe, Guyana, Jamaica, Martinique, Netherlands Antilles, Panama, Paraguay, Peru, Saint Kitts and Nevis, Saint Lucia, Saint Vincent and the Grenadines, Suriname, Trinidad and Tobago, Uruguay, and Venezuela (Bolivarian Republic of). MEA-H = high-income countries in Middle East Asia, including Bahrain, Israel, Kuwait, Oman, Qatar, Saudi Arabia, and United Arab Emirates. MEA-M = low- and medium-income countries in Middle East Asia, including Iran (Islamic Republic of), Iraq, Jordan, Lebanon, Occupied Palestinian Territory, Syrian Arab Republic, and Yemen. MEX = Mexico. NAF = countries in North Africa, including Algeria, Egypt, Libyan Arab Jamahiriya, Morocco, Tunisia, and Western Sahara. OAS-CPA = countries in Other Asia (i.e., former Centrally Planned Asia), including Cambodia, Lao People's Democratic Republic, Mongolia, and Viet Nam. OAS-L = low-income countries in Other Asia, including Bangladesh, Democratic People's Republic of Korea, Fiji, Micronesia (Fed. States of), Myanmar, Nepal, Papua New Guinea, Philippines, Samoa, Solomon Islands, Timor-Leste, Tonga, and Vanuatu. OAS-M = medium- and high-income countries in Other Asia, including Bhutan, Brunei Darussalam, French Polynesia, Guam, Malaysia, Maldives, New Caledonia, Singapore, Sri Lanka, and Thailand. PAK = Pakistan and Afghanistan. RUS = Russian Federation. SAF = South Africa. SSA-L = low-income countries in Sub Sahara Africa (excl. South Africa), including Benin, Burkina Faso, Burundi, Cameroon, Cape Verde, Central African Republic, Chad, Comoros, Congo, Côte d'Ivoire, Democratic Republic of the Congo, Djibouti, Eritrea, Ethiopia, Gambia, Ghana, Guinea, Guinea-Bissau, Kenya, Lesotho, Liberia, Madagascar, Malawi, Mali, Mauritania, Mozambique, Niger, Nigeria, Rwanda, Sao Tome and Principe, Senegal, Sierra Leone, Somalia, South Sudan, Sudan, Swaziland, Togo, Uganda, United Republic of Tanzania, Zambia, and Zimbabwe. SSA-M = medium- and high-income countries of Sub Sahara Africa (excl. South Africa), including Angola, Botswana, Equatorial Guinea, Gabon, Mauritius, Mayotte, Namibia, Réunion, and Seychelles. TUR = Turkey. TWN = Taiwan. USA = United States of America. Includes: Puerto Rico, United States Virgin Islands, United States of America.

17. SI, figure 2: The colour scheme should be changed. I cannot recognize anything.

Response:

Following your suggestion, we have changed the color scheme of Supplementary Fig. 2. To get the spatial details, we now divided the original Supplementary Fig. 2 into five figures (i.e., Figs. 2–6) in the Supplementary Information.

Reviewer #3

The manuscript addresses an important topic. Especially I found the contribution to expected level of fragmentation -in addition to loss of habitat extents- of habitats due to future urban land expansion valuable. However, there are a few methodological and interpretation issues that need to be addressed. For example, the authors need to be measured in their claims. Their study is not an empirical one, in the sense that their findings are not verifiable -yet- by observation. Certain methodological details are vague (e.g., the nature of the urban land expansion forecasts) and

important assumptions one needs to make in a study with such a long time horizon are not even discussed. Most importantly, I am not convinced that their findings lead to any novel insights beyond what is already reported in the literature by several studies. The manuscript is well-written overall though there are awkward phrases and typos throughout (e.g., (ln 66) “most of which”, (ln 174) “slight”). I listed my detailed comments below:

Response:

Thanks for your comments. To clarify, we build our research on the projection data of future urban expansion created by Chen et al. (2020). Although future urban expansion, along with its potential effects on biodiversity, is not verifiable yet by observation, an answer to this question will have important practical implications for global sustainable development. Moreover, the future urban expansion projection dataset that we used (Chen et al., 2020) had their advantages (e.g., higher resolution, diverse scenario settings, better connected with existing climate change) compared with other simulations on future urban expansion (Seto et al, 2012; Zhou et al., 2019). Chen et al. (2020) also strictly validated their models and assessed the performance of their simulations. Thus, using this dataset can help us provide more robust predictions of how future urban expansion will affect biodiversity in various domains.

We have substantially revised our paper in several respects to address your comments above. First, we have clarified the methodological details about the simulations of urban land expansion in the paper (pp. 25–26, line 486–513). Overall, we think the nature of the urban land expansion forecasts includes (a) forecasts of future urban land demand and (b) spatial simulation or allocation of urban land expansion. We build our research on the projection data of future urban expansion created by Chen et al. (2020). They used SSP database, historical urban land data, and panel data model to estimate future urban land demand, and used the Future Land-Use Simulation (FLUS) model to simulate future urban land expansion at a 1-km resolution.

Second, we have added a full description of the assumptions and development paths envisioned by the five SSP scenarios in the Supplementary Information Note 2 (pp. 3–5).

Third, our findings provided several novel insights that went beyond previous research. One novelty is that we integrated a global SSP multi-scenario framework with biodiversity analysis rather than the traditional single scenario analysis. Another novelty was that we assessed the impacts of future urban expansion on edge proximity and landscape fragmentation of natural habitats near urban areas, whereas previous research often focused on loss of habitat. In addition, we explored the relative percentage changes in species richness and species abundance in 1-km grids and examined the potential mean absolute change in local-site species richness numbers on 10-km grids, based on the model estimates of biodiversity responses to future urban land cover change from the PREDICTS database. Taken together, our work provides new insights into the relation between global urban land change and biodiversity, particularly in terms of how future urban expansion may affect biodiversity that is characterized by direct habitat loss, habitat fragmentation, species richness loss and species abundance loss, and number of species loss. We believe our work paves the way for more advanced studies on the social-ecological interaction related to future urbanization. We have now emphasized the novelty of our research compared with other relevant studies in the Discussion section (pp. 19–21, line 359–391).

Finally, we have carefully proofread our paper and have revised those awkward phrases and typos.

1. (ln 30) *State how much more instead of “much higher proportion”.*

Response:

Thank you. We have now revised this sentence into “The current key biodiversity priority areas (e.g., WWF’s Global 200) may undergo a much 37–44% of urbanized land than the global average”. (p. 2, line 30–32)

2. (ln 44) *Ref 3: There is a more recent study that reports similar findings but based on a larger number of works with more up-to-date and more detailed estimates on urban land expansion vs urban population growth. I suggest that the authors consider citing this more recent and more up-to-date study: Güneralp et al. (2020).*

Response:

Thank you for your suggestion. We have now cited the recent study by Güneralp et al. (2020) (p. 2, line 43).

References:

Güneralp, B., Reba, M., Hales, B. U., Wentz, E. A. & Seto, K. C. Trends in urban land expansion, density, and land transitions from 1970 to 2010: a global synthesis. *Environ Res Lett* 15 (2020).

3. (lns 45) *Consider writing “a major driver” instead of “the major driver” because there are other major drivers of land-use change globally including agricultural land expansion and deforestation. See Foley et al. (2005).*

Response:

Thank you for your suggestion. We have changed “the major driver” into “a major driver” and cited Foley et al. (2005) (p. 2, line 44-45).

References:

Foley, J. A. et al. Global consequences of land use. *Science* 309, 570-574 (2005).

4. (lns 53-54) *“...more than 1.8–5.9 54 times urban land expansion from 2000 to 2100...” ∅
Confusing, please rephrase.*

Response:

Sorry for causing this confusion. To clarify, 54 is the line number. To avoid confusion, we have rephrased this sentence into “...the global total urban area in 2100 will be roughly 1.8 to 5.9 times of that in 2000” (p. 3, line 53–54).

5. (lns 71-73) *To what extent does this assumption hold for cities across Eastern Europe where there has been widespread contraction across cities?*

Response:

We also considered the urban contraction or shrinkage in our study for Eastern Europe and other regions. It is important to note that urban shrinkage does not necessarily lead to a reduction of urban land demand. We assumed that future urban land demand is based on several factors, including population, urbanization rate (percentage of urban population to total population), and gross domestic product (GDP). If urban land demand decreases, the growth rate of urban expansion will decrease. A stagnation in urban expansion can also occur when population declines, and at the same time urbanization rates and GDP stagnate. Although cities across Eastern Europe have experienced widespread urban shrinkage or population decline, the demand for urban land is increasing (Supplementary Fig. 29). The main reason for this growing urban land demand is that although the total population has decreased, the urbanization rate and GDP are growing. Further analyses of the projections for future urban expansion also show an increasing trend in this region (Supplementary Fig. 30). Thus, our assumption holds for cities across Eastern Europe.

Supplementary Fig. 29 Human settlement expansion in the cities of Eastern Europe from 1975 to 2014. Data derived from the global human settlement layer (GHSL).

Supplementary Fig. 30 Future urban expansion in the cities of Eastern Europe from 2015 to 2100 across SSPs. This result based on Chen et al., 2020.

6. The authors make several statements about previous studies on forecasts of urban expansion or on their impacts on biodiversity; however, they do not cite the most relevant studies. For example, they do not support their claim “on only one aspect of impacts on biodiversity,” on ln 66. Which studies are these?

Response:

We have now cited the most relevant studies to support our claim. Findings from these cited studies suggest that the impact of urbanization on biodiversity is multi-dimensional, although most of the existing studies only focus on one aspect.

“Few studies that have quantified the potential effects of urban expansion have mainly focused on only one aspect of impacts on biodiversity^{9,15,19-22}, or projected future urban expansion with a single scenario²³ and coarse resolution (>1km)” (pp. 3–4, line 65–68).

References:

1. McKinney, M. L. Effects of urbanization on species richness: A review of plants and animals.

Urban Ecosystems 11, 161-176 (2008).

2. Aronson, M. F. J. et al. A global analysis of the impacts of urbanization on bird and plant diversity reveals key anthropogenic drivers. *Proceedings of the Royal Society B: Biological Sciences* 281, 20133330 (2014).
3. McDonald, R. I. et al. Research gaps in knowledge of the impact of urban growth on biodiversity. *Nature Sustainability* 3, 16–24 (2019).
4. Seto, K. C., Güneralp, B. & Hutyrá, L. R. Global forecasts of urban expansion to 2030 and direct impacts on biodiversity and carbon pools. *Proceedings of the National Academy of Sciences* 109, 16083–16088 (2012).
5. Güneralp, B. & Seto, K. C. Futures of global urban expansion: uncertainties and implications for biodiversity conservation. *Environ Res Lett* 8, 014025 (2013).
6. McDonald, R. I., Güneralp, B., Huang, C.-W., Seto, K. C. & You, M. Conservation priorities to protect vertebrate endemics from global urban expansion. *Biol Conserv* 224, 290–299 (2018).
7. Zhou, Y., Varquez, A. C. G. & Kanda, M. High-resolution global urban growth projection based on multiple applications of the SLEUTH urban growth model. *Scientific Data* 6, 34 (2019).

7. (In 98) How come countries with very little unaltered habitat left and with relatively slow urban expansion such as Germany and the UK are among those that are predicted to have the greatest amount of habitat loss to urban expansion?

Response:

The SSP database shows that it is still possible for the population in 15 European countries (EU15, namely European Union member states that joined prior to 2004, such as Germany and the UK) to grow under the SSP5, SSP1 and SSP2 scenarios (Fig. 2), especially in scenario SSP5 (Fossil-fueled development). In these scenarios, both urbanization and economic growth will also increase (Figs. 3–4). According to our assumption, the increase in these two indicators will drive the increasing urban land demand. Thus, increasing demand for urban land often leads to urban expansion and conversion of natural habitats into urban land.

Fig. 2 Future population growth projections in the EU15 from 2015 to 2100 across SSPs. EU15 = European Union member states that joined prior to 2004, including Austria, Belgium, Denmark,

Finland, France, Germany, Greece, Ireland, Italy, Luxembourg, Netherlands, Portugal, Spain, Sweden, and United Kingdom.

Fig. 3 Future GDP growth projections in the EU15 from 2015 to 2100 across SSPs. EU15 = European Union member states that joined prior to 2004, including Austria, Belgium, Denmark, Finland, France, Germany, Greece, Ireland, Italy, Luxembourg, Netherlands, Portugal, Spain, Sweden, and United Kingdom.

Fig. 4 Future urbanization rate projections in the EU15 from 2015 to 2100 across SSPs. EU15 = European Union member states that joined prior to 2004, including Austria, Belgium, Denmark, Finland, France, Germany, Greece, Ireland, Italy, Luxembourg, Netherlands, Portugal, Spain, Sweden, and United Kingdom.

8. (Ins 108-109) This does not make sense. So, China and India are expected to undergo population shrinkage from 2050 and 2100 but Europe's population will continue to increase such that it will be one of the regions with the most habitat loss to urban expansion? I know of no demographic

projection scenario where Europe's population growth exceeds those of China and India by the end of the century.

Response:

Thank you for raising this issue. We compared the current three demographic projection datasets, including UN-World Population Prospects 2019, IIASA-SSP database (used in this paper) and IHME-Population Forecasting. For most models, we found that China (Figs 5–7) and India (Figs 8–10)—particularly China—are expected to undergo population shrinkage from 2050 and 2100. Meanwhile, Europe's population will not continue to increase, as shown by most models of these three population projection datasets (Figs 11–13). Only under some scenarios—such as scenarios high variant and instant-replacement in UN-World Population Prospects 2019, scenario SSP5 in IIASA SSP database, and scenarios upper-Slower Met Need and Education and upper-Reference in IHME-Population Forecasting—Europe's population will increase. Future population projections are often subject to some uncertainties. The population of Europe may still increase under certain conditions, such as in scenario SSP5. However, in terms of total population, Europe's population growth will not exceed that in China and India by the end of the century. In our study, we found that, only under scenario SSP5, Europe will be a geographically hot spot for habitat loss due to urban expansion. Under the other four scenarios (SSP1-SSP4), the cold spots of urban expansion will particularly exist in eastern and southern Europe. Thus, Europe will not experience the most habitat loss due to urban expansion. We assumed that future urban land demand is based on several factors, including population, urbanization rate, and gross domestic product (GDP) (see also Chen et al., 2020). Thus, population is not the only factor that determines whether there is urban expansion or not.

Fig. 5 Future population projections for China from 2020 to 2100 across scenarios. Data were derived from UN-World Population Prospects 2019.

Fig. 6 Future population projections for China from 2020 to 2100 across scenarios. Data were derived from IIASA-SSP database.

Fig. 7 Future population projections for China from 2020 to 2100 across scenarios. Data were derived from IHME-Population Forecasting.

Fig. 8 Future population projections for India from 2020 to 2100 across scenarios. Data were derived from UN-World Population Prospects 2019.

Fig. 9 Future population projections for India from 2020 to 2100 across scenarios. Data were derived from IIASA-SSP database.

Fig. 10 Future population projections for India from 2020 to 2100 across scenarios. Data were derived from IHME-Population Forecasting.

Fig. 11 Future population projections for European countries from 2020 to 2100 across scenarios. Data were derived from UN-World Population Prospects 2019.

Fig. 12 Future population projections for European countries from 2020 to 2100 across scenarios. Data were derived from IIASA-SSP database.

Fig. 13 Future population projections for European countries from 2020 to 2100 across scenarios. Data were derived from IHME-Population Forecasting.

9. (Ins 142-147) This information is not useful or can even be misleading. The urban forecasts did not account for protected status of the PAs or even their existence. Therefore, it does not mean much what proportion of urban forecasts occupy protected areas of which categories.

Response:

Thank you for pointing this out. We fully agree with you that the information about what proportion of urban forecasts occupy protected areas of which categories is not useful. To avoid confusion, we have now removed this information from the paper.

10. (Ins 145-146) What is the rationale to assume no category PAs as having intermediate status?

Response:

Following your earlier suggestion, we have now removed this information from the paper.

11. (ln 179-ln 480) The authors' use of the term "edge effects" is incorrect. Edge effect refers changes in community or population structure at the boundary of two or more habitats. What the authors refer to as edge effects appears to be proximal effects of urban areas on nearby habitats.

Response:

Thank you for your suggestion. We have changed "edge effects" to "proximal effects" across the paper.

12. (lns 213-215) Supporting citations?

Response:

We have added relevant citations to support our statements (p. 12, line 215–219).

"Future urban expansion also tends to cause an increase in the edge density of natural habitat, which often reflects smaller patches or more irregular shapes and poses a threat to biodiversity that influences many ecological processes (e.g., the spread of dispersal and predation)^{13,27,28}."

References:

1. Fahrig, L. Effects of habitat fragmentation on biodiversity. *Annual Review of Ecology, Evolution, and Systematics* 34, 487–515 (2003).
2. Haddad, N. M. et al. Habitat fragmentation and its lasting impact on Earth's ecosystems. *Science Advances* 1, e1500052 (2015).
3. Oliver, T. H. et al. Interacting effects of climate change and habitat fragmentation on drought-sensitive butterflies. *Nature Climate Change* 5, 941 (2015).

13. (lns 286-288) Empirical? What the authors did is taking urban expansion forecasts and several spatial biodiversity/habitat datasets to generate future projections on biodiversity/habitat impacts of urban expansion. Being a forward-looking study, there is nothing that is verifiable by observation in these analyses. The authors need to be measured in their claims.

Response:

Thank you for pointing this out. To avoid misinterpretation, we have removed this claim from the revised manuscript.

14. (lns 312-316) First, the sentence is too long. Second, it is crammed with catch phrases (e.g., urban development path, growth-oriented urban planning paradigm) with no insight (at least no novel insight) that we did not know from the previous studies. Third, there are no references cited to support any of its parts.

Response:

To address your concerns, we have now removed the original sentence from the revised manuscript. The relevant practical implications were summarized in the last paragraph of the discussion.

15. (lns 326-331) Similar to the preceding comment, these comments amount to hand-waving repeating what is already well-known in conservation community, providing no new insights.

Response:

Following your suggestion, we have revised the relevant implications and statements. Specifically, we have now offered new perspectives, insights, and practical implications that are directly derived from our findings (p. 25, line 474–484).

“Finally, we suggest that a new framework for biodiversity conservation should be established based on the dominant functions of ecosystems and human pressures^{55,56}. In this framework, the ecosystem can be divided into three types of space based on the dominant function type of region: urban space, agricultural space, and ecological space. Correspondingly, the three control lines—urban development boundary, farmland protection line, and ecological conservation line—can be delineated respectively and different governance measures can be implemented to balance the relationship between urban development, food security, and biodiversity conservation⁵⁷. In this way, global and local biodiversity conservation responsibilities can be effectively identified in a differentiated and explicit way.”

References:

1. Bai, Y. et al. Developing China's Ecological Redline Policy using ecosystem services assessments for land use planning. *Nat Commun* 9, 3034 (2018).
2. Fan, J., Sun, W., Zhou, K. & Chen, D. Major Function Oriented Zone: New method of spatial regulation for reshaping regional development pattern in China. *Chinese Geogr Sci* 22, 196–209 (2012).
3. Locke, H. et al. Three global conditions for biodiversity conservation and sustainable use: an implementation framework. *National Science Review* 6, 1080–1082 (2019).

16. (Ins 354-355) *How so? What evidence did the authors present apart from the focus of their study, biodiversity/habitat impacts?*

Response:

Our findings suggest that SSP1 is the most optimal choice for urban development and for biodiversity/habitat conservation. For instance, there is smaller habitat loss, least habitat fragmentation, and lower local species richness loss in scenario SSP1. Popp et al. (2017) provided some indirect evidence that supports this conclusion. Specifically, they found that in SSP1, with low demands for agricultural goods and high intensification of agricultural production, agricultural land decreases and is significantly lower than in the other SSPs. Notably, the low demand for agricultural commodities, rapid growth in agricultural productivity, and globalized trade, largest forest area, all of which are most pronounced in scenario SSP1, and have the potential to sustain natural ecosystems and biodiversity. Thus, SSP1 is most conducive to sustainable development, including sustainable food consumption, rapid increase in agricultural productivity, and forest conservation. According to the estimation of Popp et al. (2017), cropland area is lowest in SSP1 and highest in SSP3, leading to higher forest area in SSP1 and lower forest area in SSP3, and cumulative CO₂ emissions from land use change are lowest in SSP1 and highest in SSP3. Moreover, Schipper, et al. (2020) found that SSP1 scenario is the most optimal choice for biodiversity conservation.

We have now cited other evidence that supports our conclusions (p.17, line 299–304):

“In particular, if the sustainable pathway (i.e., scenario SSP1) of urban expansion is properly implemented, humans will be able to maintain a relatively low natural habitat loss, low habitat edge density, and a high level of species conservation (SSP1 also is overall better for biodiversity with lowest cropland expansion and higher forest area according to several existing estimations^{30,31}).”

References:

1. Popp A, Calvin K, Fujimori S, et al. Land-use futures in the shared socio-economic pathways. *Global Environmental Change*, 42:331-345 (2017).
2. Schipper, A. M. et al. Projecting terrestrial biodiversity intactness with GLOBIO 4. *Glob Change Biol* 26, 760-771 (2020).

17. (Ins 382-383) Please explain briefly how the two differ.

Response:

To clarify, our estimated habitat loss due to future urban expansion (11–33 million ha by 2100) is smaller than that found by McDonald et al. (2019) (29 million ha between 2000 and 2030). Notably, the estimation of McDonald et al. (2019) was based on the urban growth forecasts by Seto et al. (2012), who tended to overestimate urban growth. We have now described the specific differences in the Discussion section (pp. 20–21, line 377–388).

“Compared to McDonald et al.⁹, our estimated habitat loss due to future urban expansion (11–33 million ha by 2100) is smaller than theirs (which had 29 million ha between 2000 and 2030). The main reason may be that the projection of urban expansion used by McDonald et al.⁹ is up to 2030 derived from Seto et al.¹⁵ and this projection did not fully capture the dynamics of future urban land use due to coarse spatial resolution, and did not consider future population decline trend that may cause stagnation in urban land growth. For example, the simulation that our research was based on showed that most regions in the world will continue to experience urban expansion, but a few regions (e.g., China and other Asian countries) will have a relatively low urban expansion rate or even a decline in urban land demand after 2050 (see Supplementary Figs. 7–11 for China), which is different from Seto et al.¹⁵.”

Supplementary **Fig. 25** Global comparison between urban expansion forecasts of Seto et al. (2012) and our projection result under SSP2. (A) China, (B) India, (C) East Africa, and (D) West Africa.

We have also illustrated some of the details in Supplementary Figs 25–28. Seto et al.’s (2012) overestimations of urban growth in Asia, Africa, and South America are large. In particular, 63% of

the ecoregions were overestimated in terms of urban expansion. There are more than 4 times differences between the estimated urban growth of Seto et al. (2012) and our projections for 14% of the ecoregions.

Supplementary Fig. 26 Difference ratio of urban expansion forecasts between Seto et al. (2012) and our SSP2 projection results based on Chen et al. (2020).

Supplementary Fig. 27 Difference of urban details for some metropolitan areas around the world for the year 2030 using 1-km resolution (our results based on Chen et al., 2020) and 5-km resolution (Seto et al., 2012).

Supplementary Fig. 28 The difference in projected urban growth by Seto et al. (2012) and ours (based on Chen et al., 2020) in the Mount Cameroon and Bioko montane forests. White lines indicate the boundary of the ecoregion.

18. (lms 386-389) Did the authors generate the urban expansion forecasts or used published forecasts? From what is stated under “Forecasting future urban expansion” in Methods, it is not clear if they used existing forecasts or created their own. Therefore, please clarify and rephrase.

Response:

We used published forecasts developed by Chen et al. (2020). The co-corresponding author of our paper, Xiaoping Liu, was also a contributing author of that paper. To avoid misinterpretation, we have now rephrased the section “Forecasting future urban expansion” in the Methods section (pp. 25–27, line 486–513) and the sentence in the Discussion (p. 20, line 371–375).

“We base our study on the global projection of urban expansion dataset with five SSP scenarios (see detailed assumptions about urbanization patterns and urban planning for five SSPs in Supplementary Tables 8 and 9) and a 1-km resolution¹⁷. This dataset was developed using a panel data regression to estimate future urban land demand. First, Chen and colleagues built panel data regression using historical urban land data (obtained from the GHSL dataset⁵⁸

for the years 1975, 1990, 2000 and 2014) and statistical data to estimate per capita urban land demand from urbanization rate (i.e., urban population/total population) and per capita gross domestic product (GDP) (obtained from the World Bank⁵⁹ and United Nations⁶⁰). Then, the established panel data regression model was used to predict, for each scenario, per capita urban land demand from the future per capita GDP and urbanization data obtained from the SSP database (<https://tntcat.iiasa.ac.at/SspDb>). Thus, the regional urban land demand in a future year t was obtained by multiplying the estimated urban land demand per capita at year t by the projected total population in a region at year t . The data used in the regression are for the 32 macro regions that were created in the SSP database by aggregating the world's countries or regions. Subsequently, the Future Land-Use Simulation (FLUS) model³⁹ was employed to allocate and simulate the spatially-explicit distribution of future urban expansion based on the forecasted urban land demand and the urban development potential for the 32 regions. It is assumed that even if the population shrinks, the urbanization rate and GDP can still grow, and there is still a certain urban land demand. Only when population, urbanization, and economic growth all stagnate, the urban land demand will stagnate, and then urban land may not continue to expand. Thus, the scenario of regional population shrinking but urban area growing can be captured by this model framework (for example, cities in Eastern Europe, see Supplementary Figs. 29 and 30). For regions with a decline in urban land demand, it was assumed that the land conversion from nonurban land to urban land is irreversible. In the spatial simulations, the substantial conversion of urban land to non-urban land will not occur. It is worth noting that in the urban growth simulation, if the estimated urban land area of a region in the future is smaller than its current urban area, then the existing urban lands will remain unchanged because future urban land demand can be met by the existing area¹⁷.”

“To reduce overestimation, we used more recent datasets with higher resolution (1 km) and longer time span (2015–2100). Besides, we adopted a more advanced Future Land-Use Simulation (FLUS) model that can couple with the latest SSPs¹⁷ to simulate future urban expansion³⁹.”

19. (Ins 418-421) *I am not convinced that the study has significant theoretical implications. As the authors state at the end of the paragraph, it suggests significant impacts on biodiversity/habitats from future urban expansion but does not advance any findings to inform theory, at least not any more than previous studies have already done.*

Response:

We have removed the discussions about the theoretical implications of our study to avoid over-interpretation.

20. (Ins 424-449) *Here, the authors appear to put forward a few ideas although most of the paragraph repeats what is already known from previous studies. For example, one of these “update the IUCN Protected Areas Management Category” is interesting but needs further elaboration. Other two, “upgrade the management level of key biodiversity areas, and prevent development encroachment (urban expansion) on protected areas.” have already been suggested. This also raises an important issue regarding the urban expansion forecasts. They were developed assuming there is no protection in the PAs from urban expansion. However, the enforcement varies widely around the world and none*

of these are addressed and discussed in the paper.

Response:

We have revised the paper in several respects to address your comment here.

First, we have further elaborated “update the IUCN Protected Areas Management Category” (pp. 24–25, line 469–474).

“Fourth, to facilitate wider coordination, we propose to update the IUCN Protected Areas Management Category. This can be achieved by (1) assigning IUCN categories to protected areas with unknown or missing category (roughly 30% of all protected areas) and increasing the strictness of protection areas⁴⁶, and (2) developing a new category system of protected areas to represent the specific role of different protected areas in biodiversity conservation rather than for management purposes⁵⁴.”

Second, we have removed “upgrade the management level of key biodiversity areas, and prevent development encroachment (urban expansion) on protected areas.” that has been suggested in previous research.

We have added extra discussions about variations in the enforcement around the world (pp. 22–23, line 422–437).

“Fourth, because urban expansion has occurred within protected areas in the past few decades¹⁹, we assumed that this trend of urban encroachment within protected areas will continue without restrictions within protected areas in the future. However, our work is subject to one caveat: We may have overestimated the impacts of urban expansion on protected areas, especially on direct encroachment, because the protected areas around the world vary in their enforcement effectiveness in preventing urban encroachment and alleviating human pressure^{33,44}. The enforcement effectiveness of protected areas depends on multiple factors, such as resources used to manage protected areas, law enforcement, and governance quality⁴⁵. However, it is still difficult to accurately identify future enforcement effectiveness of protected areas, because protected area downgrading, downsizing, and degazettement (PADDD) are becoming increasingly prevalent in some developed countries (e.g., the United States)³⁴. Even strictly protected areas (i.e., IUCN categories I and II) are subject to increasing human pressure, which suggests that the IUCN management category cannot inhibit the aggravation of human pressure⁴⁶.”

References:

1. Boitani, L. et al. Change the IUCN protected area categories to reflect biodiversity outcomes. *PLoS Biol* 6, e66 (2008).
2. Jones, K. R. et al. One-third of global protected land is under intense human pressure. *Science* 360, 788-791 (2018).
3. Geldmann, J., Manica, A., Burgess, N. D., Coad, L. & Balmford, A. A global-level assessment of the effectiveness of protected areas at resisting anthropogenic pressures. *Proc Natl Acad Sci U S A* 116, 23209-23215 (2019).
4. Gray, C. L. et al. Local biodiversity is higher inside than outside terrestrial protected areas

- worldwide. *Nat Commun* 7, 12306 (2016).
5. Güneralp, B. & Seto, K. C. Futures of global urban expansion: uncertainties and implications for biodiversity conservation. *Environ Res Lett* 8, 014025 (2013).
 6. Watson, J. E., Dudley, N., Segan, D. B. & Hockings, M. The performance and potential of protected areas. *Nature* 515, 67-73 (2014).
 7. Golden Kroner, R. E. et al. The uncertain future of protected lands and waters. *Science* 364, 881-886 (2019).

21. (Ins 452-463) *So did the authors use the forecasts reported in Ref. 16 of generate their own forecasts?*

Response:

Yes, we used published forecasts developed by Chen et al. (2020). The co-corresponding author of our paper, Xiaoping Liu, was also a contributing author of that paper. We have now clarified this in the paper to avoid confusion.

22. (Ins 464-478) *How about potential shifts in habitats and species ranges due to climate change? In a study that has reports potential impacts of urban expansion by 2100, the lack of any discussion on these is not acceptable.*

Response:

We agree that climate change can also cause potential shifts in habitats and species ranges. We have now discussed the potential impacts of climate change on biodiversity in the Discussion section (p. 23, line 437–451).

“Finally, it is plausible that both urban expansion and biodiversity change may interact with future climate change, which we did not investigate in our study. For instance, urban expansion can accelerate climate change (particularly the change in urban microclimate)⁴⁷, cause warming in urban and surrounding areas (urban heat island), and increase the intensity of precipitation and runoff in local areas⁴⁸. Moreover, future climate change—such as more extreme weather events in urban areas and faster sea-level rise in most coastal urban areas⁴⁸—can also affect urban development, urban environment, and urban expansion process^{49,50}. In addition, climate change affects all aspects of life on Earth, perhaps with the most pervasive impact on species redistribution⁵¹, such as poleward and elevational range shifts^{51,52}. Climate-driven changes in species redistribution, which may be more profound in the future when climate change intensifies, will affect global biodiversity pattern and shape new hotspots. Thus, future research on the effect of urban expansion on biodiversity needs to take into account the effect of climate change in this process.”

References:

1. Pecl, G. T. et al. Biodiversity redistribution under climate change: Impacts on ecosystems and human well-being. *Science* 355 (2017).
2. Scheffers, B. R. & Pecl, G. Persecuting, protecting or ignoring biodiversity under climate change. *Nature Climate Change* 9, 581–586 (2019).
3. Krayenhoff, E. S., Moustauoui, M., Broadbent, A. M., Gupta, V. & Georgescu, M. Diurnal interaction between urban expansion, climate change and adaptation in US cities. *Nature Climate*

Change 8, 1097–1103 (2018).

4. Kahn, M. E. Urban growth and climate change. *Annu Rev Resour Econ* 1, 333–350 (2009).
5. McDonald, R. I. et al. Urban growth, climate change, and freshwater availability. *Proc Natl Acad Sci U S A* 108, 6312–6317 (2011).
6. Doblus-Reyes, F. J. et al. Linking Global to Regional Climate Change. In: *Climate Change 2021: The Physical Science Basis. Contribution of Working Group I to the Sixth Assessment Report of the Intergovernmental Panel on Climate Change* [Masson-Delmotte, V., P et al., (eds.)]. Cambridge University Press. In Press., (2021).

23. *(Supplementary Fig.2 and Supplementary Fig.3) The captions are missing any information on what each map represents.*

Response:

We have now added the captions for Supplementary Fig. 3 (now is Supplementary Fig. 12). Supplementary Fig. 2 was divided into five maps (Supplementary Figs. 2–6) that represent different SSPs. We have also added the captions for each map.

24. *(Supplementary Fig.9) To which SSP forecasts do these maps belong?*

Response:

These maps belong to SSP5 forecasts. We now added the information about scenario SPP5 in the original Supplementary Fig. 9, which is now Supplementary Fig. 32.

Reviewers' Comments:

Reviewer #1:

Remarks to the Author:

My rereview of this manuscript has focused on the responses the reviewers made to my initial comments. Overall, they have made constructive efforts to discuss my major comments, and the manuscript is much improved, although I am not fully convinced in all cases with their response.

Concern 1: There analysis showing that urban growth sometimes occurred in gazetted PAs is interesting, and the inclusion of new data adds value to the manuscript. I think it is worth some additional discussion in the text between the spatial resolution of many PA boundaries, and that of the land-cover they are using for this comparison. In many cases, the PA boundaries within the WDPA do not fully resolve inholdings (existing towns or other ownership), so what they are observing is not necessarily conversion of protected habitat within PAs. Nevertheless, the analysis they present is helpful for showing that some conversion occurs within PAs. I would suggest two things:

- 1.) Rather than presenting the area of urban land within protected areas, please present the percent of all urban land development that occurs within protected areas. This normalizes between the very different land areas in different countries (an issue with interpreting Figure S13), and also puts the absolute urban area within PAs in context.
- 2.) Consider rerunning your analysis making the assumption urban development would not occur in protected areas. How does this change your results, quantitatively?

Concern 2: There is a helpful clarification by the authors of how their panel data regression works. The changes to the text partially mitigate my concerns with this analysis.

I am fine with the statement that "urban land demand" may well decline in China, as population declines. However, I want to ensure that there is no assumption that "urban land area" will decline (with regrowth of natural habitat). This was not the case in post-industrial cities like Pittsburgh or eastern Europe (impervious surface stagnated but did not sharply decline, even as population sharply declined). It sounds like from the authors clarification in their response to reviewers that they treat urban land area as staying constant if future urban land demand is less than currently. If so, that seems like an appropriate assumption to me.

Concern 3: Thanks for making the change.

Concern 4: I like the additional analyses here, and think they add value to the manuscript by showing how patterns of endemism intersect with patterns of urbanization and species loss.

Reviewer #3:

Remarks to the Author:

I found the revisions mostly satisfactory. However, while amended to some extent, the authors' need to be more measured in their claims remains. There are also sections of the text that make misleading (or outright incorrect) use of certain terms in landscape ecology or that use confusing language. I listed my detailed comments below:

1. (Ins 25-26) Abstract: The authors should be more measured in their claims. How future urban expansion will affect global biodiversity needs to be understood better but our understanding of it is not completely unclear as the sentence here implies. Also, this study contributes to filling the knowledge gap but by no means answers all the questions regarding impact of future urban expansion on global biodiversity.
2. (Ins 91-92) Suggest deleting the sentence. "Obvious" is too colloquial and sentence does not really convey any significant information.
3. (In 182) How do you define "edge regions"? These are areas in close proximity to natural habitats. I suggest you use this term (proximity or vicinity) instead of edge, which has a specific

meaning in landscape ecology. Also clearly define the buffer area you define as proximate or near the natural habitats.

4. (Ins 181-199) This part needs to be completely rewritten. It looks like the authors use the term edge in a completely different meaning than in ecology. What they refer to here are the areas in close proximity to natural habitats and the text should clearly reflect that. In its current form, it does not.

5. (Ins 324-325) Linked to the preceding comment: "making the natural habitats closer to the natural habitat edges" does not make sense at all. Edges are part of the natural habitats. I suspect the authors mean urban areas will get a lot closer to natural habitats but not sure. This part needs to be rewritten using correct terms in their correct meaning.

6. (In 332) I strongly urge the authors to refrain using engineering-like language such as "optimizing". Very misleading and assumes uncertainty plays no role.

7. (In 370) What is wrong with these models? You need to briefly specify here why your model is a better choice over these.

8. (Ins 377-389) The authors are correct that relatively lower spatial resolution in Seto et al forecasts are a significant factor in overestimating urban land expansion -thus habitat loss due to this- in that study. However, there are probably other factors as well behind the overestimation such as the data and the particular modeling approach. It is, however, unlikely that not accounting for population shrinkage projected to happen sometime after 2050 is one of these reasons. The overestimation would occur even if their forecasts went beyond 2050 (because of the irreversibility of urban land assumption).

9. (Ins 476-479) Please rephrase here to clarify what you mean by 'dominant'. Also, such strict measures rarely succeed primarily because the institutional structure to enforce these policies is weak. This is an issue the authors raise earlier in the manuscript. So, if you will keep these policy recommendations, you should also acknowledge that they are not easy to implement.

Reviewer #1

My rereview of this manuscript has focused on the responses the reviewers made to my initial comments. Overall, they have made constructive efforts to discuss my major comments, and the manuscript is much improved, although I am not fully convinced in all cases with their response.

Response:

Thanks again for your positive and constructive comments. We have incorporated all your comments and suggestions into our revisions to further strengthen our manuscript (see our point-to-point responses below). We hope you find this revision satisfactory.

1. Concern 1: There analysis showing that urban growth sometimes occurred in gazetted PAs is interesting, and the inclusion of new data adds value to the manuscript. I think it is worth some additional discussion in the text between the spatial resolution of many PA boundaries, and that of the land-cover they are using for this comparison. In many cases, the PA boundaries within the WDPA do not fully resolve inholdings (existing towns or other ownership), so what they are observing is not necessarily conversion of protected habitat within PAs. Nevertheless, the analysis they present is helpful for showing that some conversion occurs within PAs.

Response:

Thank you for raising this point. We agree with you that the boundaries of protected areas within the WDPA do not fully resolve inholdings (existing cities, towns or other ownership) in many cases. To ensure the robustness of our results, we further utilized high-resolution (with 30m resolution) global urban expansion datasets, including the dataset by Liu et al. (2020), Huang et al. (2021), Gong et al. (2020), and Pesaresi et al. (2016), to identify urban expansion or human settlement changes within PAs. We performed data fusion analysis on these four datasets. To reduce uncertainties, we treated the overlapping urban areas among at least three of these four datasets as pixels that were mapped as urban areas. Through spatial overlap analysis between terrestrial PA boundaries of WDPA and the urban expansion fusion datasets, we found a considerable amount of urban land or human settlement within PAs. This analysis revealed 1,9504 km² of urban land in PAs, which accounts for 2.04% of the total urban area of the world. Moreover, we further confirmed that many urban areas are growing within the PAs, and protected habitats within these PAs also experienced obvious conversion. Specifically, 38% of the urban land use change (4,862 km²) within PAs was due to the conversion of natural habitats into urban land between 1992 and 2015 based on the Climate Change Initiative Land Cover (CCI-LC) data. We also observed some conversions of protected habitat into urban land within PAs on a global scale. This evidence indicates that urban growth in gazetted PAs does exist and is less likely to be caused by the difference in spatial resolution between protected areas boundaries and land-cover data (especially given that we used the 30 m high-resolution datasets).

To highlight this interesting point and to ensure the robustness of our results, as you suggested, we have now included this analysis in the Results section (p.8 lines 143-145) and in the Supplementary Information (p. 8, Supplementary Note 5):

“Moreover, 38% of the urban land use changes within protected areas were due to the conversion

of natural habitats into urban land between 1992 and 2015 based on the CCI-LC data.”

“Supplementary Note 5: Global urban expansion in protected areas

Protected areas serve as the core tool and cornerstone of global biodiversity conservation, yet the protected area boundaries within the World Database on Protected Areas (WDPA) do not fully resolve inholdings (e.g., existing cities, towns, or private ownership of lands). To ensure the robustness of our results, we further utilized high-resolution (with 30m resolution) global urban expansion datasets²⁰⁻²³ to identify urban expansion or human settlement changes within protected areas. Through overlapping analysis between terrestrial protected area boundaries of WDPA and urban expansion datasets (from 1972 to 2019), we still found a considerable amount of urban land or human settlement within protected areas (1,9504 km² and accounts for 2.04% of the total urban area of the world). Moreover, we confirm that many urban areas (or privately owned lands) are growing within the protected areas, and protected habitats within these protected areas also experienced obvious conversion. This is because 38% of the urban land use changes within protected areas were due to the conversion of natural habitats into urban land between 1992 and 2015 based on the CCI-LC data. Therefore, there is an urgent need to gradually reduce human disturbance and urban expansion within the protected areas, and to explore the coordinated symbiosis of urban development and biodiversity conservation.”

References:

- 1 Huang, X. et al. 30 m global impervious surface area dynamics and urban expansion pattern observed by Landsat satellites: From 1972 to 2019. *Science China Earth Sciences*, 64, (2021).
- 2 Liu, X. et al. High-spatiotemporal-resolution mapping of global urban change from 1985 to 2015. *Nature Sustainability* 3, 564-570, (2020).
- 3 Gong, P. et al. Annual maps of global artificial impervious area (GAIA) between 1985 and 2018. *Remote Sensing of Environment* 236, 111510, (2020).
- 4 Pesaresi, M. et al. Operating procedure for the production of the Global Human Settlement Layer from Landsat data of the epochs 1975, 1990, 2000, and 2014. *EUR-OP*, 1-62 (2016).

I would suggest two things:

1.) Rather than presenting the area of urban land within protected areas, please present the percent of all urban land development that occurs within protected areas. This normalizes between the very different land areas in different countries (an issue with interpreting Figure S13), and also puts the absolute urban area within PAs in context.

Response:

Following your suggestion, we have now presented the percent of all urban land development that occurs within protected areas (see Figure S13 and S14). We have also added the absolute urban area within PAs and the number of PAs affected by urban land in the figure captions. We found that the mean percent of urban land that occurs within protected areas increased from 1.24% in 1992 to 2.65% in 2015. The number of protected areas affected by urban land increased from 21,217 in 1992 to 35,161 in 2015. Urban areas within in PAs increased from 8,290 km² in 1992 to 20,625 km² in 2015.

Supplementary Fig. 13 The percent of urban land within protected areas in 1992. The urban area within PAs was 8,290 km² in 1992. The number of protected areas affected by urban land was 21,217. The number and total area of protected areas with IUCN categories I and II were 558 and 579.31 km², respectively. These results were based on global LC 1992 map produced by the European Space Agency (ESA) Climate Change Initiative (CCI) with a resolution of 300 meters.

Supplementary Fig. 14 The percent of urban land within protected areas in 2015. The urban area within PAs was 20,625 km² in 2015. The number of protected areas affected by urban land was 35,161. The number and total area of protected areas with IUCN categories I and II were 813 and 1229.95 km², respectively. These results were based on global LC 2015 map produced by the European Space Agency (ESA) Climate Change Initiative (CCI) with a resolution of 300 meters.

2.) Consider rerunning your analysis making the assumption urban development would not occur in protected areas. How does this change your results, quantitatively?

Response:

We rerun our analysis assuming that urban development would not occur in protected areas. This analysis revealed that by 2100, urban expansion within protected areas will be reduced by 16,111–59,306 km² across five SSP scenarios, which accounts for 4.59–8.02% of newly-added urban land between 2015 and 2100. According to our model specification, if these urban areas would not occur in protected areas, they will equivalently occur in other potential areas of urban development. For global urban expansion, changing the assumption regarding the occurrence of urban expansion within protected areas will have a certain, but not fundamental, impact on the global urban expansion forecast. However, it is undeniable this may have a substantial impact on some regions, such as European countries.

Following your suggestions, we have discussed about this additional results in the Discussion section where we acknowledged the limitations and unaddressed issues of our research (p. 23 lines 438–441):

“If we assume no urban development in protected areas, then by 2100, urban expansion within protected areas is expected to decrease by 16,111–59,306 km² across SSP scenarios, which accounts for 4.59–8.02% of newly-added urban expansion between 2015 and 2100. Nevertheless, according to our model assumption, these urban areas will equivalently occur in other potential areas if they do not occur in protected areas.”

2. Concern 2: There is a helpful clarification by the authors of how their panel data regression works. The changes to the text partially mitigate my concerns with this analysis.

I am fine with the statement that "urban land demand" may well decline in China, as population declines. However, I want to ensure that there is no assumption that "urban land area" will decline (with regrowth of natural habitat). This was not the case in post-industrial cities like Pittsburgh or eastern Europe (impervious surface stagnated but did not sharply decline, even as population sharply declined). It sounds like from the authors clarification in their response to reviewers that they treat urban land area as staying constant if future urban land demand is less than currently. If so, that seems like an appropriate assumption to me.

Response:

Thanks for your positive and constructive comments. Your understanding is right. We do assume that urban land area is set constant if future urban land demand is less than the current demand. The existing evidence shows that our assumption is appropriate.

3. Concern 3: Thanks for making the change.

Response:

Thank you for your positive comments about our previous revision.

4. Concern 4: I like the additional analyses here, and think they add value to the manuscript by

showing how patterns of endemism intersect with patterns of urbanization and species loss.

Response:

Thank you for your positive comments about our previous revision.

Reviewer #3

I found the revisions mostly satisfactory. However, while amended to some extent, the authors' need to be more measured in their claims remains. There are also sections of the text that make misleading (or outright incorrect) use of certain terms in landscape ecology or that use confusing language. I listed my detailed comments below:

1. (lns 25-26) Abstract: The authors should be more measured in their claims. How future urban expansion will affect global biodiversity needs to be understood better but our understanding of it is not completely unclear as the sentence here implies. Also, this study contributes to filling the knowledge gap but by no means answers all the questions regarding impact of future urban expansion on global biodiversity.

Response:

Thank you for your suggestion. We have now revised this sentence into “However, how future urban expansion will affect global biodiversity needs to be better understood. We contributed to filling this knowledge gap by...”. (p. 2, lines 27–28)

2. (lns 91-92) Suggest deleting the sentence. “Obvious” is too colloquial and sentence does not really convey any significant information.

Response:

Thank you for your suggestion. We have now deleted this sentence.

3. (ln 182) How do you define “edge regions”? These are areas in close proximity to natural habitats. I suggest you use this term (proximity or vicinity) instead of edge, which has a specific meaning in landscape ecology. Also clearly define the buffer area you define as proximate or near the natural habitats.

Response:

Sorry for causing this confusion and thank you for your suggestions. Using the term—“edge regions” in our manuscript, we intended to say that we investigated the impact of future urban expansion on the nearest distance between urban areas and natural habitat (see Supplementary Fig. 31). That is to say, we only calculated the distance changes from patch edges of urban areas to patch edges of the nearest natural habitats, but we did not analyze the impact of urban expansion on the core areas of natural habitats. In landscape ecology, core area represents the interior area of patches of natural habitats after a specified edge buffer is eliminated. Thus, we did not define the buffer area in this revision. To avoid confusion, we have changed “edge regions adjacent to natural habitat” into “the nearest distance between urban areas and natural habitat (i.e., the distance from patch edges of urban areas to patch edges of the nearest natural habitats)” (p. 10 lines 183–187). We hope you agree that this revision has

clarified our message.

Supplementary Fig. 31 Illustration of changes in the Euclidean nearest distance between urban areas and natural habitat due to future urban expansion. Euclidean nearest distance between urban land and natural habitat gradually decreases from t_1 to t_2 due to rapid urban expansion.

4. (Ins 181-199) This part needs to be completely rewritten. It looks like the authors use the term *edge* in a completely different meaning than in ecology. What they refer to here are the areas in close proximity to natural habitats and the text should clearly reflect that. In its current form, it does not.
Response:

Sorry for causing this confusion. The term “edge” in this paragraph refers to the patch edges. We have now clearly defined this term and substantially revised this section (lines 181–203).

“The increasing exposures of natural habitat to urbanized land use may cause long-term changes in the function and structure of the natural habitat that is adjacent to urban areas¹³. To examine proximity effect, we investigated the impact of future urban expansion on the nearest distance between urban areas and natural habitat (i.e., the distance from patch edges of urban areas to patch edges of the nearest natural habitats) under different SSP scenarios. Although the global urban areas are expected to increase by 36–74 Mha by 2100, the impacts of future urban expansion on adjacent natural habitat are disproportionately large. The future urban expansion will make urban areas much closer to patch edges of 34–40 Mha natural habitat, which will inevitably threaten the natural habitat and increase the risk of biodiversity decline. The effects of urban expansion on adjacent patch edges of natural habitats are remarkably different across different scenarios. Specifically, the area of affected adjacent natural habitat is expected to be 38.45 Mha, 34.24 Mha, 40.31 Mha, 37.84 Mha, and 39.42 Mha under SSP1 to SSP5 scenarios by 2100, with the smallest effect under scenario SSP2, and the largest effect under scenario SSP3. Moreover, the scale of urban expansion does not correspond directly with the size of the impact.

Several countries, including Mauritania, Algeria, Saudi Arabia, Western Sahara, and the United States, will have a large change in the distance from future urban areas to natural habitats due to urban expansion (Supplementary Table 5). Such effects also varied across different natural habitat types. The distance from the patch edges of urban areas to patch edges of (a) wetland, other land, and forest, (b) grassland, and (c) shrubland will generally be shortened by ~2000 m, ~1500 m, and ~900 m, respectively.”

5. (In 324-325) *Linked to the preceding comment: “making the natural habitats closer to the natural habitat edges” does not make sense at all. Edges are part of the natural habitats. I suspect the authors mean urban areas will get a lot closer to natural habitats but not sure. This part needs to be rewritten using correct terms in their correct meaning.*

Response:

Thank you for pointing out this error in writing. We have now corrected this mistake by changing “making the natural habitats closer to the natural habitat edges...” into “making the urban areas closer to the patch edges of natural habitat...”. (lines 328-329)

6. (In 332) *I strongly urge the authors to refrain using engineering-like language such as “optimizing”. Very misleading and assumes uncertainty plays no role.*

Response:

Thank you for raising this issue. We have now changed “optimizing” into “reshaping”. We hope you agree that this is an appropriate revision.

7. (In 370) *What is wrong with these models? You need to briefly specify here why your model is a better choice over these.*

Response:

Thank you for pointing this out. We have now briefly specified why our model is a better choice in this revision (pp. 20-21 lines 372-389):

“Second, previous studies on the effects of urban expansion often used datasets with lower resolution (e.g., 5 km) to simulate future urban land changes. Yet, low-resolution data can cause overestimation of future urban expansion³⁸. To reduce overestimation, we used more recent datasets with higher resolution (1 km). Besides, we adopted a more advanced Future Land-Use Simulation (FLUS) model that can couple with the latest SSPs¹⁷ to simulate future urban expansion³⁹. This model can explicitly simulate the spatial trajectories of multiple land cover changes under different scenarios by coupling human-related and natural environmental impacts³⁹. The characteristics and advantages of the FLUS model are the self-adaptive inertia mechanism and roulette selection mechanism, which can reflect the complexity and uncertainty of land use changes in the real world. These advantages are not available in other models (e.g., SLEUTH model²³ and SELECT model¹⁸). In particular, these other models often set more priorities to the edge growth transition rule or existing urban areas and thus are limited in simulating other urban development processes, such as leapfrog development pattern. Moreover,

these other models often simplified the randomness and complexity of the urban expansion process and thus affects the simulation performance. In addition, the spatial allocation algorithm embedded in these models may not effectively capture urban population shrinkage.”

8. (Ins 377-389) *The authors are correct that relatively lower spatial resolution in Seto et al forecasts are a significant factor in overestimating urban land expansion -thus habitat loss due to this- in that study. However, there are probably other factors as well behind the overestimation such as the data and the particular modeling approach. It is, however, unlikely that not accounting for population shrinkage projected to happen sometime after 2050 is one of these reasons. The overestimation would occur even if their forecasts went beyond 2050 (because of the irreversibility of urban land assumption).*

Response:

Thank you for your comments. The projection of urban expansion of Seto et al. did not fully capture the dynamics of future urban land use. This may be due to coarse spatial resolution, the specific datasets they used, or their particular model configurations. A relatively lower spatial resolution (5km) in Seto et al forecasts is a significant factor in overestimating urban land expansion. The datasets used in Seto et al forecasts also proved to have some limitations. For instance, the Global Rural-Urban Mapping Project dataset has the blooming effect due to the use of night-time light data and the insufficient detection of small settlements. Moreover, the model of Seto et al. assumes that urban land continues to increase. Based on this model assumption, they did not capture future population decline trend that may cause stagnation in urban land growth.

To make this clearer, we have revised this section (p. 21 lines 390–404). We hope you find this revision satisfactory.

“Compared to McDonald et al.⁹, our estimated habitat loss due to future urban expansion (11–33 million ha by 2100) is smaller than theirs (which had 29 million ha between 2000 and 2030). For example, the simulation that our research was based on showed that most regions in the world will continue to experience urban expansion, but a few regions (e.g., China and other Asian countries) will have a relatively low urban expansion rate or even a decline in urban land demand after 2050 (see Supplementary Figs. 7–11 for China), which is different from Seto et al.¹⁵. Notably, the projection of urban expansion used by McDonald et al.⁹, which is up to 2030 derived from Seto et al.¹⁵, did not fully capture the dynamics of future urban land use. This may be due to coarse spatial resolution, the specific datasets they used, or their particular model configurations. The model of Seto et al.¹⁵ assumes that urban land continues to increase. Thus, they might overestimate urban growth (Supplementary Figs. 25–28). Indeed, out of the 30 conservation priority ecoregions, we identified 19 ecoregions with high species number of vertebrates but high future urban growth potential, which were not covered by the result of McDonald et al.²⁰.”

9. (Ins 476-479) *Please rephrase here to clarify what you mean by ‘dominant’. Also, such strict measures rarely succeed primarily because the institutional structure to enforce these policies is weak. This is an issue the authors raise earlier in the manuscript. So, if you will keep these policy recommendations, you should also acknowledge that they are not easy to implement.*

Response:

Thank you for pointing this out. To clarify, ecosystems have many functions, one of which must be the primary function. For example, the primary function of an ecological function area is to provide ecological services, while its secondary function is to provide agricultural products, industrial and other products. If an ecological function area does not serve its primary function, then its capacity to provide ecological services may be impaired. Therefore, it is necessary to distinguish the primary functions of ecosystems, and determine the main tasks according to the primary functions. We have revised this section (pp. 25-26 lines 492–507). We hope you find this revision satisfactory.

“Finally, we suggest that a new framework for biodiversity conservation should be established based on human pressures and the primary functions of ecosystems^{55,56}. Despite its multifunctional nature, an ecosystem often serves a primary function. For example, the primary function of an ecological function area is to provide ecological services, while its secondary function is to provide agricultural products, industrial and other products. If an ecological function area does not serve its primary function, then its capacity to provide ecological services may be impaired. In this framework, the ecosystem can be divided into three types of areas based on its primary function: urban area, agricultural area, and ecological area. Accordingly, different governance measures can be implemented to balance the relationship between urban development, food security, and biodiversity conservation⁵⁷. In this way, global and local biodiversity conservation responsibilities can be effectively identified in a differentiated and explicit way. However, it is undeniable that such strict measures are not easy to implement, which is primarily because the current institutional structure to enforce these policies is weak.”

Reviewers' Comments:

Reviewer #1:

Remarks to the Author:

My rereview of the manuscripts has focused on the authors response to my comments. Overall, they have made good faith efforts to respond to most of my concerns, and I think the manuscript is improved.

Concern 1: The additional analysis the authors provide here is helpful, and certainly addressed the spatial resolution issues I highlighted originally. My only suggestion at this point is that the authors review their language in the manuscript and ensure that they are not assuming that ALL urban land within protected areas is bad ecological, nor that ALL urban land expansion is a protected area is a violation of the legal protection of the PA. For instance, it is quite common in Europe to have protected areas whose boundaries encompass a core block of natural habitat, in addition to a network of agriculture fields and small villages, with an associated transportation network. Even where the rule of law is enforced, it is quite possible in situations like this to have expansion in the agricultural or village areas, which would show up in the datasets the authors are using as a conversion. To give a specific example, I live near a protected area in France that has an existing highway passing through it that has recently been expanded. This is not encroachment in a legal sense (the road right of way is exempt from the land protection), but since the road is inside the gazetted area, the authors would count this as a land conversion event, and it certainly might limit ecological connectivity in the park. To be clear, I am not suggesting any new analysis. Rather, the authors should just check their language describe their finding, to make sure it is appropriately caveated- what they are showing is change in land cover within park boundaries, and they cannot infer that this is necessarily encroachment in a legal sense.

Concern 2: The authors have clarified their current approach, which I appreciate. I have no further concerns here.

Concern 3: Resolved in the last round of revisions.

Reviewer #3:

Remarks to the Author:

I found the revisions satisfactory except one significant issue: The only major thing left to address is the authors' attempt to lay out a framework at the very conclusion of their paper. They are clearly out of their expertise here so I strongly urge them to leave out that text. Leaving it in the paper will only show they do not really know what they are talking about and it is a very unfortunate way to conclude a paper that otherwise rests on a strong analysis of future impacts of urban expansion. Their paper is not about what sort of framework would be needed to address the trends they identify in their study. Also included in the list below are three minor corrections:

1. (p3 lns51-53) Aichi Targets were put forward to be met by 2015 or 2020. This sentence should be revised to reflect that most of these were not met by 2020.
2. (p18 ln 330) Suggest replacing "making the urban areas closer" with "as urban areas get closer".
3. (p10 lns 181-199) The authors repeat that the model used in Seto et al (2030) did not account for population shrinkage. This is correct; however, it is irrelevant in explaining the overestimation in their forecasts because their forecasts go out to 2030 and no population shrinkage is predicted anywhere around the world by that year. If what they allude to is that their model allows for shrinkage, that too is controversial as it is a very slow process for a piece of urban land to be completely reverted back to 'nature' once it is abandoned. Nothing to change in response to this in the text; but I wanted to highlight that not accounting for population shrinkage in Seto et al is not relevant here.
4. (pp25-26 lns 494-507) I strongly urge the authors to remove the text. They are clearly out of their depth discussing the various 'uses' of ecosystems and how to govern them. Best for their sake is to leave that out of their paper. There are many who will take issue -and rightly so- with what is written here. The contents of the text here can variously be called confused (e.g., when they are referring to industrial uses of an ecosystem) or incorrect (e.g., the division among urban

area, agricultural area, and ecological area assuming the former two have no ecological value). Most importantly, the paper is not about coming up with a governance framework. Just end your otherwise strong paper by writing that governing conflicting demands of consumption on ecosystems and ensuring their integrity is a very challenging task to be addressed. And leave it there.

Reviewer #1 (Remarks to the Author):

My rereview of the manuscripts has focused on the authors response to my comments. Overall, they have made good faith efforts to respond to most of my concerns, and I think the manuscript is improved.

Response:

Thanks again for your positive and constructive comments. We have further revised our manuscript based on your additional comments and suggestions (see our point-to-point responses below). We hope you find this revision satisfactory.

Concern 1: The additional analysis the authors provide here is helpful, and certainly addressed the spatial resolution issues I highlighted originally. My only suggestion at this point is that the authors review their language in the manuscript and ensure that they are not assuming that ALL urban land within protected areas is bad ecological, nor that ALL urban land expansion in a protected area is a violation of the legal protection of the PA. For instance, it is quite common in Europe to have protected areas whose boundaries encompass a core block of natural habitat, in addition to a network of agriculture fields and small villages, with an associated transportation network. Even where the rule of law is enforced, it is quite possible in situations like this to have expansion in the agricultural or village areas, which would show up in the datasets the authors are using as a conversion. To give a specific example, I live near a protected area in France that has an existing highway passing through it that has recently been expanded. This is not encroachment in a legal sense (the road right of way is exempt from the land protection), but since the road is inside the gazetted area, the authors would count this as a land conversion event, and it certainly might limit ecological connectivity in the park. To be clear, I am not suggesting any new analysis. Rather, the authors should just check their language describe their finding, to make sure it is appropriately caveated- what they are showing is change in land cover within park boundaries, and they cannot infer that this is necessarily encroachment in a legal sense.

Response:

Thank you for raising this point. We agree with you that not ALL urban land within protected areas is bad in an ecological sense, nor that ALL urban land expansion in protected area is a violation of the legal protection of the protected areas. Following your suggestion, we have checked our language throughout the manuscript, particularly the Results section, and have revised our wordings whenever possible. For instance, we have changed “encroach” and “encroachment” into “impact”, “expansion” or “occur in” (see pp.8-9, lines 143–157). We have also added a few sentences to address this concern in the Discussion section (p.25, lines 464-468):

“(b) it should be noted that not all urban land within protected areas has negative consequences in an ecological sense, nor that all urban expansion in protected areas reflects a violation of the legal protection of protected areas. We should understand urban expansion in protected areas differently and seek potential solutions to sustain the harmonious coexistence of human and nature in the future”.

Concern 2: The authors have clarified their current approach, which I appreciate. I have no further

concerns here.

Response:

Thank you for your positive comments about our previous revision.

Concern 3: Resolved in the last round of revisions.

Response:

Thank you for your positive comments about our previous revision.

Reviewer #3 (Remarks to the Author):

I found the revisions satisfactory except one significant issue: The only major thing left to address is the authors' attempt to lay out a framework at the very conclusion of their paper. They are clearly out of their expertise here so I strongly urge them to leave out that text. Leaving it in the paper will only show they do not really know what they are talking about and it is a very unfortunate way to conclude a paper that otherwise rests on a strong analysis of future impacts of urban expansion. Their paper is not about what sort of framework would be needed to address the trends they identify in their study. Also included in the list below are three minor corrections:

Response:

Thanks for your positive and constructive comments. To address your concern about the section on the framework on biodiversity conservation, we have removed this section. We have incorporated all your additional comments and suggestions into our revision in this round to further strengthen our manuscript (see our point-to-point responses below). We hope you find this revision satisfactory.

1. (p3 lns51-53) Aichi Targets were put forward to be met by 2015 or 2020. This sentence should be revised to reflect that most of these were not met by 2020.

Response:

Thank you for pointing this out. We have revised this sentence into “Most of the 20 Aichi Biodiversity Targets were not met by 2020 due to anthropogenic impacts, particularly the natural habitat loss and fragmentation caused by agricultural and urban land use changes.” (p.3 lines 53–55)

2. (p18 ln 330) Suggest replacing "making the urban areas closer" with "as urban areas get closer".

Response:

Following your suggestion, we have now revised the original sentence of “...future urban expansion will disproportionately affect the natural habitat around the urban areas, making the urban areas closer to the patch edges of natural habitat and thus increasing the risk of biodiversity loss” into “...future urban expansion will disproportionately affect the natural habitat around the urban areas as urban areas get closer to the patch edges of natural habitat, and thus increase the risk of biodiversity loss”. (p. 19, line 338–340)

3. (p10 lns 181-199) The authors repeat that the model used in Seto et al (2030) did not account for

population shrinkage. This is correct; however, it is irrelevant in explaining the overestimation in their forecasts because their forecasts go out to 2030 and no population shrinkage is predicted anywhere around the world by that year. If what they allude to is that their model allows for shrinkage, that too is controversial as it is a very slow process for a piece of urban land to be completely reverted back to 'nature' once it is abandoned. Nothing to change in response to this in the text; but I wanted to highlight that not accounting for population shrinkage in Seto et al is not relevant here.

Response:

Thank you for raising this point. We agree with you that population shrinkage is irrelevant in explaining the overestimation in Seto et al.'s forecasts given that their forecasts go out to 2030 and population shrinkage is unlikely to occur by 2030. To address your concern, we have carefully checked our paper and revised our writings to avoid this issue (see p. 22, lines 399–413).

“Compared to McDonald et al.⁹, our estimated habitat loss due to future urban expansion (11–33 million ha by 2100) is smaller than their estimation (i.e., 29 million ha between 2000 and 2030). Notably, the projection of urban expansion used by McDonald et al.⁹, which is up to 2030 derived from Seto et al.¹⁵. They might overestimate urban growth (Supplementary Figs. 25–28). This may be due to coarse spatial resolution, the specific datasets they used, or their particular model configurations. Indeed, out of the 30 conservation priority ecoregions, we identified 19 ecoregions with high species number of vertebrates but high future urban growth potential, which were not covered by the result of McDonald et al.²⁰. In addition, the spatial allocation algorithm embedded in these models may not effectively capture urban population shrinkage (except for the study of Seto et al.¹⁵, because their forecasts go out to 2030). For example, the simulation that our research was based on showed that a few regions (e.g., China and other Asian countries) will have a relatively low urban expansion rate or even a decline in urban land demand after 2050 (see Supplementary Figs. 7–11 for China).”

4. (pp25-26 lns 494-507) *I strongly urge the authors to remove the text. They are clearly out of their depth discussing the various 'uses' of ecosystems and how to govern them. Best for their sake is to leave that out of their paper. There are many who will take issue -and rightly so- with what is written here. The contents of the text here can variously be called confused (e.g., when they are referring to industrial uses of an ecosystem) or incorrect (e.g., the division among urban area, agricultural area, and ecological area assuming the former two have no ecological value). Most importantly, the paper is not about coming up with a governance framework. Just end your otherwise strong paper by writing that governing conflicting demands of consumption on ecosystems and ensuring their integrity is a very challenging task to be addressed. And leave it there.*

Response:

Thank you for raising this issue. We have now removed the text you mentioned and have ended the discussion with the sentence “Of course, it is undeniable that governing conflicting demands of consumption on ecosystems and ensuring their integrity is a very challenging task that requires joint effort from different stakeholders around the world.” (p. 27, lines 505-508)